



# Flex_extract v7.1 – A software to retrieve and prepare ECMWF data for use in FLEXPART

Anne Philipp[1,2], Leopold Haimberger[2], and Petra Seibert[3]

[1]Aerosol Physics & Environmental Physics, University of Vienna, Vienna, Austria
[2]Department of Meteorology and Geophysics, University of Vienna, Vienna, Austria
[3]Institute of Meteorology, University of Natural Resources and Life Sciences, Vienna, Austria

**Correspondence:** Anne Philipp (anne.philipp@univie.ac.at)

**Abstract.** Flex_extract is an open-source software package to efficiently retrieve and prepare meteorological data from the European Centre for Medium-Range Weather Forecasts (ECMWF) as input for the widely-used Lagrangian particle dispersion model FLEXPART and the related trajectory model FLEXTRA. ECMWF provides a variety of data sets which differ in a number of parameters (available fields, spatial and temporal resolution, forecast start times, level types etc.). Therefore, the

selection of the right data for a specific application and the settings needed to obtain them are not trivial. Therefore, the data sets which can be retrieved through flex_extract by both authorised member state users and public users and their properties are explained. Flex_extract 7.1 is a substantially revised version with completely restructured software, mainly written in Python3, which is introduced with all input and output files and for the four different application modes. Software dependencies and the methods for calculating the native vertical velocity $\dot{\eta}$, the handling of flux data and the preparation of the final FLEXPART

input files are documented. Considerations for applications give guidance with respect to the selection of data sets, caveats related to the land-sea mask and orography, etc. Formal software quality assurance methods have been applied to flex_extract. It comes with a set of unit and regression tests as well as code metric data. A short description of the installation and usage of flex_extract as well as information about available detailed documentation is also provided.

## 1 Introduction

The widely used off-line Lagrangian particle dispersion model (LPDM) FLEXPART (Stohl et al., 1998; Stohl et al., 2005; Pisso et al., 2019) and its companion, the trajectory model FLEXTRA (Stohl et al., 1995; Stohl and Seibert, 1998), require meteorological data in GRIB format as input. A software tool, flex_extract, is provided to retrieve and prepare these data from the Meteorological Archival and Retrieval System (MARS) of the European Centre for Medium-Range Weather Forecasts (ECMWF) to run FLEXPART. Because of specific requirements of FLEXPART and FLEXTRA and the variations between the

various ECMWF products, this is a complex task.

After the retrieval of the meteorological fields, flex_extract calculates, if necessary, the vertical velocity in the native coordinate system of ECMWF's Integrated Forecast Model (IFS), the so-called hybrid coordinate (Simmons and Burridge, 1981); furthermore, it calculates approximate instantaneous fluxes from the accumulated flux data provided by the IFS (precipitation and surface fluxes of momentum and energy). It also takes care of packaging and naming the fields as expected by FLEXPART





and FLEXTRA. The retrieval software is an integral part of the FLEXPART / FLEXTRA modeling system which is needed by users who apply the main branch based on the ECMWF meteorological fields (Pisso et al., 2019).

Flex_extract is an open-source software package with a history starting in 2003 which has undergone adaptations and extensions ever since. After the release of version 7.0.2, which was very specific as it could retrieve data only from a subset of ECMWF's products, the demand for additional data sources and to adapt to new versions of ECMWF's software packages used arose. Unfortunately, the existing code was not very flexible and thus difficult to maintain and expand. User friendliness was insufficient, as knowledge about flex_extract's driving parameters, the various ECMWF data sets, and their interaction was expected from users; with the increasing popularity of the FLEXPART model, improvements were necessary also in this respect. One of the priorities was to enable the extraction of fields from the reanalysis data sets ERA5 and CERA-20C. Additionally, the need for retrieving ensemble members in combination with forecast products arose. A recently developed new algorithm for disaggregating of the precipitation fields (Hittmeir et al., 2018) to improve the wet deposition calculation in FLEXPART should also be considered. With respect to ECMWF software packages on which flex_extract depends, the Meteorological Interpolation and Regridding library MIR superseded the previous EMOSLIB, a package called `eccodes` replaced `GRIB_API` for decoding and encoding GRIB messages.

Recently, ECMWF opened the access to selected reanalysis data sets for non-authorised, so-called public users from anywhere in the world, while before only users with a member-state account could access the data. Along with this change, two new web interfaces (ECMF's WebAPI and the Copernicus Data Service [CDS] WebAPI) were introduced, which allow to download data without direct access to ECMWF servers. This required a further adaptation, so that flex_extract can now be used also on a local host in combination with these APIs for both authorised and public users.

All these developments led to the new and totally revised version 7.1 of flex_extract introduced in this software description paper. It constitutes a more significant change of the code base than one might expect from the version number increment. The code was modularised for implementing software quality standards and as a prerequsite of the extension of the functionality. A more comprehensive set of test cases was developed, the documentation was significantly enhanced with more details. A big step forward was thus achieved in terms of user friendliness.

This paper contains the first published software documentation of flex_extract. The current version 7.1 replaces all previous versions which will no longer be supported.

## 1.1 Structure of the paper

Section 2 gives an overview of available ECMWF data sets and their accessibility for member-state and public user, respectively. The diversity of available data sets, possible combinations, and accessibility is a key piece of information for users. The code of flex_extract is described in Section 3. This is followed by considerations for application in Section 4, and the methods applied for the quality assurance in Section 5. The final remarks in Section 6 include information support options for users and plans for future development. The instructions for the installation and usage of the software are outlined in the Appendix.



## 1.2 The history of flex_extract

When the FLEXTRA model was developed in the 1990ies, one aim was to optimise its accuracy by avoiding unnecessary vertical interpolation. Therefore, it was implemented to directly use the three-dimensional wind fields on the IFS model levels

rather than fields interpolated to pressure levels as most other off-line trajectory and particle dispersion models do (Stohl et al., 1995; Stohl and Seibert, 1998). This also solves the issue of the lower boundary conditions over topography (trajectories should not interSect. the surface) in an optimum way. The IFS model uses a hybrid coordinate system, terrain-following near ground and approaching a pressure ($p$) based coordinate towards the model top; the vertical coordinate is called $\eta$ and thus the corresponding native vertical velocity is $\dot{\eta}$.

At that time, most ECMWF/IFS model fields were available on $\eta$-levels, however, $\dot{\eta}$ was not routinely stored in the MARS archive. Thus, a preprocessing software was needed to calculate accurate $\dot{\eta}$ values from available fields. A second motivation was the need of a chemical transport model (POP model, see Wotawa et al. (1998)) coupled with FLEXTRA and later of FLEXPART for instantaneous surface fluxes (latent and sensible heat, surface stresses, precipitation) instead of accumulated values of these fluxes as stored in MARS.

When the Comprehensive Nuclear Test Ban Treaty Organization (CTBTO) started to use FLEXPART operationally, a new version was created. It still consisted of Korn shell scripts and Fortran programmes for the numerically demanding calculation of $\dot{\eta}$. This was the first numbered version of flex_extract, v1, released in 2003. In version 2 (2006), it became possible to extract subregions of the globe and the Fortran code was parallelised with OpenMP. In Version 3, the option to use $\dot{\eta}$ from MARS, which became available for some forecast products from 2008 on, was introduced. Version 4 was neeeded to adapt the

software to the then new GRIB2 standard for meteorological fields. Versions 5 and 6 (2013) where adaptations to allow for higher horizontal resolutions and additional data sources, e. g. global reanalysis data. At this time, the Korn shell scripts had become quite complicated and difficult to maintain.

In 2015, the demand was raised to retrieve fields from long-term forecasts, not only analyses and short-term forecasts. At this stage, it was decided to rewrite flex_extract in Python2. The Python part controls the program flow by preparing and shell

scripts which are then submitted to ECMWF servers to start flex_extract in batch mode. The Fortran program for the calculation of the vertical velocity, `calc_etadot.f90` (previously also called `CONVERT2` or `preconvert`), was still used and called from the python code. Version 7.0.3 allowed to retrieve CERA-20C and ERA5 data, and introduced local retrieval of MARS data through the ECMWF Web API. Version 7.0.4 enabled the retrieval of multiple ensemble members at a time. It includes also bug fixes for ERA5 and CERA-20C data.

For the current version 7.1, the Python part was completely revised by refactoring and modularisation, and it was ported to Python3. Instead of ECMWF's `grib_api` for decoding and encoding GRIB messages, its successor `ecCodes` was implemented. The installation process has been simplified. In addition to the ECMWF Web API, also the new CDS API is supported. The disaggregation of precipitation data offers to alternatively use the new algorithm of Hittmeir et al. (2018) which maintains non-negativity and preserves the integral precipitation in each time interval. The code quality of flex_extract was improved by



adding a first set of unit tests and the introduction of regression tests. A new, detailed on-line documentation was created with Sphinx / FORD, hosted on the FLEXPART community website http://flexpart.eu.

### 1.3  FLEXPART and FLEXTRA

The FLEXible PARTicle model (FLEXPART) is one of the most widely used Lagrangian particle dispersion models (LPDM) for multi-scale atmospheric transport studies (Stohl et al., 1998; Stohl et al., 2005; Pisso et al., 2019) with a world-wide user

base. It is an open source model under the GNU General Public License (GPL) Version 3. As an off-line model, it requires meteorological fields (analysed or forecast) as input. Such data are available from numerical weather prediction (NWP) models and thus several model branches have been created for input from different models (Pisso et al., 2019). The main branch of the FLEXPART model is able to use data from the European Centre for Medium-Range Weather Forecasts' (ECMWF) Integrated Forecast System (IFS) and the US National Centre for Environmental Prediction's Global Forecast System (GFS). The data

extraction software flex_extract supports the ECMWF/IFS data extraction, which is generally considered the most accurate data source. As an LPDM, FLEXPART solves a Langevin equation for the trajectories of computational particles under the influence of turbulence (stochastic component) and quantifies changes to the trace substance mass or mixing ratio represent by these particles due to various processes. All details can be found in the literature mentioned.

Applications include a wide range of topics, such as air pollution, natural and man-made atmospheric radioactivity, stratosphere-

troposphere exchange, and atmospheric water cycle studies and airflow patterns. With the domain-filling mode the entire atmosphere can be represented by particles representing an equal share of mass.

FLEXTRA is a model that calculates simple trajectories as a function of fields of the mean 3D wind Stohl et al. (1995); Stohl and Seibert (1998). FLEXPART is based on it and shares some parts of the code. It ingests the same input fields in GRIB format as FLEXPART, thus it may be considered as a companion model. It is also free software and can be downloaded as well

from the FLEXPART community website.

Both FLEXTRA and FLEXPART can be used from within ECMWF's Metview software ECMWF (2019m).

### 2  ECMWF data

The European Centre for Medium-Range Weather Forecasts produces reanalysis data sets and global numerical weather predictions in operational service to its supporting Member States. All data are available to the national meteorological services

in the Member States and the Co-operating States. Some data sets are also publicly available (ECMWF, 2019a). The data are stored in GRIB or BUFR format in the Meteorological Archival and Retrieval System (MARS) ECMWF (2019b). The smallest addressable object is a meteorological field or an observation, grouped into logical entities such as "a forecast". These entities can be addressed through metadata organised in a tree-like manner. The meteorological fields are archived in one of three spatial representations: spherical harmonics (mainly model level fields), Gaussian grid (mainly surface fields, but also some

model level fields), or a regular latitude / longitude grid (ECMWF, 2019b).



## 2.1 Access to ECMWF

For the access to its MARS archive, ECMWF distinguishes two users groups: member-state and public users.

Member-state users have the possibility to work directly on the ECMWF Member State Linux servers as well as via a Web Access Toolkit (ECaccess) through a Member State gateway server. This mode provides full access to the MARS archive.
Nevertheless, there might be some limitations in user rights, particularly regarding current forecasts and ensemble forecasts. Member-state user accounts are granted by the Computing Representative of the corresponding Member State.

Public users access the ECMWF public data sets directly from their local facilities, anywhere in the world. The main differences to the member state users are the method of access – through a Web API – and the limited availability of data. Public users have to explicitly accept the license for the data set to be retrieved.
Member-state users may also access data via a Web API, without gateway server, in the same way as public users. The only difference is that different MARS databases are to be used. Flex_extract automatically chooses the correct ones.

Users can explore the availability of data in MARS via a web interface where they are guided through a stepwise selection of metadata. With this method, it is also possible to estimate the download size of a data set before actually retrieving it through flex_extract. There is a web interface "MARS Catalogue" for member-state users[1] with the full content and an interface "Public
data sets" for public users[2] with the subset of public data. The availability of data can also be checked by MARS commands on ECMWF servers. MARS commands[3] are used by flex_extract to retrieve the data on ECMWF servers.

## 2.2 Data sets available through flex_extract

ECMWF has a large variety of data sets varying in model physics, temporal and spatial resolution as well as forecast times. Only the subset of data which are most commonly used with FLEXPART can be retrieved through flex_extract. The accessible
data sets are:

1. The operational atmospheric high-resolution forecast (HRES),

2. the operational atmospheric ensemble forecast (ENS),

3. the ERA-Interim reanalysis,

4. the CERA-20C reanalysis, and

5. the ERA5 reanalysis.

Public users have access to the public version of ERA-Interim and CERA-20C Laloyaux et al. (2018) reanalysis. Even though ERA5 is in principle a public data set, currently only a subset can be accessed by public users. Unfortunately, it does not include the model-level fields which are essential for FLEXPART. Flex_extract is already prepared (requires only some code activation) to retrieve public ERA5 data as soon as model-level fields will be available.

---

[1]https://apps.ecmwf.int/mars-catalogue/; Last accessed: 17.08.2019

[2]https://apps.ecmwf.int/datasets/; Last accessed: 17.08.2019

[3]https://confluence.ecmwf.int/display/UDOC/MARS+command+and+request+syntax; Last accessed: 17.08.2019





The retrievable data sets are identified by the key meta data listed in the "Identification" section of Table 1. The relevant data period for each data set is also listed. Furthermore, the table presents the available temporal and spatial resolution as well as the number of ensemble members (may change in the future for the operational data). The availability of $\dot{\eta}$ is important for the mode of preparing the vertical velocity fields (see Sect. 3.7) and is therefore marked for accessibility as well. With the current operational data, a temporal resolution of 1 h can be established with a well-selected mix of analysis and forecast fields

(see Sect. 4). The horizontal grid type refers to the way how it fields are archived in MARS. Table 4 provides the relationship between corresponding spectral, Gaussian and latitude / longitude resolutions.

In this paper, we collect the essential changes in forecast steps and spatial resolution since the first IFS release, as they need to be known for using flex_extract. Table 2 lists the evolution of horizontal and vertical resolutions for all operational data sets. The evolution of the forecast steps and the introduction of additional forecast times in "DET-FC" and "ENS-CF" are

summarised in Table 3.

The reanalysis data sets are naturally more homogeneous. Nevertheless, they all have their individual characteristics, making the selection process with flex_extract complex. Table 1 provides an overview of the main differences in the reanalysis meta data. ERA-Interim has a 3-hourly resolution with an analysis and forecast field mix in the full access mode but only a 6-hourly resolution for public users. It lacks the $\dot{\eta}$ fields which makes the retrieval of ERA-Interim computationally demanding

(Sect. 3.7). The ERA5 and CERA-20C reanalyses can be retrieved with 1 h resolution and include ensembles; however, ERA5 ensemble members are not yet retrievable with flex_extract and therefore omitted in the tables. Even though the availability of 1-hourly analysis fields means that forecast fields are not required for most of the variables, accumulated fluxes are only available as forecasts. One should also pay attention to differen forecast start times in both data sets and the complication inplied by forecasts starting from 18 UTC as the date will change until the subsequent start time; see also Sect. 3.6.

With the establishment of the Copernicus Climate Change Service (C3S) in March 2019, a new channel for accessing ECMWF reanalysis data, most prominently ERA5 (Hersbach et al., 2020), has been opened. At the same time, access to this data set via the ECMWF Web API was cancelled. While access directly from ECMWF servers is not affected, in local retrieval modes now one has to submit requests to the Copernicus Climate Data Store (CDS), which uses another Web API called CDS API; in the background, this API retrieves the public data from dedicated web servers for faster and easier access.

Unfortunately, only surface and pressure level data are available in CDS at the moment; this might change in the future.

In the case of member-state users, it is possible to pass the request to the MARS archive even through the CDS interface. Flex_extract is already modified to use this API so a member user can already retrieve ERA5 data. However, experience shows that the performance of this access mode is not good, thus currently it cannot be recommended.

## 3   Software description and methods

The flex_extract software package allows to retrieve and prepare the meteorological input files from ECMWF for FLEXPART (and FLEXTRA) easily and in an automated fashion. The ecessary meteorological parameters to be retrieved are predefined





according to the requirements of FLEXPART and the characteristics of various data sets. The post-processing after retrieval for the calculations of the flux fields (Sect. 3.6) and the vertical velocity (Sect. 3.7) is also incluced.

The actions executed by flex_extract (also called "the software" henceforth) depend on the user access mode (see Sect. 2.1),
the application mode, and the data to be retrieved. There are three possible application modes, using the ECMWF Member State Linux servers, the Member State Gateway server, or a local host. As not all combinations are possible, a total of four different application modes result which are described in Sect. 3.1. Because of the depencies of flex_extract, the respective application environments need to be prepared in different ways as described in Section 3.2. The software comprises a Python part for the overall control of the processing, including the data extraction, a Fortran part for the calculation of the vertical
velocity, korn-shell scripts for batch jobs to run on ECMWF servers, and bash shell scripts as a user-friendly interface to the Python code. Available settings and input files are described in Sect. 3.4. The output files are divided into temporary files (Sect. 3.8) which are usually deleted at the end and the final output files (Sect. 3.9) which serve as FLEXPART input. An overview of the program structure and the work flow together with an example is given in Sect. 3.3.

A general overview of the structure of the flex_extract root directory is provided in Table 5; it is completely different than in
previous versions. The installation script `setup.sh` is directly stored under the root directory together with basic information files. `Source` contains all Python and Fortran source files, each in a separate directory. Flex_extract works with template files, stored in `Templates`. The on-line documentation is included in `Documentation` so that it can also be read off-line. The actual work by users takes place in the `Run` directory. There are the `CONTROL_*` files in the `Control` directory, the `korn` shell job scripts in `Jobscripts` and, in the case of applying the *local mode*, also a `Workspace` directory where
the retrieved GRIB files and final FLEXPART output files will be stored. The `ECMWF_ENV` file is only created for the *remote* and *gateway mode*; it contains the user credentials for ECMWF servers. The `run.sh` and `run_local.sh` scripts are the top-level scripts to start flex_extract. Like in the previous versions, users can also directly call the `submit.py` script. There is also a directory `For_developers` which contains the source files of the online documentation, source files for figures, and sheets for parameter definitions.

**3.1   Application modes**

Arising from the two user groups described in section 2.1 and the three possible locations of application, three different user application modes are defined, namely `Remote`, `Gateway` and `Local mode`. However, the `Local mode` is further split in the `Local member` and the `Local public` mode. A summary of the necessary registration method per mode and user group is outlined in Table 6. An overview of locations and modes is sketched in Figure 1 and a definition is given in the
following list:

**Remote (member)**   Users work directly on ECMWF Linux Member State servers, such as `ecgate` or `cca`. The software will be installed and run in the users `$HOME` directory. Users do not need to install any of the additional library packages mentioned in Section 3.2 since ECMWF provides everything with a module system. Flex_extract takes care of loading the necessary modules.





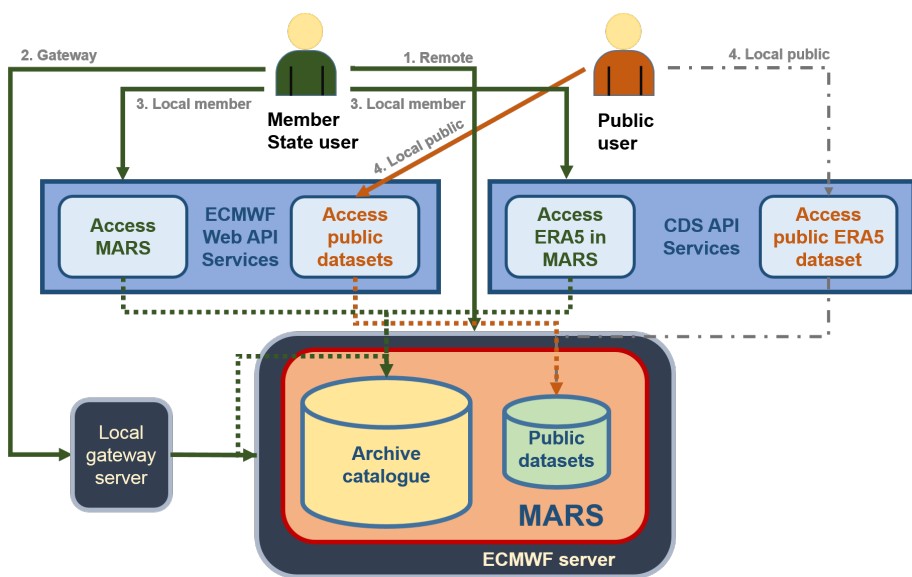

**Figure 1.** Schematic overview of access methods to the ECMWF MARS archive implemented in flex_extract.

**Gateway (member)** This mode is recommended in the case a local Member State Gateway server is in place (ECMWF, 2019j) and the user has a member-state account. Job scripts would then be prepared locally (on the gateway server) and submitted to the ECMWF Linux Member State server via the ECMWF web access toolkit `ECaccess`. The actual data extraction and post-processing is then done at the ECMWF servers and the final data are, if selected, transferred back to the local gateway server. The installation script of flex_extract must be executed at the local gateway server. However, this will install flex_extract in the users `$HOME` directory on the ECMWF server and some extra setup is done in the local gateway version. For instructions about establishing a gateway server, please consult ECMWF (2019j) directly. The necessary software environment has to be established before installing flex_extract.

**Local member** Member-state users work on their local machines which require a similar software environment as the one on ECMWF servers plus the provided Web API's as the interface for the MARS archive.

**Local public** Public users can work on their local machines by not only preparing the general software dependencies but also adding the ECMWF Web API as the interface to the public MARS archive. In this case, a direct registration at the ECMWF is necessary, and all users have to accept a specific license agreement for each data set which is intended to be retrieved.

### 3.2 Software dependencies

The software required to run flex_extract depends on the application mode. Basic requirements for all application modes are listed in Table 7. The *local mode* requires in addition Python packages `ecmwf-api-client` and / or `cdsapi`, depending



on the data set to be retrieved, to connect to the MARS archive as Table 6 shows. Users should make sure that all dependencies are satisfied before starting the installation. The software is tested only in a GNU / Linux environment, although it might be possible to use it also under other operating systems.

## 3.3 Program structure

The work of flex_extract cam be decomposed into the following three separate tasks:

1. Setting the parameters controlling the retrieval and the data set:
   Reading of the `CONTROL` file, command line arguments, and ECMWF user credential file (in the case of *remote* or *gateway* mode). Depending on the application mode, flex_extract prepares a job script which is sent to the ECMWF batch queue, or proceeds with the tasks 2 and 3.

2. Retrieve data from MARS:
   MARS requests are created in an optimised way (jobs split with respect to time and parameters) and submitted. Retrieved data are arranged in separate GRIB files. If the parameter `REQUEST` was set to 1, the request is not submitted and only a file `mars_requests.csv` is created. If it is set to 2, this file is created in addition to retrieving the data.

3. Post-process retrieved data to create final FLEXPART input files:
   After all data are retrieved, flux fields are disaggregated, and vertical velocity fields are calculated by the Fortran program `calc_etadot`. Finally, the GRIB fields are merged into a single GRIB file per time step with all the fields FLEXPART expects. Depending on the parameter settings, file transfers are initiated and temporary files deleted.

In task 1, the software differentiates depending on the application mode. In the case of *remote* or *gateway* mode (see also Fig. 2), the job script for the ECMWF batch system is prepared and submitted to the batch queue. The program finishes with a message to standard output. In the case of the *local* application mode, the work continues locally with tasks 2 and 3, as illustrated in Figure 3.

Each application mode has its unique process steps and its own connection to the MARS archive. Figure 4 demonstrates the involved input files, execution scripts and connection methods as well as the locations where each step takes place. In the *gateway mode*, the setup task will be done on the gateway server and the created job script is sent via the `ECaccess` command to the batch queue before flex_extract terminates. As soon as the job script is processed, the job environment is created and flex_extract is started in *local mode*, reading the prepared `CONTROL` file, extraction of data, and post-processing tasks. In this mode, the extraction is done with a MARS command. If it was selected, the final output files are sent to the local member-state gateway server. The *remote mode* works completely on ECMWF servers but has the same process sequence as the *gateway mode*. In the *local mode*, all work is done on the local host, except the data extraction which is done by an HTTP request via the Web APIs. The data are sent back to the local host instantly.



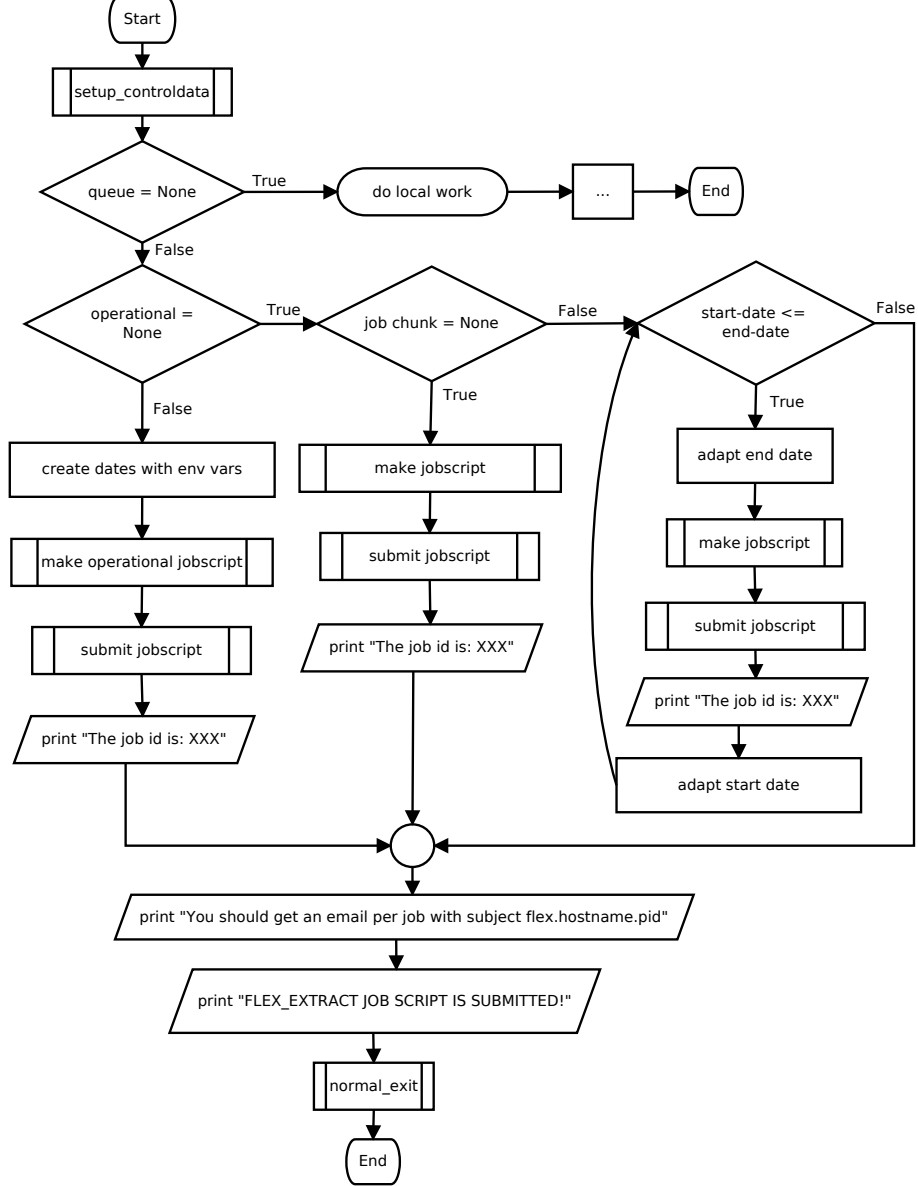

**Figure 2.** Flow diagram for the *remote* and *gateway* mode. The job script is created and submitted to the batch queue on the ECMWF server. The job script will then start on the ECMWF server and apply flex_extract in *local* mode. In the case of `queue = None`, the local mode is selected and Figure 3 applies as is indicated by the branch *do local work*. Trapezoidal boxes mark standard output, simple rectangles mark the execution of sequential instructions, and the rectangles with a side border mark the execution of subroutines. The boxes in diamond form indicate decisions.



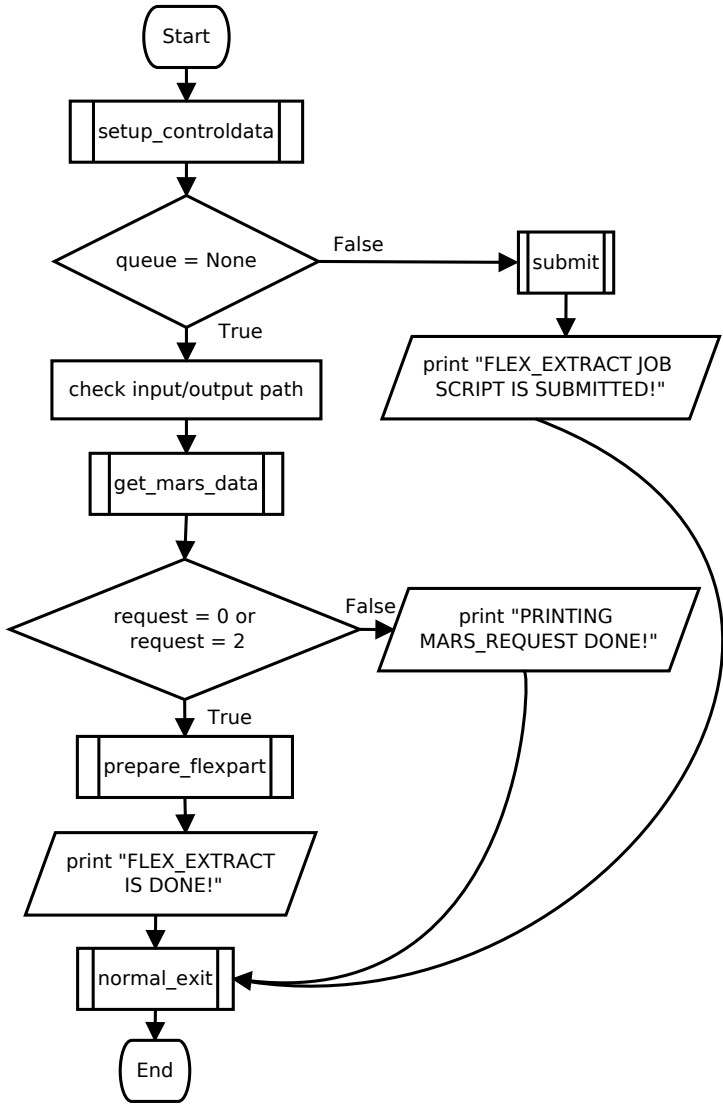

**Figure 3.** Flow diagram for the *local* application mode. If `queue` ≠ `None`, flex_extract assumes that a job script has to be sent to an ECMWF server and selects another branch shown in Fig. 2. This is marked by the *submit* block. In the case of `request == 1`, flex_extract is forced to skip the retrieval and post-processing steps and just writes the `mars_request` file. Within the pure local mode the retrieval and post-processing parts are conducted. Symbols as in Fig. 2





### 3.4 Input files

#### 3.4.1 The `CONTROL` file

Flex_extract needs a number of controlling parameters. They are initialised by flex_extract with their default values and will be
overwritten by the settings in the `CONTROL` file. It is necessary to understand these parameters and to set them to proper and
consistent values. They are listed in Tables 8 and 9 with their default values and a short description. More detailed information,
hints about the conditions of settings, and possible value ranges are available in the supplemental material and partially in
Section 4. The files are read during task 1. Only those parameters which which deviate from the default values have to be pro-
vided. The file `CONTROL.documentation` provides a collection of the available parameters in grouped sections together
with their default values. Users can start from this file to define their setup or use one of the sample application `CONTROL`
files as a template (in `flex_extract_v7.1/Run/Control/`). These samples correspond to the data sets described in
Section 2.2. One example for each data set is provided with some variations in resolution, type of field, or method for the cal-
culation of the vertical verlocity. The naming convention is `CONTROL_<dataset>[.optionalIndications]`, where
the `optionalIndications` is an optional string to provide further characteristics about the retrieval method or the data
set. For the operational data sets (OD) this string contains information of the stream, the field type of forecasts, the method for
extracting the vertical velocity and other aspects such as time or horizontal resolution.

Regarding the file content, the first string in each line is the parameter name, the following string(s) (separated by spaces) are
the parameter values. The parameters may appear in any order, with one parameter per line. Comments can be added as lines
beginning with #-sign, or after the parameter value. Some of these parameters can be overruled by command line parameters
provided at program call. In earlier versions, each parameter name contained the leading string `M_`; this was removed for
version 7.1 but is still possible for compatibility. The grid resolution had to be provided in 1/1000 of a degree before, while
now it can be also provided as a decimal number with unit degree. Flex_extract is able to check for correct setting of the `GRID`
parameter with the domain-specific settings.

It is now also possible to reduce the number of data values for the combination of `TYPE`, `TIME` and `STEP` parameter
combination to the actual temporal resolution. Previous versions expected to have 24 values per parameter, one for each hour
of the day, even though a 3-hourly temporal resolution was selected as shown in the following example:

```
DTIME 3
TYPE AN AN AN AN ... AN AN AN AN
TIME 00 01 02 03 ... 20 21 22 23
STEP 00 00 00 00 ... 00 00 00 00
```

The more intuitive solution of providing just the data for the time steps to be retrieved leads, for example, to eight data values
per parameter for a 3-hourly retrieval as shown in the next example:

```
DTIME 3
TYPE AN AN AN AN AN AN AN AN
```





```
TIME 00 03 06 09 12 15 18 21
     STEP 00 00 00 00 00 00 00 00
```

or four values for a 6-hourly retrieval.

```
     DTIME 6
     TYPE AN AN AN AN
TIME 00 06 12 18
     STEP 00 00 00 00
```

The only necessity is the consistent setting of the `DTIME` parameter to define the temporal resolution. For backward compatibility, this means that `DTIME` can be coarser than the number of temporal points provided in `TYPE`, `TIME` and `STEP`, but not finer.

### 305  3.4.2  User credential file `ECMWF_ENV`

In the *remote* and *gateway* mode, the software sends job scripts to the batch system of an ECMWF server, thus it is necessary to provide the user and group name which are given in file `ECMWF_ENV`. Additionally, this file provides the name of the local member-state gateway server and the destination so that unattended file transfer[4] (`ectrans`) between ECMWF and member gateway servers can be used. The destination is the name of the so-called `ectrans` association; it has to exist on the local

gateway server.

### 3.4.3  Template files

Some files are highly variable depending on the setting in the other input files. They are created during run time by using template files. The templates are listed in Table 11. Flex_extract uses the Python package `genshi` to read the templates and substitute placeholder by values. These placeholders are marked by a leading $ sign. In the case of the korn shell job scripts,

where (environment) variables are used, the $ sign needs to be escaped by an additional $ sign. Usually, users do not have to change these files.

### 3.5  Executable scripts

### 3.5.1  Installation

The installation of flex_extract is done by the shell script `setup.sh`, which is located in the root directory of flex_extract.

This script sets all available command line arguments for the installation, does some plausibility checks and finally calls the Python script `install.py`. Users are supposed to provide these arguments according to their environment and application mode. Parameters which must be set for all application modes are defined in Table 12. The Python script does all necessary operations depending on the selected application mode. In the case of *remote* and *gateway* mode, the `ECMWF_ENV` file is created

---

[4]https://confluence.ecmwf.int/display/ECAC/Unattended+file+transfer+-+ectrans; Last accessed: 09.09.2019





(settings of related parameters to be done in `setup.sh`, see Table 10), the job script template is prepared, and the korn shell
script for compiling the Fortran source code `compilejob.ksh` is created. After these preparations, a tar ball with the core
content is created and copied to the target location (ECMWF server or local installation path). Next, the `compilejob.ksh` is
submitted to the batch system of ECMWF servers via `ECaccess` commands, or just untar-ed at the target location. It compiles
the Fortran code and, in the case of *remote / gateway* mode, a log file is sent to the user's email address.

### 3.5.2 Execution

In earlier versions of flex_extract, an extraction was initiated by calling the Python script `submit.py` with suitable command
line arguments. The Python script constitutes the main entry point and controls the program flow including the call of the
Fortran program. Now, it is still possible to work in that way, but a wrapper shell script `run.sh` is provided in addition. This
shell script contains a user section where the Python command line arguments are to be set. This mode of operation is simpler
for beginners, and it is useful for repetitive tasks, as no arguments can be forgotten. The available command line arguments are
listed in Table 13. Some of the arguments occur only with the program call, while others are also defined in the `CONTROL` file.
In this case, the values in `submit.sh` take precedence over those from the `CONTROL` file.

The `submit.py` script interprets the command line arguments (overview available through `./submit.py --help`
from the Python source directory) and, based on the input parameter `QUEUE`, it decides which application mode is active.
In *local mode*, data are fully extracted and post-processed, while in the *remote* and *gateway mode*, a korn shell script called
`job.ksh` is created from the template and submitted to the ECMWF batch system via the gateway server.

The underlying template is `job.temp`, stored in the `Templates` directory. This template was generated in the installation
process from `job.template` where some basic settings were done, which are the same for each flex_extract execution.
The job script sets necessary directives for the batch system, creates the run directory and the `CONTROL` file, sets some
environment variables (such as the `CONTROL` file name) and executes flex_extract. The standard output is collected in a log
file which will be sent to the users' email address in the end. The batch system settings are fixed and they differentiate between
the `ecgate` and the `cca/ccb` server systems to load the necessary modules for the environment when submitted to the batch
queue. The `ecgate` server has directives marked with `SBATCH`[5] for the SLURM workload manager, the high performance
computers `cca` and `ccb` have `PBS`[6] comments for PBSpro. The software environment dependencies mentioned in Section 3.2
are fulfilled by loading the corresponding modules. It should not be changed without further testing.

Just for completeness, there are two more entry points in the software which are provided for debugging. The Python scripts
`getMARSdata.py` and `prepare_flexpart.py` are normally used as modules, but can also be used as an executable pro-
gram. The `getMARSdata.py` script controls the complete extraction of ECMWF data, while the `prepare_flexpart.py`
controls the complete post-processing.

---

[5]https://confluence.ecmwf.int/display/UDOC/Writing+SLURM+jobs; Last accessed: 10.09.2019
[6]https://confluence.ecmwf.int/display/UDOC/Batch+environment%3A++PBS; Last accessed: 10.09.2019





### 3.6 Disaggregation of aggregated flux data

FLEXPART interpolates meteorological input data linearly to the position of computational particles in time and space Stohl et al. (1998); Stohl et al. (2005). This method requires point values in the discrete input fields. However, flux data (as listed in Table 14) from ECMWF represent cell averages or integrals and are accumulated over a time interval, which depends on the data set. Hence, to conserve the integral quantity with FLEXPART's linear interpolation, a pre-processing scheme has to be applied.

The first step is to de-accumulate the fields in time so that each value represents an integral in $(x, y, t)$-space. Afterwards, a disaggregation scheme is applied. While the horizontal cell values are simply ascribed to the cell centre, with respect to time, a more complex procedure is needed because the final values should correspond to the same time as the other variables. In order to be able to carry out the disaggregation procedure of Paul James, additional flux data are retrieved automatically for one day before and one day after the period specified. Note that these additional data are temporary and used only for disaggregation

within flex_extract. They are not contained in the final FLEXPART input files. The flux disaggregation produces files named `fluxYYYYMMDDHH`, where `YYYYMMDDHH` is the date. Note that the first and last two flux files do not contain any data. Note that for operational retrievals which use the `BASETIME` parameter, forecast fluxes are only available until `BASETIME`, so that interpolation is not possible in the last two time intervals. This is the reason why setting `BASETIME` is not recommended for regular on-demand retrievals.

### 3.6.1 Disaggregation of precipitation in older versions


In versions 7.0.x and before, a relatively simple method was applied to process the precipitation fields, consistent with the linear temporal interpolation applied in FLEXPART for all variables. At first, the accumulated values are divided by the number of hours (i.e., 3 or 6). For the disaggregation, precipitation sums of four adjacent time intervals $(p_a, p_b, p_c, p_d)$ are used to generate the new instantaneous precipitation (disaggregated) value $p$ which is valid at the boundary between time intervals 1 and 2 as

follows:

$$p_{ac} = \begin{cases} 0.5 \ p_b & \text{for } p_a + p_c = 0 \\ \frac{p_b \ p_c}{p_a + p_c} & \text{for } p_a + p_c > 0 \end{cases} \tag{1}$$

$$p_{bd} = \begin{cases} 0.5 \ p_c & \text{for } p_b + p_d = 0 \\ \frac{p_b \ p_c}{p_b + p_d} & \text{for } p_b + p_d > 0 \end{cases} \tag{2}$$

$$p = p_{ac} + p_{bd} \tag{3}$$

The values $p_{ac}$ and $p_{bd}$ are temporary variables. The new precipitation values $p$ constitute the deaccumulated time series used

later in the linear interpolation scheme of FLEXPART. If one of the four original time intervals has a negative value, it is set to 0 prior to the calculation. Unfortunately, this algorithm does not conserve the precipitation within the interval under consideration, negatively impacting FLEXPART results as discussed by Hittmeir et al. (2018) and illustrated in Figure 5.





Horizontally, precipitation is given as cell averages. The cell midpoints coincide with the grid points at which other variables are given, which is an important difference to the temporal dimension. FLEXPART uses bilinear interpolation horizontally.

### 3.6.2 Disaggregation for precipitation in version 7.1

Due to the shortcomings described above, a new algorithm was developed by Hittmeir et al. (2018). In order to achieve the desired properties (Hittmeir et al., 2018, p. 2513), a linear formulation with two additional supporting points within each interval is used. The current version of flex_extract implements this algorithm for the temporal dimension. Figure 6 shows how these requirements are fulfilled in the new algorithm for the simple case presented in Figure 5.

Flex_extract allows to choose between the old and the new disaggregation method for precipitation. In the latter case, the two additional sub-grid points are added in the output files. They are identified by the parameter "step" which is 0 for the original time at the left boundary of the interval, and, respectively, 1 or 2 for the two new sub-grid points. Filenames do not change.

### 3.6.3 Disaggregation for the other flux fields

The accumulated values for the other variables are first divided by the number of hours and then interpolated to the exact times using a bicubic interpolation which conserves the integrals of the fluxes within each timespan. Disaggregation uses integrated values $F$ during four adjacent time-spans ($F_0, F_1, F_2, F_3$) to generate a new, disaggregated point value $F$ valid at the boundary between intervals 1 and 2 as follows:

$$F_a = \frac{F_3 - F_0 + 3\,(F_1 - F_2)}{6} \tag{4}$$

$$F_b = \frac{F_2 + F_0}{2} - F_1 - 9\,\frac{F_a}{2} \tag{5}$$

$$F_c = F_1 - F_0 - 7\,\frac{F_a}{2} - 2\,F_b \tag{6}$$

$$F_d = F_0 - \frac{F_a}{4} - \frac{F_b}{3} - \frac{F_c}{2} \tag{7}$$

$$F = 8\,F_a + 4\,F_b + 2\,F_c + F_d. \tag{8}$$

## 3.7 Preparation of vertical velocity

An accurate representation of the vertical velocity is a key component for atmospheric transport models. One of the considerations for the design of FLEXTRA was to work entirely in the native coordinate system of ECMWF's IFS model to minimise interpolation errors. This meant that the same hybrid $\eta$ coordinate (terrain-following near ground, approaching pressure levels towards the model top) would be used, which implied to use the corresponding native vertical velocity ("*etadot*")

$$\dot{\eta} = \frac{\mathrm{d}\eta}{\mathrm{d}t} \tag{9}$$

rather than the more commonly used ordinary vertical velocity in a simple $z$-sytem (units of $\mathrm{m\,s^{-1}}$) or the vertical motion $\omega$ of pressure-based systems (unit $\mathrm{Pa\,s^{-1}}$). For reasons that we can't reconstruct, however, FLEXTRA did not use $\dot{\eta}$ strictly, but





rather a quantity

$$\dot{\eta}_p = \frac{\mathrm{d}\eta}{\mathrm{d}t}\frac{\partial p}{\partial \eta} \tag{10}$$

which obviously has units of $\mathrm{Pa\,s}^{-1}$. The code calls this quanitity *etapoint*, not to be confused with *etadot*. Even though in FLEXPART this concept had to be abandoned in favour of a terrain-following $z$-system to allow a correct implementation of

the Langevin equation for turbulent motion, FLEXTRA and FLEXPART share the same requirement for the vertical motion with respect to their input. Over many years, ECMWF would store only the post-processed perssure vertical velocity $\omega=\mathrm{d}p/\mathrm{d}t$. Transforming this back to $\dot{\eta}$, with approximations and interpolations involved in both operations, leads to vertical velocities that do not fulfill continuity. Therefore, $\dot{\eta}$ was reconstructed from the fields of divergence using the continuity equation, integrated from the model top downward as described in Simmons and Burridge (1981). In the IFS model, dynamical variables are

horizontally discretised by spherical harmonics. It is best to do this on the reduced Gaussian grid that is used in IFS when a grid-point representation is required.

   In September 2008, ECMWF started to archive the model's native vertical velocity fields ($\dot{\eta}$) for the operational analyses and forecasts. This allowed flex_extract to skip the cumbersome reconstruction and directly use this parameter. The amount of data that needs to be extracted from MARS, the CPU time and the memory requirements are all reduced substantially. The

ERA5 and CERA-20C reanalyses also provide $\dot{\eta}$. Thus, even though it is possible to use the old method on new data sets, there is no reason to do so and it would be a waste of resources. It is, however, still kept in flex_extract to allow extraction of data from the older data sets, in particular ERA-Interim. In the following, the two methods are briefly characterised.

### 3.7.1   Reconstruction of the vertical velocity using the continuity equation

The most accurate algorithm for the reconstruction of the native vertical velocity requires the extraction of the horizontal

divergence fields and the logarithm of the surface pressure in spectral representation (and thus always global, regardless of the final domain), their transformation to the reduced Gaussian grid (introduced by Ritchie et al. (1995)), on which the continuity equation is solved, a transformation back to the spectral space, and finally the evaluation on the latitude-longitude grid desired by users. Especially for high spectral resolution, this is a compute- and memory-intensive process that also takes time, even when making use of OpenMP parallelisation. Larger data sets can only be treated on the supercomputer (`cca`) but not on

`ecgate`. The code for these calculations is written in Fortran90.

   Alternatively, data can be extracted from MARS immediately on the latitude-longitude grid for the domain desired, and the continuity equation is then solved on this grid, but this method is not as accurate as the calculations on the Gaussian grid, particularly for higher spatial resolutions.

### 3.7.2   Preparation of the vertical velocity using archived $\dot{\eta}$

If the vertical velocity is available in MARS, it only needs to be multiplied with $\partial p/\partial \eta$. In the flex_extract version discussed here, this is done by the Fortran program whose functionality is described below.





### 3.7.3  Short description of the functionality of the `calc_etadot` code

A dedicated working directory is used where all input and output files are kept. Currently, the files have names of the form `fort.xx` where `xx` is some number.

The control file steering the code is `fort.4` and has the form of a Fortran namelist. An overview of the options set by this namelist is contained in Table 15. The control file is prepared automatically by the Python code, but some of these parameters appear also as input to the Python part. Note that the selection of the method for obtaining $\dot{\eta}$ follows the logic laid out in Table 16.

All other input files are data in GRIB format that were retrieved from MARS. The code is using dynamic memory allocation

and thus does not need to be recompiled for different data sets.

The code is provided with a set of makefiles. The standard version assumes a typical GNU/Linux environment with a `gfortran` compiler and the required libraries: *openmp* for parallelisation – it comes with the `gcc` compiler package, `libgrib_api` or *libeccodes* for handling GRIB files, *libemos* for transformation between the various representations of fields. Note that the latter two typically require also so-called developer packages containing the Fortran module files. It is

assumed that these libraries have been installed as a package from the distribution and thus are at their standard locations and compatible with the `gfortran` compiler. There is one makefile called `makefile_fast` with optimisation that is used for production. In addition, there is `makefile_debug` which is optimised for debugging. There are also makefiles for `cca` and `ecgate` as well as a makefile for the Intel Fortran compiler to be used locally. The latter one may require adaptations by users with respect to the library and include paths.

If the program finishes successfully, the last line written to `standard output` is SUCCESSFULLY FINISHED calc_etadot: CONGRATULATIONS which is useful for automated checking the success of the run. The output file into which the fields of $\dot{\eta}_p$ and the other three-dimensional variables (temperature, specific humidity, $u$ and $v$ components of the wind – not the recently introduced cloud water variable) are combined is `fort.15`; it is a GRIB file.

The code also foresees options for certain checks where different forms of the vertical velocity are obtained, statistically

compared, and also written out (see Table 15). These options were used for quality control in the development process and should not normally be activated by users.

Currently, the code also unifies the three-dimensionsal fields extracted from MARS and stored in separate GRIB files with the calculated vertical velocity by writing out all fields into a single GRIB file; later this is unified with the 2D fields and the new 3D parameters such as cloud water and written out into a final single GRIB file as required by FLEXTRA and FLEXPART.

## 3.8  Temporary output files

These temporary output files are usually deleted after a successful data extraction. They are only kept in debugging mode, which is the case if the `DEBUG` parameter is set to true.





### 3.8.1 MARS GRIB files

All extracted meteorological fields from MARS are in GRIB format and stored in files ending with `.grb`. MARS requests
are split in an optimised way to reduce idle times and considering the limit of data transfer per request. The output from each
request is stored in one GRIB file whose name is defined as `<field_type><grid_type><temporal_property>`
`<level_type>.<date>.<ppid>.<pid>.grb`. The field type can be analysis (AN), forecast (FC), 4d variational anal-
ysis (4V), validation forecast (CV), control forecast (CF) and perturbed forecast (PF). The grid type can be spherical harmonics
(SH), Gaussian grid (GG), output grid (OG) (typically lat/lon) or orography (_OROLSM) while the temporal property distin-
guishes between an instantaneous field (__) or an accumulated field (_acc). Level types can be model (ML) or surface level
(SL) and the date is specified in the format `YYYYMMDDHH`. The last two placeholders are the process number of the parent
process of submitted script (ppid) and the process number of the submitted script (pid). The process IDs are incorporated to
avoid mixing of fields if several flex_extract jobs are performed in parallel (which is, however, not recommended).

### 3.8.2 MARS request file

This file contains a list of the MARS requests from one flex_extract run, with one request per line. This is an optional file users
are able to create in addition to full extraction; it can also be created without actually extracting the data which is useful for
test purposes. Each request consist of the following parameters whose meaning is explained in Tables 8 and 9, and in more
detail in the supplemental material or are self-explanatory: request number, accuracy, area, dataset, date, expver, Gaussian,
grid, levelist, levtype, marsclass (alias class), number, param, repres, resol, step, stream, target, time and type. The parameters
Gaussian (defines whether the field is regular or a reduced Gaussian grid), levtype (distinguishes between model levels and
surface level) and repres (defines the grid type – SH, GG, OG) are internal parameters not defined as any available input
parameter.

### 3.8.3 Vertical discretization constants

The file `VERTICAL.EC` is created by the Fortran program as a temporary storage for internal usage and contains the A and B
constants to calculate the model level heights in m.

### 3.8.4 Index file

The index file is called `date_time_stepRange.idx`. It contains indices pointing to specific GRIB messages from one or
more GRIB files so Python can easily loop over these messages. The messages are selected with a predefined composition of
GRIB keywords.

### 500 3.8.5 Files with forecast vertical flux data

The flux files, in the format `flux<date>[.N<xxx>][.<xxx>]`, contain the de-accumulated and disaggregated flux fields
which are listed in Table 14. The files are created per time step with the date being in the format `YYYYMMDDHH`. The optional



block `[.N<xxx>]` marks the ensemble forecast, where `<xxx>` is the ensemble member number. The second optional block `[.<xxx>]` marks a pure forecast with `<xxx>` being the forecast step.

Note that, in the case of the new dis-aggregation method for precipitation, two new sub-intervals are added in between each original time interval. They are identified by the forecast step parameter `STEP` which is 0 for the original time interval and 1 or 2 for the two new intervals respectively.

### 3.8.6 `fort.*` files

There are a number of input files for the `calc_etadot` Fortran program named `fort.xx`, where `xx` is the number which defines the meteorological fields stored in these files. They are generated by the Python part of flex_extract by just splitting the meteorological fields for a unique time step from the `*.grb` files. Table 17 explains the numbers and the corresponding content. Some of the fields are optional and are retrieved only with specific settings, for example the divergence is retrieved only if $\dot{\eta}$ is not available in MARS, and the total cloud water content is an optional field for FLEXPART v10 and newer. The output of `calc_etadot` file `fort.15`.

### 3.9 Final output – FLEXPART input files

The final output files are the FLEXPART input files containing the meteorological information. FLEXPART expects one file with all relevant fields per time step. Table 18 and 19 list all of the meteorological fields that flex_extract retrieves and FLEXPART expects. The naming of these files depends on the extracted data. In the following sections we describe the differences and how the filenames are built.

### 3.9.1 Standard output files

The standard file names have the format `<prefix>YYMMDDHH`, where the `<prefix>` is by default defined as `EN` and can be re-defined in the `CONTROL` file. Each file contains all meteorological fields on all selected model levels on a latitude-longitude grid needed by FLEXPART for the specified time step in the filename. If not otherwise stated, model level fields are in GRIB2 format and surface levels in GRIB1. Forecast times and steps are summed up to the corresponding analysis hour.

### 3.9.2 Output files for pure forecasts

For the selection of forecasts longer than 23 h, a different naming scheme has to be applied to avoid collisions of time steps for forecasts of more than one day. This case is defined as pure forecast mode and file names are defined as `<prefix>YYMMDD.HH.<FORECAST_STEP>`. The `<prefix>` is, as in the standard output files, `EN` by default and can be re-defined in the `CONTROL` file. In this case, the date format `YYMMDD` does not include the hour directly and the hour represents the starting forecast time and is placed separately by `.HH`. The `FORECAST_STEP` is a 3-digit number which represents the forecast step in hours.





### 3.9.3   Output files for ensemble predictions

If flex_extract retrieves ensembles, we obtain multiple fields per meteorological variable (the ensemble members) for a single time step which are distinguished by the GRIB parameter `NUMBER`. For each ensemble member, all fields are collected together in one file. The standard filenames are supplemented by the letter `N` for "number" and the ensemble member number in a 3
digit format such as `<prefix>YYMMDDHH.N<ENSEMBLE_MEMBER>`.

### 3.9.4   Additional fields with new precipitation dis-aggregation

The new dis-aggregation method for precipitation fields produces two additional fields for each time step and precipitation type. They contain the sub-grid points in the correspoinding original time intervals as described above in Sect. 3.6.2. The two additional fields are marked with the step parameter in the GRIB messages, are set to "1" and "2", respectively. The output
filenames do not change in this case.

## 4   Considerations for application

First of all, users should be aware of the different natures of operational and reanalysis data sets (see Table 1). Operational data are available since the start of ECMWF's operational forecasts, and are influencend by frequent changes in IFS model physics and resolution. Reanalysis data sets were created using a single IFS model version throughout the whole period covered. More
precisely, the CERA-20C data set (with 91 vertical levels, 1.25° horizontal and 3 h temporal resolution) has a lower resolution but covers a very long period (from 1901 to 2010) and will thus be suitable for certain climate applications. ERA-Interim data set (with 60 vertical levels, a medium resolution of 0.75° horizontally and 3 h temporally) was the standard ECMWF reanalysis until recently, but has no $\dot{\eta}$ stored in the MARS archive which make retrievals computationally demanding. The new ERA5 data set has the highest resolution (0.25° horizontally and 1 h temporally, 137 vertical model levels). Users are encouraged to use
ERA5 data rather than the ERA-Interim data set (production ended in August 2019). In addition to its better resolution, ERA5 covers a longer period than ERA-Interim, provides uncertainty estimates with a 10-member ensemble of data assimilation and uses a newer IFS model version (ECMWF, 2019l).

   With respect to the relation between temporal and spatial resolution, it is important to consider the use in FLEXPART and their influence on numerical errors. It is not useful to apply high horizontal resolution in combination with, for example, 6-
hourly temporal resolution as in such a case, small fast-moving structures are resolved in space, but their movement will not be properly represented. Interpolation will not let the structures move, but rather jump from their position at time $t$ to that at time $t + 6$ h if the displacement between two subsequent times where fields are available is comparable to or larger than their characteristic width normal to the phase speed. Users can orient themselves with the spatial and temporal resolutions at which ECMWF provides reanalysis data, and the sample `CONTROL` files.





On the other hand, one has to keep in mind the requirements of the FLEXPART application. For a climatological study on global scales, a horizontal resolution of 1° could be a reasonable choice, whereas tracking point releases in complex terrain would call for the best available resolution.

Attention should also be paid to the model topography and the land-sea mask. Due to limited resolution, a coastal site with a given geographical coordinate could be over water in the model. Then it might be better to shift the coordinates of a release

or receptor point in FLEXPART slightly. Another aspect is that the smoothed representation of the topography could mean that the model topography is above or below the real height of a site. It is therefore important to select the proper kind of $z$ coordinate in the FLEXPART RELEASES file. As a compromise, one can place a release location at a hight between real and model topography (for mountain sites which are usually lower in the model topography than in reality). In such cases, it is strongly recommended to retrieve the model topography and land-sea mask and investigate them carefully before deciding on

the FLEXPART set-up details, or even before retrieving the full meteorological data set, as one might come to the conclusion that one with better resolution should be used.

The vertical levels used in FLEXPART follow a hybrid $\eta$ coordinate system. This is much more efficient than pure pressure levels since hybrid $\eta$ coordinates are terrain following near the ground and approach pressure levels close to the model top. This has the advantage of better resolution of strong gradients in the boundary layer irrespective of the terrain height, and allows

to easily fulfill the lower boundary condition of a flow parallel to the surface whereas pressure levels do not follow the terrain (Stohl et al., 2001). ECMWF data sets either directly provide the $\dot{\eta}$ variable or the data needed to reconstruct it accurately. This is a big advantage of ECMWF data compared to other data sources, most notably the NCEP model data, which are available only on pressure levels.

Attention should be paid to the number of vertical model levels to be extracted and used in FLEXPART, as the computa-

tional cost of the FLEXPART `verttransform` subroutine (reading and preparing meteorological input) increases with the third power of the number of vertical levels. Thus, only data that are really needed for the application (e.g., troposphere, or troposphere and lower stratosphere) should extracted.

Operational data sets and ERA-Interim have analysis fields on a 6- or 12-hourly basis (0, (6), 12 and (18) UTC) only. The gaps inbetween can be filled with forecast fields. Mixing analysis and forecast fields should be done by considering at which

time steps the differences between two IFS run segments will be the lowest. For example, using all four analysis fields, but the forecasts starting from 00 and 12 UTC only would lead to unnecessary rate of changes between 05, 06, and 07 UTC and 17, 18, and 19 UTC. This should be avoided by using only 00 and 12 UTC analysis fields and the forecast fields for +1 to +11 hours for the forecasts starting at times 00 and 12 UTC, respectively.

## 5  Quality assurance

Nowadays software development is mainly dominated by adding new features as well as maintaining and adjusting specifications rather than developing from scratch (Beizer, 1990). To assure a certain quality of the software, the testing part is at least as important as developing the software itself. Adding new functionalities requires to develop new tests to find bugs or show





that the software works under specified conditions. As a consequence, tests from the previous software version can be used to show that there are no undesired changes in the unchanged part of the software. This is called regression testing (Beizer, 1990; Spillner, 2012). Also, tests need constantly be updated to follow up the changes. For this flex_extract version, a huge part was about code refactoring which necessitated the development of a number of regression tests. First of all, a first set of *unit tests* (Sect. 5.1), which are a kind of regression test, have been developed within the refactoring process as they are the established best practice in software engineering to investigate small code blocks. Furthermore, we defined test cases to compare the outcome of two software versions after three different stages of the software, first the prepared MARS requests (Sect. 5.2), second the obtained vertical velocity for the different possibilities (Sect. 5.4) and third the final output files in GRIB format (Sect. 5.3). In addition, generic tests were performed by applying the software with predefined CONTROL files (Sect. 5.5) which are distributed with the software to serve as examples for the typical applications. Finally, on top of these tests, some code metrics were determined to track the quality of the code. Combining all of these tests establishes a sustainable testing environment to improve the future development process. They are not important for the normal user of flex_extract.

## 5.1 Unit tests

Unit tests are used to test the smallest pieces of code (single code blocks) independently to identify a lack of functional specification (Beizer, 1990). Applying unit tests does not guarantee error-free software rather than limiting the chance of occurrence. Once the tests were written they are also a kind of documentation and serve as a protection to not altering a functional behavior after applying code changes (Wolff, 2014). Hence, they are also kind of a regression test.

As a first step, we launched unit tests for functions which were designed or partly refactored to be testable code blocks. Since unit tests are for the verification of small and independent code blocks, functions which are too complex or too long are badly testable most of the times. In the future, our intention is to increase the number of unit tests and refactor the still too complex functions into smaller ones (see also Sect. 5.6 for identifying complex functions).

We used the `pytest` package which is a part of standard Python as well as the `mock` package which simulates external dependencies or results for the tests solely. This gives the opportunity to test the good and bad pathes in a function and usually a function holds as many unit tests as there are different branches. It is a matter of defining all possible results depending on the input states and verify the expected results. The first set of unit tests were applied for functions from the `install` and `tools` modules as well as for the `UIOFiles` and `EcFlexpart` class. The details for each test are not described here; their functionality is obvious from the code.

## 5.2 Regression testing for MARS requests

The parameters in the MARS requests produced by flex_extract are a key component of the extraction process. Flex_extract v7.1 contains a test to compare the content of MARS requests as produced by two versions. It checks whether the number of columns (parameters) in the request files (see Sect. 3.8.2) is unchanged, whether the number of requests is equal, and whether the content of the request is identical (excuept for the desired differences and the environment-dependent data such as paths).





The MARS request files for the current version in use are generated automatically at runtime without actually retrieving the data while the files for the reference version have to be in place already. Since the MARS request files are grouped by version and are kept saved, the number of reference data sets will grow with each new version.

The release comes with a predefined set of `CONTROL` files explicitly for this test as well as with a set of MARS request reference files from the previous version 7.0.4. The test can compare any number of MARS request files emerging from a set of
`CONTROL` files. However, one has to make sure that the reference version contains the request files from the same `CONTROL` files. Results are saved in log files. Instructions on how the test can be conducted are given in a `README.md` file.

The comparison between version 7.0.4 and 7.1 only showed expected differences related to a bug fix in the determination of the time period.

## 5.3 Regression testing for GRIB files

The final product of flex_extract, the FLEXPART input files in GRIB format (see Sect. 3.9), should be identical between the previous and the current version, apart from the new or modified features features. Since there is always a possibility to have tiny (insignificant) deviations in the actual field values when retrieving at different points in time (changing environment, library versions, interpolation uncertainties, etc.), the focus of this test lies on the files themselves and the GRIB message headers which should not be different. Future improvments may also test for value differences considering a significance threshold.

A regression test was created which compares the GRIB files produced by two versions with respect to the number of files produced, the file names, the number of GRIB messages per file, the content of the GRIB messages header, and statistical parameters for the data themselves. If differences are reported, the developer has to judge whether they are expected or an indication of problems. The current release version 7.1 includes a minimal set of reference data from version 7.0.4, one for each type of data set (see Sect. 2.2). There will be more test data in the future which can then be downloaded from the
community website to limit the size of the distributed release tarball. The corresponding reference `CONTROL` files are also distributed with the tarball to enable the retrieval of the data with the new version. This has to be done manually followed by placing the resulting GRIB files in a specific path as described in the *README.md* file.

## 5.4 Functionality and performance tests for the Fortran code

Regression tests were set up to reflect the three possibilities for obtaining the vertical velocity $\dot{\eta}$ listed in Table 16. In addition
to a basic test for each, enriched tests are implemented where all checks and additional outputs are activated (names with appended `all`). These tests use a pre-specified small domain ($10° \times 10°$, 11 levels) and low spectral resolution (T159) and thus run quite fast. As high spectral resolution and a large domain may pose specific problems, and as it will be relevant to watch the run times, additional high-resolution tests have been created for the `gauss` and `etadot` cases with a domain covering the northern hemisphere and all 137 vertical levels. The `gausshigh` test uses a grid spacing of 0.25° and the corresponding
spectral resolution of T799; the `etadot` case uses 0.1° and T1279.

The code package contains a set of reference outputs, and scripts to create the reference output and to run the actual regression tests. It checks for bitwise identity of the output files (data files and `standard output` written to a log file). A quantitative





comparison of the resulting $\dot{\eta}_p$ which would be useful for modifications that affect the results is not yet implemented. The scripts run each test with both the fast and the debug version of the executable. The script for creating the reference also

ensures that both yield identical results. In addition, the runtimes are saved to a `csv` file.

### 5.5 Generic test using predefined `CONTROL` files

Flex_extract comes with a set of `CONTROL` files listed in Table 20; executing it with each of them constitutes a generic test ensuring that the data extraction works without problems for all typical applications. This has been verified for version 7.1 by manually executing the software with all these files, and inspecting the results produced. Note that public users can only use

files ending with `.public`; they were tested in local public mode. All other cases were tested both in local and in gateway mode. Since the remote mode does not differ much from the gateway mode, only a subset of files were also tested in this mode. Results were evaluated by inspecting the log files for "success" messages and, where possible, with the regression test for GRIB file comparison (Sect. 5.3). Regarding new features, the files were inspected manually for the expected result.

### 5.6 Code Metrics

Metrics for the maintainability and complexity of code as well for the documentation are a useful tool for developers who should aim at maintaing or reaching good scores in these metrics. For the Python code of flex_extract, a number of metrics were calculated for the previous version 7.0.4 and the current version 7.1.

Basic metrics, taken from Lacchia (2019) and calculated through the Python package `radon` (Lacchia, 2019), are the total number of lines of code (`LOC`), the number of logical lines of code (`LLOC`), the number of source lines of code (`SLOC`), the

number of (single) comment lines (`comments`), the number of lines in multi-line comment strings (`multi`), and the number of blank lines (`blank`), with the following relation between these numbers:

$$\mathrm{LOC} = \mathrm{SLOC} + \mathrm{multi} + \mathrm{comment} + \mathrm{blank}. \tag{11}$$

The comparison shown in Table 21 indicates a significant increase not only in the logical lines of code, but even more in `comment` and `multi`, mostly representing an improvement of in-line documentation by splitting large code blocks into

smaller ones, each with a new `docstring`. A so-called `docstring` is a specific multi-line comment for the documentation of functions, methods and classes, describing their input and return values, which can be read by tools for automatic generation of a separate documentation. The re-factorization of code blocks, additional code for new features, and compliance with certain code style rules (e.g., maximum length of lines), about 1000 lines of pure code were added. The ratio of comment lines (`multi` + `comment`) to source-code lines (`SLOC`) grew from 20 % to 117 %.

A further metric for code quality is the so-called *cyclomatic complexity* (CC), also called the McCabe metric since it was developed by Thomas J. McCabe Sneed et al. (2010). It is equal to the number of linearly independent paths through the control flow graph of the code or the number of decisions plus one. A lower CC score indicates lower the complexity which is deemed an advantage. CC is calculated as

$$\mathrm{CC} = E - N + 2P, \tag{12}$$





where $E$ is the number of edges (or also called links) of the graph, $N$ is the number of nodes of the graph and $P$ is the number of connected components (which are sub-graphs from functions independent of the supergraph) (Lacchia, 2019; Beizer, 1990; Sneed et al., 2010). The nodes represent the conditional branch instructions and program junctions, and edges are the segments between such points. Regarding code testing, CC provides a lower-bound number of how many test cases (unit tests) are necessary to provide complete path coverage (Beizer, 1990). This metric was also calculated with the `radon` package (Lacchia,

2019). It provides the CC rank for each function, class and class method. Table 22 gives an overview of the interpretation of these ranks. In general its said that the score should not be above 10, corresponding to rank C to F. From the statistical point of view, Table 23 shows that only 10.3 % of flex_extract version 7.1 code blocks have higher complexity, while in version 7.0.4 this was the case for 30.8 %.

The mean cyclomatic complexity of all code blocks in the new Python code is 5.74 (B); for those blocks with C to F (Table

24), it is 21 (C). In version 7.0.4, the corresponding numbers are 13 (C) and 31.86 (E), indicating a substantial improvement. Table 24 lists all the code blocks with their ranks and scores. For example, the class `ControlFile` was improved significantly, as well as the class renamed from `EIFlexpart` to `EcFlexpart`. On the other hand, the class method `deacc_fluxes` became more complex in version 7.1. This is mainly due to two new features, of ensemble retrieval and the new disaggregation. Nevertheless, the overall code complexity was reduced.

Another software metric is the maintainability index (MI), ranking between 0 and 100; it is a function of `SLOC`, `CC` and the Halstead volume (`V`) (Lacchia, 2019):

$$\mathrm{MI} = \max\left[0, \frac{100}{171}\left(171 - 5.2\ln\mathrm{V} - 0.23\,\mathrm{CC} - \right.\right.$$
$$\left.\left. 16.2\ln\mathrm{SLOC} + 50\sin\sqrt{2.4\mathrm{C}}\right)\right] \tag{13}$$

where C is the fraction of comment lines (converted to radians) (Lacchia, 2019). The Halstead volume is defined as

$$\mathrm{V} = (N_1 + N_2)\log_2(\eta_1 + \eta_2) \tag{14}$$

with $\eta_1$ being the number of distinct operators, $\eta_2$ being the number of distinct operands, $N_1$ the total number of operators and $N_2$ the total number of operands. Table 26 defines the classification ranks. The index is calculated for a complete Python file and Table 26 shows the ranks of version 7.0.4 and 7.1 respectively.

Additionally, we used a source code quality checker program called `pylint` (Thénault, 2001) to support following the Python style guide PEP8 (van Rossum et al., 2001). This tool provides an overall rating index with a maximum number of

10. Applying this tool, flex_extract version 7.0.4 has a rating of -8.77 and version 7.1 a rating of 9.09. This shows a massive improvement in following the official style guides.



## 6  Final remarks and outlook

### 6.1  Conclusions

This paper describes the software package flex_extract v7.1, which retrieves meteorological fields from the ECMWF IFS model and prepares them for the use in the Lagrangian particle dispersion model FLEXPART. The software was initially developed in the 1990ies and underwent various developments to adapt to the ECMWF environment and the data set characteristics. In the past two years, the ECMWF introduced considerable changes in its software environment to retrieve, read and access data as well as the preparation of new data sets. This necessitated a substantial upgrade of flex_extract to adapt to these changes. Additionally, the user community had new requirements for data retrievals which were considered in this version. In the development process, substantial refactoring was carried out, the number of retrievable data sets was increased, user-friendliness was improved, current ECMWF software packages considered, an online documentation was built, and a first set of test cases for future regression testing was created. Furthermore, a recently developed and improved disaggregation method for precipitation fields was implemented as an option.

The number of groups using FLEXPART grew substantially over the past decade and with the new opportunity of publicly available reanalysis data sets there will likely be even more users who will try out and apply FLEXPART for their research. Alongside this reference paper, the newly established git repository on the FLEXPART community website https://flexpart.eu and the online documentation should assist all these users with up-to-date information about changes, releases of new versions, installation and usage including a documentation useful for future developers.

### 6.2  Support

FLEXPART has a community website http://flexpart.eu, where flex_extract as a pre-processor has its own sub page[7]. The website features a ticket system to report issues or feature requests. The tickets can be viewed by anyone, to create a ticket, registration[8] is necessary. There is also a mailing list for discussion among FLEXPART / FLEXTRA users and with developers, where questions may be asked or experiences be shared, including pre- and post-processing issues. Announcements for all FLEXPART users, such as new releases, are distributed through the list as well.

### 6.3  Future work

In its current status, the on-line documentation provides a basic reference. In the future, more examples should be provided, including answers to typical user questions and workarounds for known problems. Information about updates and new releases will also be implemented in this documentation.

It is also intended to optionally retrieve meteorological fields from ECMWF needed as input for the WRF model to support the FLEXPART-WRF community.

---

[7]https://www.flexpart.eu/wiki/FpInputMetEcmwf; Last accessed: 17.08.2019
[8]https://www.flexpart.eu/wiki/RegisteredUser; Last accessed: 17.08.2019



The unification of the three-dimensional fields into a single file shall be removed from the Fortran code as this is a simple task that can be fulfilled more efficiently and transparently with `ECcodes` command-line tools.

The ERA5 reanalysis has ensemble members stored in the *enda* stream, but the flux data have a different accumulation period and therefore are not yet retrievable. It is planned to allow the retrieval of these ensemble members in the future. Up to now, it is possible to set flex_extract to retrieve reduced Gaussian fields. This should be extended to include the octahedral reduced Gaussian grid.


The hybrid vertical velocity $\dot{\eta}$ is now stored not only for the operational forecasts but also for the new reanalyses, thus the need to calculate it is diminishing. Therefore, the option to calculate it on the native Gaussian grid might be removed in the future. This would allow to do all the remaining calculations in Python3 without resorting to Fortran code.

The flex_extract software is currently provided as a compressed tar file. In the future, a package shall be made available to be installed as a system software for all users. Then, only user-specific data need to reside in a user directories.


*Code and data availability.* The flex_extract software is a collection of Python scripts, Shell scripts and a Fortran program which alltogether are licensed under the CC-BY-4.0 software license. The latest version of the code is available on the flex_extract project webpage (https://www.flexpart.eu/wiki/FpInputMetEcmwf; Last accessed: 12.12.2019) which is part of the FLEXPART community website and where also the documentation is hosted (https://www.flexpart.eu/flex_extract/; Last accessed: 12.12.2019).


The flex_extract version 7.1 described here as well as previous versions are available from https://www.flexpart.eu/wiki/FpInputMetEcmwf (links to tarball and git repository). The exact version at the time of manuscript submission is archived on Phaidra (https://phaidra.univie.ac.at/view/o:1070149 with DOI:10.25365/phaidra.130), the permanent secure storage of the University of Vienna.

The software comes with a number of test cases which should be used by developers in the future. Some tests need additional reference data which have to be downloaded from the project website in addition.


The following open-source libraries have to be available in addition to the libraries mentioned in the installation section in order to run the flex_extract test cases: `numpy/scipy` (Walt et al., 2011), `pandas` (McKinney, 2010), `xarray` (Hoyer and Hamman, 2017), `pytest` (Krekel, 2019), `mock` (Foord and the mock team, 2019). For the generation of the online documentation, the Python package `sphinx` (Brandl, 2019) is required, and for the documentation of the Fortran part FORD[9].

The current version 7.1 of flex_extract was developed under GNU/Linux and was tested only on this platform. Application under other operating systems may be possible, but without supported by the developers.


## Appendix A: Installation instructions

First of all, download the release version from the FLEXPART community website. Alternatively, if git is installed, you may clone the latest version from our git repository master branch.


```
git clone --single-branch --branch master
          https://www.flexpart.eu/gitmob/flexpart
```

---

[9]http://fortranwiki.org/fortran/show/FORD, accessed 20 Dec 2019





Currently, the software was only tested for a GNU/Linux environment. The installation process depends on the user group (see Sect. 2.1) and the application mode (see Sect. 3.1). One should first decide for the modes and then follow the compact instructions in the corresponding subsections. For more details see the instructions in the online documentation.

## A1    Registration and licenses

Table 6 summarizes which registration is required. Follow the given links from the literature to the registration websites (or footnotes).

A separate license has to be accepted for each ECMWF public data set, regardless of the user group. For the ERA-Interim and CERA-20C datasets this can be done at the website for "Available ECMWF Public Datasets"[10]. Log in and follow the license links on the right side for each data set and accept it. For the ERA5 data set this has to be done at the "Climate Data Store (CDS) website"[11]. Log in and select, on the left panel, product type "Reanalysis" for finding ERA5 data sets. Then follow any link with ERA5 to the full data set record, click on tab "Download data" and scroll down. There is a section "Terms of use" where the "Accept terms" button has to be clicked. The licenses for member state users are accepted by the user when receiving a so-called "Token", which generates new passwords for each log in.

## A2    Preparing the system

### Remote mode

ECMWF servers provide all required libraries (see Table 7) via a module system. Flex_extract takes care of loading the right modules at runtime except the Python3 module which needs to be loaded prior to its execution by `module load python3`. This is due to the fact that flex_extract is first started to prepare the job script with the correct settings before submitting the job to the batch queue.

### Gateway mode

In this mode, access to the ECMWF computing and archiving facilities is enabled through an *ECaccess* gateway server on a local member state server. The *ECaccess* framework is necessary to interactively submit jobs to the ECMWF batch system and to transfer files between ECMWF and local gateway server. As a consequence, a member state gateway server has to be established[12] and a so-called association[13] has to be created to use the *ECaccess* file transfer service `ectrans`.

The next step is to create an *ECaccess* certificate to authorize the file transfers and job submissions. This certificate has to be renewed periodically (every 7 days). The certificate is created by executing the command `ecaccess-certificate-create` on the command line of the local gateway server and the user is prompted for the ECMWF member state user name and a password (generated by a token).

---

[10]https://confluence.ecmwf.int/display/WEBAPI/Available+ECMWF+Public+Datasets; Last accessed: 11.11.2019
[11]https://cds.climate.copernicus.eu/cdsapp#!/search?type=dataset; Last accessed: 11.11.2019
[12]https://confluence.ecmwf.int/display/ECAC/ECaccess+Home; Last accessed: 31.10.2019
[13]https://confluence.ecmwf.int/download/attachments/45759146/ECaccess.pdf see page 17 ff. for instructions; Last accessed: 28.10.2019





```
$ ecaccess-certificate-create
      Please enter your user-id: example_username
      Your passcode: ***
```

Additional dependencies on the local gateway server are `Python3` and the Python packages `NumPy` and `genshi`. Use the package management system of your Linux distribution which required admin rights. The installation was tested under GNU/Linux Debian buster and Ubuntu 18.04 Bionic Beaver. The following installation instructions refer to a Debian-based system and use `apt-get` as package manager; of course, other package managers (e. g. `aptitude`), or other GNU/Linux distributions can be used as well.

```
      apt-get install python3
      apt-get install python3-genshi
apt-get install python3-numpy
```

**Local mode**

For the local mode, all software dependencies listed in Section 3.2 have to be provided. The installation process is the same for the *member* and *public* access modes. Use the package management system of your Linux distribution (requires admin rights) to establish the dependencies if not already available.

```
apt-get install python3
      apt-get install python3-eccodes
      apt-get install python3-genshi
      apt-get install python3-numpy
      apt-get install gfortran
apt-get install fftw3-dev
      apt-get install libeccodes-dev
      apt-get install libemos-dev
```

As currently the CDS and ECMWF API packages are not available as Debian packages, they need to be installed outside the Debian (Ubuntu etc.) package management system. The CDS API (`cdsapi`) is required for ERA5 data and the ECMWF Web API (`ecmwf-api-client`) for all other public datasets. Since public users currently do not have access to the full ERA5 data set, they can skip the installation of the CDS API. The recommended way is to use the Python package management system `pip`:

```
      apt-get install pip
      pip install cdsapi
pip install ecmwf-api-client
```

Note that if you would like to use Anaconda Python we recommend you follow the installation instructions of Anaconda Python Installation for Linux and then install the eccodes package from conda with:

```
      conda install conda-forge::python-eccodes
```

Both user groups have to provide keys with their credentials for the Web APIs in their home directory. Therefore, follow
these instructions:





**ECMWF Web API** Go to MARS access website[14] and log in with your credentials. Afterwards, on this site in section "Install ECMWF KEY" the key for the ECMWF Web API should be listed. Please follow the instructions in this section under 1 (save the key in a file `.ecmwfapirc` in your home directory).

**CDS API** Go to CDS API registration[15] and register there too. Log in at the cdsapi website and follow the instructions at
section "Install the CDS API key" to save your credentials in a `.cdsapirc` file.

Since a single retrieval run of flex_extract can take a while, it is recommended to do some basic tests for the local access method to identify problems with the Web APIs early on. A very simple test retrieval for both Web APIs are enough to be sure that everything works.

For the ECMWF Web API and as a *member* user please use this piece of Python code:

```
from ecmwfapi import ECMWFService

     server = ECMWFService('mars')

     server.retrieve({
'stream'    : "oper",
     'levtype'   : "sfc",
     'param'     : "165.128/166.128/167.128",
     'dataset'   : "interim",
     'step'      : "0",
'grid'      : "0.75/0.75",
     'time'      : "00/06/12/18",
     'date'      : "2014-07-01/to/2014-07-31",
     'type'      : "an",
     'class'     : "ei",
'target'    : "download_erainterim_ecmwfapi.grib"
     })
```

For the ECMWF Web API and as a *public* user please use that piece of Python code:

```
     from ecmwfapi import ECMWFDataServer

server = ECMWFDataServer()

     server.retrieve({
     'stream'    : "enda",
     'levtype'   : "sfc",
'param'     : "165.128/166.128/167.128",
     'dataset'   : "cera20c",
     'step'      : "0",
     'grid'      : "1./1.",
     'time'      : "00/06/12/18",
'date'      : "2000-07-01/to/2000-07-31",
     'type'      : "an",
     'class'     : "ep",
     'target'    : "download_cera20c_ecmwfapi.grib"
     })
```

---

[14]https://confluence.ecmwf.int//display/WEBAPI/Access+MARS; Last accessed: 20.10.2019
[15]https://cds.climate.copernicus.eu/api-how-to; Last accessed: 25.10.2019





Extraction of ERA5 data via CDS API (currently only for *member* users) might take time as currently, at the time of

publication, there is a high demand for ERA5 data. Therefore, as a simple test for the API, just retrieve pressure-level data

(even if that is NOT what we need for FLEXPART), as they are stored on disk and don't need to be retrieved from MARS

(which is the time-consuming action):

Please use this piece of Python code to retrieve a small sample of ERA5 pressure levels:

```
import cdsapi

       c = cdsapi.Client()

       c.retrieve("reanalysis-era5-pressure-levels",
{
       "variable": "temperature",
       "pressure_level": "1000",
       "product_type": "reanalysis",
       "year": "2008",
"month": "01",
       "day": "01",
       "time": "12:00",
       "format": "grib"
       },
"download_cdsapi.grib")
```

An example for retrieving ERA5 data from MARS is shown below and can be tested if the code from above worked.

```
       import cdsapi

       c = cdsapi.Client()
       c.retrieve('reanalysis-era5-complete',
       {
       'class'   : 'ea',
       'expver'  : '1',
'stream'  : 'oper',
       'type'    : 'fc',
       'step'    : '3/to/12/by/3',
       'param'   : '130.128',
       'levtype' : 'ml',
'levelist': '135/to/137',
       'date'    : '2013-01-01',
       'time'    : '06/18',
       'area'    : '50/-5/40/5',
       'grid'    : '1.0/1.0',
'format'  : 'grib',
       }, 'download_era5_cdsapi.grib')
```

## A3   Building flex_extract

**Remote mode**

First, log in on one of the ECMWF servers, such as *ecgate* or *cca/ccb*.





```
scp <localuser>@<localmachine.tld>:</path/to/tarfile/>
              $HOME/
       cd $HOME
       tar xvf flex_extract_vX.X.tar.gz
       cd flex_extract_vX.X
```

Substitute the `<localuser>` and `<localmachine.tld>` placeholders with your local user name and the IP name or address of your local machine. Untar the flex_extract release file and change into the flex_extract root directory.

Eventually, adapt the parameters (described in Table 12 and 10) in the `setup.sh` script and execute it.

Flex_extract uses the email address connected to the user account to notify the user about successful or failed installation.

**Gateway mode**

The actual execution of flex_extract with retrieval and preparation of the data will be run on an ECMWF servers. The only difference is the preparation of the job script, which is done on the local gateway server and sent to ECMWF servers by the *ECaccess* services.

Unpack the release tarball and change into its directory. Substitute `X.X` with the actual release version number.

```
       tar xvf flex_extract_vX.X.tar.gz
cd flex_extract_vX.X
```

Afterwards, prepare the setup.sh script by configuring its parameters (described in Table 12 and 10) and execute it. The makefile has to be selected according to the selection of the target, e.g. *ecgate* or *cca/ccb* servers. In this mode the `DESTINATION` and `GATEWAY` parameters have to be set to be able to use the `ectrans` service. A confguration job script is then sent to the ECMWF batch queue and flex_extract uses the email address connected to the user account to notify the user about successful

or failed installation.

**Local mode**

Since flex_extract compiles the Fortran program `preconvert` during the installation process, a corresponding makefile has to be provided. Flex_extract comes with two makefiles for the local mode prepared for the gfortran (https://gcc.gnu.org/fortran/) and the ifort (https://software.intel.com/en-us/fortran-compilers)compiler. The gfortran version assumes that eccodes and emoslib

are installed as distribution packages. It is necessary to adapt the two parameters `ECCODES_INCLUDE_DIR` and `ECCODES_LIB` in these makefiles if other than standard paths are used.

Hence, if needed, prepare the Fortran makefile for your environment by starting from one of the two provided makefiles `makefile.local.gfortran` or `makefile.local.ifort`. They can be found at `flex_extract_vX.X/Source/Fortran`, where vX.X should be substituted with the current version number. Edit the paths to the `eccodes`

library on your local machine.

Eventually, adapt the command line parameters (described in Table 12 and 10) in the `setup.sh` script in the root directory of flex_extract and execute it.





### A4   Installation test

The most common errors in applying flex_extract arise from wrong installation and settings regarding the libraries for the

Fortran program. Therefore it is useful to do a simple test with a prepared minimal data set. The following instructions have to

be executed on the local system for the local mode and on the ECMWF servers in the remote and gateway mode.

From the flex_extract root directory change into the `Testing/Installation/Convert/` directory and execute the
Fortran program by

```
cd Testing/Installation/Convert
# execute the Fortran progam
../../../Source/Fortran/calc_etadot_fast.out
```

The installation was successful if you obtain on standard output:

```
readspectral:           1  records read
readlatlon:             8  records read
STATISTICS:  98842.4598 98709.7359  5120.5385
readlatlon:             4  records read
readlatlon:             4  records read
readlatlon:             4  records read
SUCCESSFULLY FINISHED CONVERT_PRE: CONGRATULATIONS
```

Note that on ECMWF servers the flex_extract root directory is placed in the `$HOME` directory.

### Appendix B:  How to use flex_extract

Flex_extract is a command-line tool which can be started by executing the `submit.py` script in the Python source directory or
more preferably with an upstream shell script `run.sh` which calls the `submit.py` script with its corresponding command-
line arguments. Therefore, the user should navigate to the `Run` directory, where the shell script is located.

`cd <path-to-flex\_extract\_vX.X>/Run`

with `X.X` as the placeholder for the version number.

This directory contains all information necessary to run flex_extract. The only files which might need modifications by the

user are the `run.sh` script and the selected `CONTROL` file within the `Control` directory. This directory contains a sample

set of the current range of possible data set retrievals.

This section describes the basic steps to start a flex_extract retrieval within the different modes based on an example. More

details about the usage can be found in Section 4 and in the online documentation, especially specifics of different data sets

and `CONTROL` file parameters.

For the first data retrieval it is recommended to use one of the example `CONTROL` files stored in the `Control` directory

to avoid unnecessary problems. We recommend to extract CERA-20C data since they are usually not highly demanded and

guarantee quick processing for the best testing experience.

#### Remote and gateway modes

For member state users it is recommended to use the remote or gateway mode, especially for more demanding tasks, to retrieve
and convert data on ECMWF machines and to transfer only the final output files to the local host.





The only difference between both modes is the location where flex_extract will be started from. In the remote mode we work
directly on the ECMWF server, therefore login to the ECMWF server of your choice and change to the `Run` directory as shown
above. Remember, at ECMWF servers flex_extract is always installed in the `$HOME` directory. To be able to start the program,
please load the Python3 environment with the module system first.

```
module unload python
module load python3
```

Within the gateway mode, only a change into the Run directory of flex_extract on the gateway server is necessary.

Otherwise, the rest of the working steps are the same in both modes. Now, open the `run.sh` script and modify the parameter
block marked in the file as shown below. The parameters are described in Table 13.

```
# --------------------------------------
# AVAILABLE COMMANDLINE ARGUMENTS TO SET
#
# THE USER HAS TO SPECIFY THESE PARAMETERS:

QUEUE='ecgate'
START_DATE=None
END_DATE=None
DATE_CHUNK=None
JOB_CHUNK=3
BASETIME=None
STEP=None
LEVELIST=None
AREA=None
INPUTDIR=None
OUTPUTDIR=None
PP_ID=None
JOB_TEMPLATE='job.temp'
CONTROLFILE='CONTROL_CERA'
DEBUG=0
REQUEST=2
PUBLIC=0
```

This would retrieve a one day (08.09.2000) CERA-20C dataset with 3 hourly temporal resolution and a small 1° domain
over Europe on the ECMWF server *ecgate*. For the ECMWF *cca/ccb* servers, the parameter `QUEUE` has to be adapted. Since
the `ectrans` parameter in the `CONTROL_CERA` file is set to 1 the resulting output files will be transferred to the local gateway
into the path stored in the destination, provided that the destination was correctly setup. The parameters listed in the `run.sh`
script would overwrite existing settings from the `CONTROL` file.

Starting the retrieval process will be done by executing the script by `./run.sh`.

Flex_extract will print some information about the job on standard output. If there is no error in the submission to the
ECMWF server a message like this will be shown:

```
---- On-demand mode! ----
The job id is: 10627807
```





```
You should get an email per job with
           subject flex.hostname.pid
       FLEX_EXTRACT JOB SCRIPT IS SUBMITTED!
```

Once submitted, the job status can be checked by using the command `ecaccess-job-list`. At the end of the job, the user should receive an email with a detailed protocol of what was done and if the job was successful.

In case the job failed, the subject will contain the keyword *ERROR!* and the job name. Then, the user can check the email or on ECMWF servers in the `$SCRATCH` directory for debugging information.

In the `$SCRATCH` directory on *ecgate* it is recommended to list the content with `ls -rthl` to list the most recent logs and temporary retrieval directories (usually `pythonXXXXX`, where `XXXXX` is the process id). Under `pythonXXXXX` a copy of the `CONTROL` file is stored under the name `CONTROL`, the protocol is stored in the file `prot` and the temporary files as well

as the resulting files are stored in a directory `work`. The original name of the `CONTROL` file can be found within this new file under parameter `controlfile`.

If the job was submitted to the High Performance Computer (HPC) (`QUEUE` is cca or ccb) you may login to the HPC and look into the directory `/scratch/ms/ECGID/ECUID/.ecaccess_do_not_remove` for job logs. The working directories are deleted after job failure and thus normally cannot be accessed.

If the resulting files can not be found in the destination path of the local gateway server, it can be checked if the files are still to be transferred to the local gateway server by using the command `ecaccess-ectrans-list`.

After this test retrieval was successful, feel free to try changing the `CONTROL` file parameters described in Tables 8 and 9 and by selecting other `CONTROL` files. Please mind the considerations of application in Section 4.

**Local mode**

Since this mode can be used by member and public users, we show an example for both user groups. Open the `run_local.sh` file and adapt the parameter block marked in the file as shown for the corresponding user group. The parameters are described in Table 13.

Take this setting as member-state user:

```
       # --------------------------------------
# AVAILABLE COMMANDLINE ARGUMENTS TO SET
       #
       # THE USER HAS TO SPECIFY THESE PARAMETERs:
       #

QUEUE=''
       START_DATE=None
       END_DATE=None
       DATE_CHUNK=None
       JOB_CHUNK=None
BASETIME=None
       STEP=None
       LEVELIST=None
       AREA=None
```





```
          INPUTDIR='./Workspace/CERA'
OUTPUTDIR=None
          PP_ID=None
          JOB_TEMPLATE=''
          CONTROLFILE='CONTROL_CERA'
          DEBUG=0
REQUEST=0
          PUBLIC=0
```

and take this setting as a public user:

```
          # --------------------------------------
          # AVAILABLE COMMANDLINE ARGUMENTS TO SET
#
          # THE USER HAS TO SPECIFY THESE PARAMETERs:
          #

          QUEUE=''
START_DATE=None
          END_DATE=None
          DATE_CHUNK=None
          JOB_CHUNK=None
          BASETIME=None
STEP=None
          LEVELIST=None
          AREA=None
          INPUTDIR='./Workspace/CERApublic'
          OUTPUTDIR=None
PP_ID=None
          JOB_TEMPLATE=''
          CONTROLFILE='CONTROL_CERA.public'
          DEBUG=0
          REQUEST=0
PUBLIC=1
```

This would retrieve a one day (08.09.2000) CERA-20C dataset with 3 hourly temporal resolution and a small 1° domain over Europe. The destination location for this retrieval is set by the `INPUTDIR` parameter and will be the `Workspace/CERA*` directory within the current `Run` directory. This can be changed to whatever path is preferred. The parameters listed in `run_local.sh` would overwrite existing settings in the `CONTROL` file.

Starting the retrieval process will be done by executing the script by `./run\_local.sh`.

While a job submission on the local host is convenient and easy to monitor (on standard output), there are a few caveats with this option.

There is a maximum size of 20 GB for single retrievals via ECMWF Web API. Normally this is not a problem but for global fields with T1279 resolution and hourly time steps the limit may already apply. If the retrieved MARS files are large but the

resulting files are relatively small (small local domain, but large time period) then the retrieval to the local host may be inefficient since all data must be transferred via the Internet. This scenario applies most notably if `ETADOT` has to be calculated via the





continuity equation as this requires global fields even if the domain is local and small. In this case, job submission via *ecgate* might be a better choice. It really depends on the patterns used and also on the speed of the internet.

After this test retrieval was successful, feel free to try changing the `CONTROL` file parameters described in Tables 8 and 9
and by selecting other `CONTROL` files. Please mind the considerations of application in Section 4.

*Author contributions.*   A. Philipp revised the complete software package (except for the Fortran part) and applied the necessary changes to keep the software up-to-date with the ECMWF software environment. She coordinated and added new implementations and guided the evaluation. She wrote the online documentation as well as most of the manuscript.

L. Haimberger is the original author of the software and provided the first implementation for the use of the ECMWF Web API and the
retrieval of ensemble members. He participated in writing introductory and history parts as well as giving feedback on all other parts.

P. Seibert revised the Fortran code and provided the Fortran code documentation and test cases, and wrote the respective section of the manuscript. She also gave feedback on all other parts, and contributed to editing the final manuscript version.

*Competing interests.*   The authors declare that they have no conflict of interest.

*Acknowledgements.*   Over the years, the development of flex_extract was partly funded by the CTBTO. We thank the ZAMG for providing access to the ECMWF MARS archive and the hosting of the community website. We would also like to thank the ECMWF user support for their assistance in converging the software environment to the current state and for their many publicly available code examples for working with GRIB files. Additionally, we thank Anne Fouilloux for an initial version of the Python routines. Moreover, we thank the users for their feedback and questions which made it possible to make progress in user friendliness, eliminate bugs and react on requirements.





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



**Table 1.** Overview of ECMWF data sets with associated parameters required in MARS requests (Berrisford et al., 2011; Laloyaux et al., 2016; ECMWF, 2019e, h). DET-FC stands for "Deterministic forecast", ENS-DA for "Ensemble data assimilation", ENS-CF for "Ensemble control forecast", ENS-CV for "Ensemble validation forecast" and ENS-PF for "Ensemble perturbed forecast". All times are in UTC, all steps in hours. Dates are written as DD/MM/YYYY (day optional). Steps and members are written in the format of Start/End/Step. The information for operational data sets is valid at the time of publication (except ENS-CV – deprecated since August 8th, 2016), and may change in the future. The grid type (Oxxx) for the operational data refers to the octahedral reduced Gaussian grid. The identification parameter "Data set" is to be used by public users only. Note that there is also the ERA40 reanalysis; however, as it has been superseded by ERA-Interim and ERA5 and thus rarely used nowadays, it is not included here (but flex_extract should still be applicable).

| | Operational data | | | | | Reanalyses | | |
|---|---|---|---|---|---|---|---|---|
| | DET-FC | ENS-DA | ENS-CF | ENS-CV | ENS-PF | ERA-Interim | ERA5 | CERA-20C |
| **Period of availability** | | | | | | | | |
| | 12/1985 – ongoing | 22/06/2010 – ongoing | 01/05/1994[2] – ongoing | 12/09/2006 – 08/03/2016 | 12/09/2006[3] – ongoing | 01/1979 – 12/2018 | 01/1979 – ongoing[4] | 09/1901 – 12/2010 |
| **Identification** (MARS keywords) | | | | | | | | |
| Class | od | od | od | od | od | ei | ea | ep |
| Stream | oper | enda/elda[1] | enfo | enfo | enfo | oper | oper | enda |
| Field type | fc/an | fc/an | cf | cv | pf | fc/an | fc/an | fc/an |
| Data set | – | – | – | – | – | interim | – | cera20c |
| **Time** (where forecast starts or analysis is valid) | | | | | | | | |
| Forecast | 00/12 | 06/18 | 00/12 | 00/12 | 00/12 | 00/12 | 06/18 | 18 |
| Analysis | 0/6/12/18 | 0/6/12/18 | – | – | – | 0/6/12/18 | 0/1/.../23 | 0/1/.../23 |
| **Step** (available forecast steps) | | | | | | | | |
| Forecast | 0/125/1 | 1/12/1 3,6,12[7] | 0/90/1 93/144/3 150/360/6 | 0/144/3 150/360/6 | 336 | 3/240/3[5] | 0/18/1 | 3/24/3 |
| **Horizontal grid type and resolution, number of vertical levels** | | | | | | | | |
| Grid[8] | O1280 | O640 | O640 | O640 | O640 | T255 (0.75°) | T639 (0.25°) | T159 (1.25°) |
| Levels | 137 | 137 | 91 | 91 | 91 | 60 | 137 | 91 |
| **Ensemble members** | – | 0/50/1 | – | – | 1/50/1 | – | – | 1/10/1 |
| **Availability of** $\dot{\eta}$ | yes[6] | yes | no | no | no | no | yes | yes |

[1] From 22/06/2010 to 18/11/2013, ENS-DA was stored in stream ENDA, afterwards in stream ELDA.

[2] Exists since 11/1992, but the available dates were unregular in the beginning before 01/05/1994.

[3] The data set exists from 11/1992, but model level data are available only from 12/09/2006 on.

[4] Available with a delay of ca. 3 months. Fast track data with shorter delay are now also available, but subject to possible revisions

[5] For public users, the forecast model level fields are not available.

[6] Available as MARS parameter since 04/06/2008.

[7] On 11/06/2019, the steps changed from 1/12/1 to the single steps 3,6,12.

[8] See Table 4 for correspondence of grid types.





**Table 2.** List of the evolution of the spatial resolution of the IFS operational forecasts. Changes are marked in bold. The ensemble data are usually provided with higher resoltion for Lag A (1–10 d) than for Lag B (10–15 d). The first part of each entry is the horizontal resolution marked with a "T" for spectral or "O" for octahedral representation. The second part, marked with "L", is the number of vertical model levels. In the case of ensembles, the number N of members is written in front of the resolution as N*. Source: Palmer et al. (1997); Buizza et al. (2003); ECMWF (2019c, e, f, g).

| | DET-FC | ENS-DA | ENS-CF | ENS-CV | ENS-PF |
|---|---|---|---|---|---|
| 20/04/1983 | **T106L16** | | | | |
| 13/05/1986 | T106**L19** | | | | |
| 17/09/1991 | **T213**L19 | | | | |
| 17/19/1991 | T213**L31** | | | | |
| 01/05/1994 | T213L31 | | **T63L19** | | |
| 10/12/1996 | T213L31 | | **T159L31** | | |
| 01/04/1998 | **T319**L31 | | T159L31 | | |
| 09/03/1999 | T319**L50** | | T159L31 | | |
| 12/10/1999 | T319**L60** | | T159**L40** | | |
| 21/11/2000 | **T511**L60 | | **T255**L40 | | |
| 01/02/2006 | **T799L91** | | **Lag A T399L62** | | |
| | | | **Lag B T255L62** | | |
| 12/09/2006 | T799L91 | | | **Lag A 2*T399L62** | **Lag A 50*T399L62** |
| | | | | **Lag B 2*T255L62** | **Lag B 50*T255L62** |
| 26/01/2010 | **T1279**L91 | | **Lag A T639**L62 | Lag A 2***T639**L62 | Lag A 50***T639**L62 |
| | | | **Lag B T319**L62 | Lag B 2***T319**L62 | Lag B 50***T319**L62 |
| 22/06/2010 | T1279L91 | **25*T399L91** | Lag A T639L62 | Lag A 2*T639L62 | Lag A 50*T639L62 |
| | | | Lag B T319L62 | Lag B 2*T319L62 | Lag B 50*T319L62 |
| 01/11/2011 | T1279L91 | 25*T399L91 | Lag A T639L62 | Lag A 2*T639L62 | Lag A 50*T639L62 |
| | | | Lag B T319L62 | Lag B 2*T319L62 | Lag B 50*T319L62 |
| 25/06/2013 | T1279**L137** | 25*T399**L137** | Lag A T639L62 | Lag A 2*T639L62 | Lag A 50*T639L62 |
| | | | Lag B T319L62 | Lag B 2*T319L62 | Lag B 50*T319L62 |
| 19/11/2013 | T1279L137 | 25*T399L137 | Lag A T639L62 | Lag A 2*T639**L91** | Lag A 50*T639L62 |
| | | | Lag B T319L62 | Lag B 2*T319**L91** | Lag B 50*T319L62 |
| 20/11/2013 | T1279L137 | 25*T399L137 | Lag A T639L91 | Lag A 2*T639L91 | Lag A 50*T639**L91** |
| | | | Lag B T319L62 | Lag B 2*T319L91 | Lag B 50*T319**L91** |
| 08/03/2016 | **O1280**L137 | 25***O640**L137 | **O640**L91 | deprecated | 50***O640**L91 |
| 11/06/2019 | O1280L137 | **50***O640L137 | O640L91 | deprecated | 50*O640L91 |





**Table 3.** List of the evolution of forecast steps and forecast start times for data sets DET-FC and ENS-CF. "Lag s" denotes different temporal resolution for forecast ranges s; "#steps" is the total number of steps. Source: (ECMWF, 2019e)

| | DET-FC | | | | | ENS-CF | | | |
|---|---|---|---|---|---|---|---|---|---|
| | #steps | Lag 1 | Lag 2 | Lag 3 | | #steps | Lag 1 | Lag 2 | Lag 3 |
| 01/04/1985 | 20[1] | 12/240/12 | | | 01/04/1994 | 33 | 0/12/3 | 18/120/6 | 132/240/12 |
| | forecast start time 12 UTC | | | | | forecast start time 12 UTC | | | |
| 01/07/1985 | 30 | 6/144/6 | 156/240/12 | | 31/07/1997 | 55 | 0/12/3 | 18/120/6 | 132/504/12 |
| 15/11/1990 | 32 | 3/12/3 | 18/144/6 | 150/240/12 | 09/06/1999 | 65 | 0/12/3 | 18/240/6 | 252/504/12 |
| 20/01/1999 | 42 | 3/12/3 | 18/240/6 | | 25/03/2003 | two forecast start times per day: 0/12 UTC | | | |
| 12/09/2000 | two forecast start times per day: 0/12 UTC | | | | 29/09/2004 | 63 | 0/240/6 | 252/504/6 | |
| 24/10/2000 | 52 | 3/72/3 | 78/240/6 | | 13/09/2006 | 85 | 0/132/3 | 138/240/6 | 252/504/12 |
| 29/06/2005 | 85 | 0/132/3 | 138/240/6 | 252/504/12 | 22/06/2015 | four forecast times per day: 0/6/12/18 UTC | | | |
| 05/10/2005 | 87 | 0/144/3 | 150/240/6 | 252/504/12 | " | 49 (6/18 UTC) | 0/144/3 | | |
| 14/03/2006 | 57 | 0/96/3 | 102/240/6 | | 23/11/2016 | 145 (0/12UTC) | 0/90/1 | 93/144/3 | 150/260/6 |
| 13/09/2006 | 65 | 0/144/3 | 150/240/6 | | " | 109 (6/18UTC) | 0/90/1 | 93/144/3 | |
| 16/11/2011 | 125 | 0/90/1 | 93/144/3 | 150/240/6 | – | – | | | |

[1] Only surface fields.





**Table 4.** Approximate correspondences between spectral, Gaussian, and latitude / longitude grid resolutions. Source: ECMWF (2019d, e); Berrisford et al. (2011); Laloyaux et al. (2016). For the spectral grid the truncation number is denoted by "T" or "$T_L$" where the latter means linear spectral truncation. The corresponding reduced Gaussian grids are denoted by "N" followed by the number of lines between the pole and the equator. Only linearly truncated grids can be selected with flex_extract. The new octahedral grid is denoted by "$T_{CO}$", meaning "spectral cubic octahedral"; they correspond to a octahedral reduced Gaussian grid denotes with an "O".

| Spectral | Gaussian Grid | | Lat / Lon |
|---|---|---|---|
| T63 | N48 | 200 km | 1.875° |
| $T_L$95 | N48 | | 1.875° |
| T106 | N80 | 120 km | 1.125° |
| $T_L$159 | N80 | 120 km | 1.125° |
| T213 | N160 | | 0.5625° |
| $T_L$255 | N128 | 80 km | 0.75°(*) |
| $T_L$319 | N160 | 60 km | 0.5625° |
| $T_L$399 | N200 | 50 km | 0.45° |
| $T_L$511 | N256 | 40 km | 0.351° |
| $T_L$639 | N320 | 31 km | 0.25°(*) |
| $T_L$799 | N400 | 25 km | 0.225° |
| $T_L$1279 | N640 | 16 km | 0.141° |
| $T_{CO}$1279 | O1280 | 9 km | 0.07° |

(*) As GRIB1 only supports three decimals, ECMWF recommends to round the resolutions to $0.75°$ in the case of ERA-Interim (exact value: $0.703125°$) and to $0.25°$ for ERA5 (exact value: $0.28125$) (ECMWF, 2016a, b). See also Table 1.





**Table 5.** Directory structure of the flex_extract v7.1 root directory.

| File / subdirectory | Content | Description |
|---|---|---|
| `Documentation/` | `html/` | offline version of documentation |
| `For_developers/` | `Flowcharts` | source and PNG files of flow diagrams |
| | `FORD` | source files for Fortran code documentation |
| | `Sphinx` | source files for documentation |
| | `*.xls, *.sh, *` | documentation files, scripts and infos for developers |
| `Run/` | `Control/` | contains all example `CONTROL` files |
| | `Jobscripts/` | empty after distribution download; later contains korn shell job scripts |
| | `Workspace/` | not present before first local retrieval; contains downloaded data in *local mode* |
| | `ECMWF_ENV` | contains infos about user credentials |
| | `run.sh` | top-level script to start flex_extract |
| | `run_local.sh` | top-level script to start flex_extract in *local mode* |
| `Source/` | `Fortran` | complete Fortran program incl. makefiles |
| | `Python` | Python source files |
| | `Pythontest` | Python unit tests |
| `Templates/` | `compilejob.template` | template for the installation on ECMWF server |
| | `convert.nl` | namelist template for the `calc_etadot` program |
| | `ECMWF_ENV.template` | template for the ECMWF user credentials |
| | `ecmwf_grib1_table_128` | table for the assigment of parameter names and ids |
| | `job.template` | ob script template for ECMWF batch mode before the installation took place |
| | `job.temp` | job script template after installation (now includes settings such as version number) |
| `Testing/` | `Installation` | data for an installation check |
| | `Regression` | regression test cases |
| `CODE_OF_CONDUCT.md` | | rules for contribution to flex_extract |
| `LICENSE.md` | | full license text |
| `README.md` | | short introduction to the software |
| `setup.sh` | | installation script |



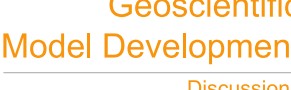

**Table 6.** Necessary account registrations per user and application mode for each data set. The registration procedure is indicated by numbers 1–3 and explained below.

| Data sets | Member-state user | | | Public user |
|---|---|---|---|---|
| | Remote | Gateway | Local | Local |
| Operational | 1 | 1 | 1, 2 | - |
| ERA-Interim | 1 | 1 | 1, 2 | 2 |
| CERA-20C | 1 | 1 | 1, 2 | 2 |
| ERA5 | 1 | 1 | 3 | - |

| No. | Registration procedure |
|---|---|
| 1 | Access as a member-state user. Account granted by the Computing Representative. Credentials have to be provided during installation. |
| 2 | Access through the ECMWF Web API. One needs to sign in at the ECMWF Web API and to configure the ECMWF key as described (ECMWF, 2019k). Member state users can sign in with their credentials. Public users have register for obtaining an account. |
| 3 | Access through the CDS API (Copernicus, 2019). Registration at CDS and configuration of the CDS key needed. |

**Table 7.** General software and library dependencies for flex_extract Python and Fortran parts in all application modes.

| Python | Fortran |
|---|---|
| Python3 | gfortran |
| numpy | fftw3 |
| genshi | emoslib |
| ecCodes for Python | ecCodes for Fortran |







**Figure 4.** General overview of the work flows and work locations in the different application modes: (a) remote, (b) gateway, and (c) local mode. The files and scripts used in each mode are outlined.





**Table 8.** Part 1 of the overview of `CONTROL` file parameters. A more detailed description on parameter handling, setting and value ranges is given in the supplemental material.

| Parameter | Default value | Format | Description |
|---|---|---|---|
| **Time section** | | | |
| START_DATE | None | String [YYYYMMDD] | first day of retrieval period |
| END_DATE | None | String [YYYYMMDD] | last day of retrieval period |
| DATE_CHUNK | 3 | Integer | number of days within one MARS request |
| DTIME | None | Integer | time step |
| BASETIME | None | Integer | end time for half-day retrievals |
| **Data section** | | | |
| CLASS | None | String [xx] | data set class identifier in MARS archive |
| DATASET | None | String | public data set identifier |
| STREAM | None | String [xxxx] | identifier for forecasting stream |
| NUMBER | 'OFF' | String [i/to/i] | ensemble member numbers |
| EXPVER | 1 | Integer | experiment number |
| FORMAT | 'GRIB1' | String | output format of GRIB fields |
| **Data fields section** | | | |
| TYPE | None | list of strings [xx xx … xx] | list of field type per TIME |
| TIME | None | list of strings [xx xx … xx] | list of times |
| STEP | None | list of strings [xx xx … xx] | list of forecast steps corresponding to TIME |
| MAXSTEP | None | Integer | maximum forecast step |
| **Flux data fields section** | | | |
| ACCTYPE | None | String | type of the flux forecast fields |
| ACCTIME | None | String [i/i] | forecast times of flux fields |
| ACCMAXSTEP | None | Integer | maximum forecast step of flux fields |
| RRINT | 0 | Integer | switch to select method for precipitation disaggregation |
| **Domain section** | | | |
| GRID | None | String [i/i] | horizontal resolution on longitude/latitude grid |
| RESOL | None | String | horizontal resolution of spectral grid |
| SMOOTH | 0 | Integer | spectral truncation of $\dot{\eta}$ on Gaussian grid |
| LEFT | None | String | longitude of lower left domain corner |
| LOWER | None | String | latitude of lower left domain corner |
| UPPER | None | String | latitude of upper right domain corner |
| RIGHT | None | String | longitude of upper right domain corner |
| LEVEL | None | Integer | maximum number of vertical levels |
| LEVELIST | None | String [start/to/end] | definition of vertical levels |
| **Vertical velocity section** | | | |
| GAUSS | 0 | Integer | switch to calculate $\dot{\eta}$ |
| ACCURACY | 24 | Integer | number of bits per value in GRIB coded fields |
| OMEGA | 0 | Integer | switch to retrieve $\omega$ from MARS |
| OMEGADIFF | 0 | Integer | switch to calculate $\omega$ and $Dp_s/Dt$ from continuity equation |
| ETA | 0 | Integer | switch to read $\dot{\eta}$ from MARS |
| ETADIFF | 0 | Integer | switch to calculate $\dot{\eta}$ and $Dp_s/Dt$ from continuity equation |
| DPDETA | 1 | Integer | switch to select multiplication of $\dot{\eta}$ by $dp/d\eta$ |
| ETAPAR | 77 | Integer | GRIB parameter id for $\dot{\eta}/dp/d\eta$ |





**Table 9.** Part 2 of the overview of `CONTROL` file parameters.

| Parameter | Default value | Format | Description |
|---|---|---|---|
| **General section** | | | |
| DEBUG | 0 | Integer | switch to save the temporary files |
| REQUEST | 0 | Integer | switch to create the file `mars_requests.csv` |
| PUBLIC | 0 | Integer | switch to select public WebAPI access |
| OPER | 0 | Integer | switch to prepare operation job script |
| ECSTORAGE | 0 | Integer | switch to store results in ECFS file system |
| ECTRANS | 0 | Integer | switch to transfer final files to local system |
| PREFIX | 'EN' | String | front string in file names before the date string |
| ECFSDIR | 'ectmp:/$USER/ econdemand/' | String | destination directory on ECFS file system |
| MAILFAIL | ['$USER'] | List of strings | list of emails to send log files to |
| MAILOPS | ['$USER'] | List of strings | list of emails to send log files to |
| **Additional data section** | | | |
| CWC | 0 | Integer | switch to retrieve total cloud water content |
| DOUBLEELDA | 0 | Integer | switch to manually double ensemble member number |
| ADDPAR | None | String [p1/p2/.../pn] | additional surface fields to retrieve |

**Table 10.** Description of the parameters stored in file `ECMWF_ENV`.

| Parameter | Default value | Format | Description |
|---|---|---|---|
| ECUID | None | String | ECMWF user id |
| ECGID | None | String | ECMWF group id |
| DESTINATION | None | String | ectrans association |
| GATEWAY | None | String | name or ip address of member gateway server |

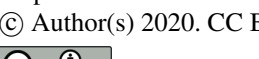



**Table 11.** Overview of templates used in flex_extract. They are stored in the `Templates` directory.

| Template | Description |
|----------|-------------|
| `convert.nl` | Used to create a Fortran namelist file called `fort.4`. It will be created in the Python part and contains controlling options for `calc_etadot`. (See Table 15) |
| `ecmwf_env.template` | Used to create the `ECMWF_ENV` file within application modes *gateway* and *remote*. |
| `compilejob.template` | Used to create the file `compilejob.ksh` during the installation process for the application modes *remote* and *gateway*. |
| `job.temp` | Used to create the actual job script file called `job.ksh` for the execution of flex_extract in the application modes *remote* and *gateway*. |
| `job.template` | Used to create the template `job.temp` in the installation process. A couple of parameters are set, such as the user credentials and the flex_extract version number. |

**Table 12.** Overview of parameters to be set in the `setup.sh` script for installation. In case of *remote* and *local* mode for member state users, the file `ECMWF_ENV` will be created, hence the parameters from Table 10 must also be set in the `setup.sh` script.

| Parameter | Default value | Format | Description |
|-----------|---------------|--------|-------------|
| `TARGET` | None | String | defines location and therefore the application mode |
| `MAKEFILE` | Makefile.gfortran | String | Makefile for compiling `calc_etadot` |
| `JOB_TEMPLATE` | job.template | String | batch job template for *gateway* and *remote* mode |
| `INSTALLDIR` | `$HOME` on ECMWF servers; `pwd` in *local* mode | String | root path for flex_extract working directory |
| `CONTROLFILE` | `CONTROL_ERA5` | String | input file with parameter settings |





**Table 13.** Overview of the parameter to be set in the `run*.sh` script. In order to provide a complete list, some already defined parameters from Tables 8, 9 and 12 are repeated here. In the case of a special format, a sample format is given in parentheses; f denotes a floating-point number.

| Parameter | Default value | Format | Description |
|---|---|---|---|
| START_DATE | None | String (YYYYMMDD) | first day of retrieval period |
| END_DATE | None | String (YYYYMMDD) | last day of retrieval period |
| DATE_CHUNK | 3 | Integer | number of days within one mars request |
| BASETIME | None | Integer | end time for half-day retrievals |
| STEP | None | blank seperated list of numbers | list of forecast steps of corresponding retrieval times |
| LEVELLIST | None | String [start/to/end] | defines list of vertical levels |
| JOB_CHUNK | None | Integer | number of days to be retrieved within a single job |
| AREA | - | String (f/f/f/f) | domain defined as north/west/south/east |
| PUBLIC | 0 | Integer | set to 1 for using public access mode |
| INPUTDIR | None | String | path to temporary working directory |
| OUTPUTDIR | None | String | path where final output files are stored |
| PPID | None | Integer | parent process id of the job (only for debugging) |
| JOB_TEMPLATE | job.temp | String | job template file for ECMWF batch queue |
| QUEUE | None | String | in case of non-local mode, the ECMWF server name |
| CONTROLFILE | CONTROL_ERA5 | String | input file with parameter settings |
| RRINT | 0 | Integer | set to 1 to select new method for precipitation dis-aggregation |
| REQUEST | 0 | Integer | set to 1 to create the file `mars_requests.csv` |
| OPER | 0 | Integer | set to 1 for operational mode (job script) |
| DEBUG | 0 | Integer | set to 1 to save the temporary files |

**Table 14.** List of flux fields retrieved by flex_extract and the disaggregation scheme applied.

| Short name | Name | Unit | Interpolation |
|---|---|---|---|
| LSP | large-scale precipitation | m | linear |
| CP | convective precipitation | m | linear |
| SSHF | surface sensible heat flux | $Jm^{-2}$ | bicubic |
| EWSS | eastward turbulent surface stress | $Nm^{-2}s$ | bicubic |
| NSSS | northward turbulent surface stress | $Nm^{-2}s$ | bicubic |
| SSR | surface net solar radiation | $Jm^{-2}$ | bicubic |



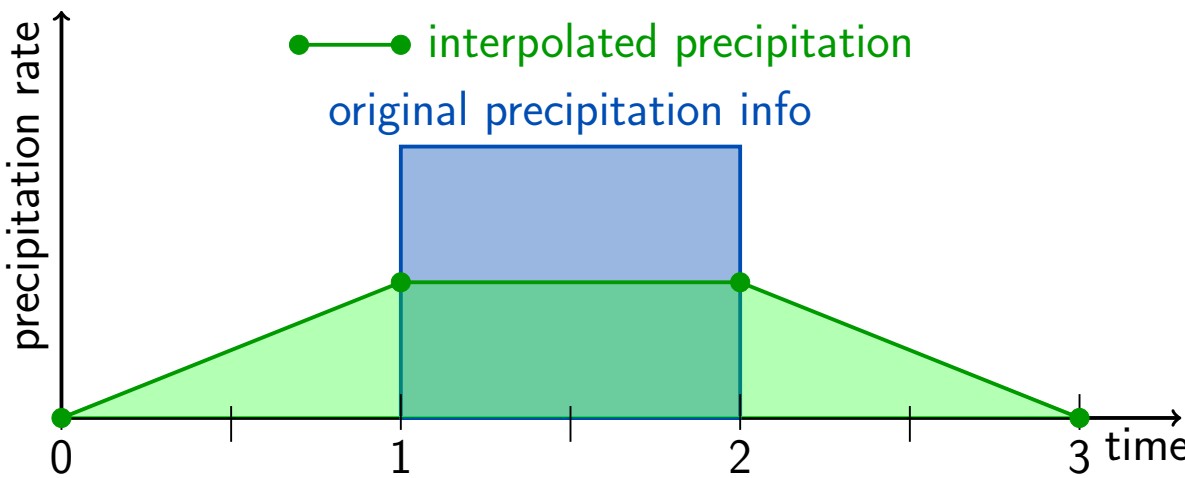

**Figure 5.** Example of disaggregation scheme as implemented in older versions of flex_extract for an isolated precipitation event lasting one time interval (thick blue line). The amount of original precipitation after de-accumulation is given by the blue-shaded area. The green circles represent the discrete grid points after disaggregation. FLEXPART interpolates linearly between them as indicated by the green line and the green-shaded area. Note that supporting points for the interpolation are shifted by half a time interval compared to the other meteorological fields. From Hittmeir et al. (2018).

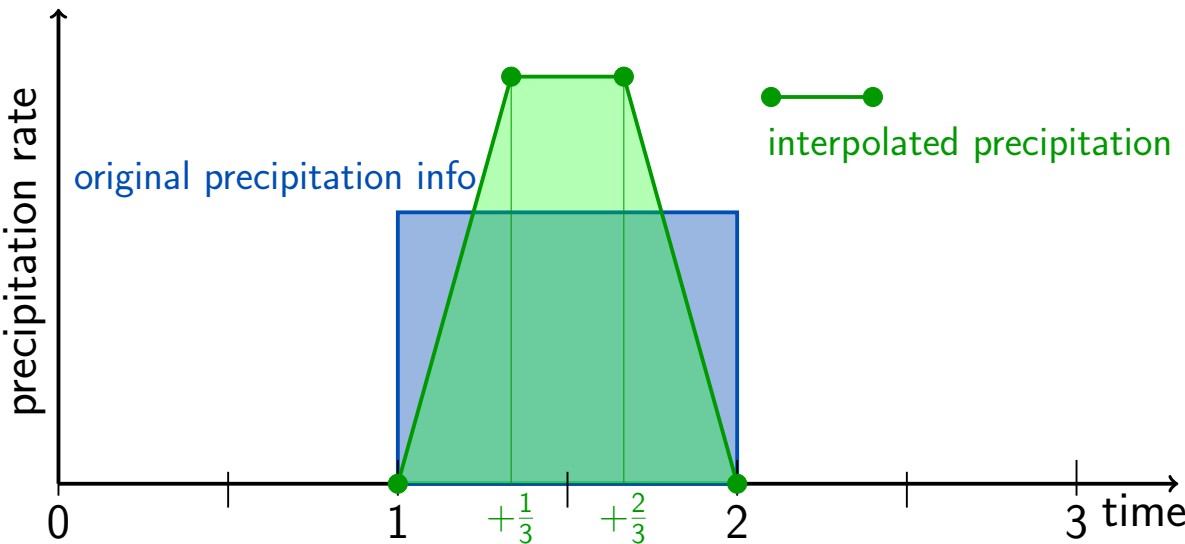

**Figure 6.** As Figure 5, but with the new interpolation scheme using additional sub-grid points. From Hittmeir et al. (2018).





**Table 15.** Overview of options controlling `calc_etadot`. Note that the resolution of the latitude-longitude grid is given implicitly by the grid dimensions and extent.

| Parameter | Description | Remarks |
|---|---|---|
| `maxl` | grid dimension – longitudes | |
| `maxb` | grid dimension – latitudes | |
| `mlevel` | grid dimension – number of levels | |
| `mlevelist` | list of levels | to be given in MARS request notation like `1/to/91` |
| `mnauf` | number of spectral coefficients in input data | |
| `metapar` | GRIB ID of vertical velocity in output | standard FLEXPART expects `=77` |
| `rlo0` | Western border of domain in degree | |
| `rlo1` | Eastern border of domain in degree | |
| `rla0` | Southern border of domain in degree | |
| `rla1` | Northern border of domain in degree | |
| `momega` | if 1, $\omega$ is calculated from $\dot{\eta}$ and output | for testing the accuracy of calculated $\dot{\eta}$ if no $\dot{\eta}$ from MARS is available |
| `momegadiff` | if 1, calculated $\omega$ is compared with $\omega$ from MARS | |
| `mgauss` | if 1, evaluate continuity equation on GG | |
| `msmooth` | if $\neq 0$, apply spectral smoothing by clipping at given truncation | |
| `meta` | if 1, use $\dot{\eta}$ from input. | |
| `metadiff` | if 1 and `meta=0`, $\dot{\eta}$ needs to be available from MARS and this is compared with calculated $\dot{\eta}$ | for testing the accuracy of $\dot{\eta}$ calculation |
| `mdpdeta` | if 1, give $\dot{\eta}_p$ as output | with the current version of FLEXPART, only `=1` is useful; future versions might used $\dot{\eta}$. |





**Table 16.** Determination of the method for obtaining $\dot{\eta}$ in `calc_etadot` as a funtion of control parameters (see also Table 15). GG stands for Gaussian grid. The names of the corresponding regression tests (see Section 5.4) are also given

| Method | mgauss | meta | Test name |
|---|---|---|---|
| Continuity eq. on lat-lon grid | 0 | 0 | `latlon` |
| Continuity eq. on GG | 1 | 0 | `gauss` |
| Use $\dot{\eta}$ from input | 0 | 1 | `etadot` |
| (Program will stop with ERROR) | 1 | 1 | – |

**Table 17.** List of `fort` files generated by the Python part to serve as input for the Fortran program, and the output file of `calc_etadot`. If the optional fields were not extracted, the corresponding files are empty.

| Number | Content |
|---|---|
| **Input to the Fortran program `calc_etadot`** | |
| 10 | U and V wind components |
| 11 | temperature |
| 12 | logarithm of surface pressure |
| 13 | divergence (optional) |
| 16 | surface fields |
| 17 | specific humidity |
| 18 | surface specific humidity (reduced Gaussian) |
| 19 | vertical velocity (pressure) (optional) |
| 21 | eta-coordinate vertical velocity (optional) |
| 22 | total cloud water content (optional) |
| **Output from Fortran program `calc_etadot`** | |
| 15 | U and V wind components, $\dot{\eta}$, temperature, surface pressure, specific humidity |



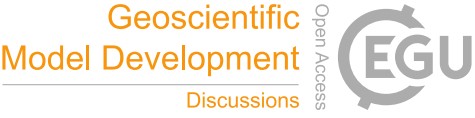

**Table 18.** List of model level parameters FLEXPART requires to run and the availability in the different data sets (ECMWF, 2019e, i). The cloud-water content fields are optional. The divergence and logarithm or surface pressure fields are only necessary for the calculation of the vertical velocity when $\dot{\eta}$ is not available directly. These fields are not transferred to the FLEXPART input files. FC stands for "forecast" and AN for "analysis".

| Variables | Short name | Parameter ID | Unit | Operational | | ERA-Interim | | ERA5 | | CERA-20C | |
|---|---|---|---|---|---|---|---|---|---|---|---|
| | | | | FC | AN | FC | AN | FC | AN | FC | AN |
| Temperature | T | 130 | K | x | x | x | x | x | x | x | x |
| Specific humidity | Q | 133 | $kgkg^{-1}$ | x | x | x | x | x | x | x | x |
| U – wind component | U | 131 | $ms^{-1}$ | x | x | x | x | x | x | x | x |
| V – wind component | V | 132 | $ms^{-1}$ | x | x | x | x | x | x | x | x |
| Eta-coordinate vertical velocity | etadot | 77 | $s^{-1}$ | $x^2$ | $x^2$ | - | - | x | x | x | x |
| Divergence | D | 155 | $kgm^{-2}$ | x | x | x | x | x | x | x | x |
| Specific cloud liquid water content | clwc | 246 | $kgkg^{-1}$ | x | x | x | x | x | x | x | x |
| Specific cloud ice water content | ciwc | 247 | $kgkg^{-1}$ | x | x | x | x | x | x | x | x |
| Logarithm of surface pressure[1] | lnsp | 152 | - | (x) | (x) | (x) | (x) | (x) | (x) | (x) | (x) |

[1] Only available on model level 1.

[2] Available from 4 June 2008 onward.





**Table 19.** List of surface level parameters FLEXPART requires to run and the availability in the different data sets (ECMWF, 2019e, i). FC stands for "forecast" and AN for "analysis". Deviating versions of FLEXPART or pre/post-processing software may require additional surface level fields which are not listed here.

| Variables | Short name | Parameter ID | Unit | Operational | | ERA-Interim | | ERA5 | | CERA-20C | |
|---|---|---|---|---|---|---|---|---|---|---|---|
| | | | | FC | AN | FC | AN | FC | AN | FC | AN |
| 2 metre temperature | 2t | 167 | K | x | x | x | x | x | x | x | x |
| 2 metre dewpoint temperature | 2d | 168 | K | x | x | x | x | x | x | x | x |
| 10 metre U wind component | 10u | 165 | $ms^{-1}$ | x | x | x | x | x | x | x | x |
| 10 metre V wind component | 10v | 166 | $ms^{-1}$ | x | x | x | x | x | x | x | x |
| Geopotential | z | 129 | $m^{-2}s^{-2}$ | x | x | x | x | x | x | x | x |
| Land-Sea Mask | lsm | 172 | $0-1$ | x | x | x | x | x | x | x | x |
| Mean sea level pressure | msl | 151 | Pa | x | x | x | x | x | x | x | x |
| Snow depth | sd | 141 | m of w. eq. | x | x | x | x | x | x | x | x |
| Standard deviation of orography | sdor | 160 | - | - | x | - | x | - | x | - | x |
| Surface pressure | sp | 134 | Pa | - | x | x | x | - | x | x | x |
| Total cloud cover | tcc | 164 | $0-1$ | x | x | x | x | x | x | x | x |
| Convetice precipitation | cp | 143 | m | x | - | x | - | x | - | x | - |
| Large-scale precipitation | lsp | 142 | m | x | - | x | - | x | - | x | - |
| Surface sensible heat flux | sshf | 146 | $Jm^{-2}$ | x | - | x | - | x | - | x | - |
| Eastward turbulent surface stress | ewss | 180 | $Nm^{-2}s$ | x | - | x | - | x | - | x | - |
| Northward turbulent surface stress | nsss | 181 | $Nm^{-2}s$ | x | - | x | - | x | - | x | - |
| Surface net solar radiation | ssr | 176 | $Jm^{-2}$ | x | - | x | - | x | - | x | - |
| Forecast surface roughness[1] | fsr | 244 | m | x | - | - | - | x | x | x | x |

[1]Necessary in CERA-20C due to missing surface roughness parameter





**Table 20.** List of generic `CONTROL` files coming with the flex_extract version 7.1 release. Each file name contains information about some key aspects of the data set to be retrieved. The first name component is an abbreviation of the data set name (`OD`, `EA5`, `CERA`, `EI`) and whether the domain is global (if not, no name component). Reanalysis data sets are divided into *public* and non-public retrievals. File names for operational data sets also contain information about the stream, the forecast type and the method of deriving the vertical velocity. Further information is optional and mostly indicates time resolution or period to be retrieved or whether a specific `CONTROL` parameter was used.

```
CONTROL_CERA
CONTROL_CERA.global
CONTROL_CERA.public
CONTROL_EA5
CONTROL_EA5.global
CONTROL_EI
CONTROL_EI.global
CONTROL_EI.public
CONTROL_OD.ELDA.FC.eta.ens.double
CONTROL_OD.ENFO.CF.36hours
CONTROL_OD.ENFO.CV.36hours
CONTROL_OD.ENFO.PF.36hours
CONTROL_OD.ENFO.PF.ens
CONTROL_OD.OPER.4V.eta.global
CONTROL_OD.OPER.FC.36hours
CONTROL_OD.OPER.FC.eta.basetime
CONTROL_OD.OPER.FC.eta.global
CONTROL_OD.OPER.FC.eta.highres
CONTROL_OD.OPER.FC.gauss.global
CONTROL_OD.OPER.FC.gauss.highres
CONTROL_OD.OPER.FC.operational
CONTROL_OD.OPER.FC.twiceaday.1hourly
CONTROL_OD.OPER.FC.twiceaday.3hourly
CONTROL_OD.temporary
```

**Table 21.** Basic metrics.

| Version | LOC | SLOC | Comments | Multi | Blank |
|---------|------|------|----------|-------|-------|
| 7.0.4 | 2538 | 1820 | 346 | 13 | 374 |
| 7.1 | 7543 | 2842 | 1072 | 2265 | 1397 |





**Table 22.** Ranks of cyclomatic complexity (CC) taken from the manual of the Python package `radon` (Lacchia, 2019).

| CC score | Rank | Risk |
|---|---|---|
| 1 - 5 | A | low – simple block |
| 6 - 10 | B | low – well structured and stable block |
| 11 - 20 | C | moderate – slightly complex block |
| 21 - 30 | D | more than moderate – more complex block |
| 31 - 40 | E | high – complex block, alarming |
| 41+ | F | very high – error-prone, unstable block |

**Table 23.** Number of code blocks (classes, methods, functions) with a specific rank of cyclomatic complexity and the percentage of the total blocks for version 7.1 (116 in total) and 7.0.4 (45 in total). Determined with the Python package `radon` (Lacchia, 2019).

| Rank | Version 7.0.4 | | Version 7.1 | |
|---|---|---|---|---|
| A | 21 | 44.6 % | 76 | 65.52 % |
| B | 10 | 22.2 % | 28 | 24.14 % |
| C | 3 | 6.6 % | 9 | 7.76 % |
| D | 4 | 8.8 % | 1 | 0.86 % |
| E | 3 | 6.6 % | 1 | 0.86 % |
| F | 4 | 8.8 % | 1 | 0.86 % |





**Table 24.** Python code blocks with CC ranks C-F, with rank class and CC score. The block types are classes (C), class methods (M) and functions (F)

(a) Version 7.0.4

| Block | Block type | Rank | CC score |
|---|---|---|---|
| **Class methods** | | | |
| GribTools.setkeys | M | C | 11 |
| MARSretrieval.dataRetrieve | M | C | 15 |
| Control | C | D | 23 |
| EIFlexpart.process_output | M | D | 26 |
| EIFlexpart | C | E | 31 |
| EIFlexpart.deacc_fluxes | M | E | 34 |
| EIFlexpart.retrieve | M | F | 43 |
| EIFlexpart.__init__ | M | F | 49 |
| Control.__init__ | M | F | 56 |
| EIFlexpart.create | M | F | 57 |
| **Module functions** | | | |
| install_args_and_control | F | C | 12 |
| getMARSdata | F | D | 25 |
| install_via_gateway | F | D | 30 |
| interpret_args_and_control | F | E | 34 |

(b) Version 7.1

| Block | Block type | Rank | CC score |
|---|---|---|---|
| **Class methods** | | | |
| EcFlexpart | C | C | 13 |
| EcFlexpart._create_params | M | C | 13 |
| EcFlexpart._prep_new_rrint | M | C | 14 |
| EcFlexpart._create_field_types | M | C | 15 |
| MarsRetrieval.data_retrieve | M | C | 16 |
| ControlFile._read_controlfile | M | C | 17 |
| EcFlexpart.retrieve | M | D | 25 |
| EcFlexpart.create | M | E | 36 |
| EcFlexpart.deacc_fluxes | M | F | 57 |
| **Module functions** | | | |
| install.py::check_install_conditions | F | C | 11 |
| install.py::mk_tarball | F | C | 17 |
| disaggregation.py::IA3 | F | C | 18 |





**Table 25.** Definition of maintainability ranks. This classification was taken from the documentation of the Python package `radon` (Lacchia, 2019).

| MI score | Rank | Maintainability |
|---|---|---|
| 20 – 100 | A | Very high |
| 10 – 19 | B | Medium |
| 0 – 9 | C | Extremely low |





**Table 26.** Maintainability index in increasing order for the Python files of both versions. This was determined with the Python package `radon` (Lacchia, 2019).

Version 7.0.4

| File | Rank | MI score |
|------|------|----------|
| Classes/EcFlexpart.py | B | 10.79 |
| Classes/MarsRetrieval.py | A | 26.92 |
| Mods/checks.py | A | 26.15 |
| Classes/MarsRetrieval.py | A | 26.92 |
| Mods/disaggregation.py | A | 28.55 |
| Mods/profiling.py | A | 38.10 |
| Mods/tools.py | A | 38.32 |
| Mods/get_mars_data.py | A | 44.77 |
| install.py | A | 47.33 |
| Mods/prepare_flexpart.py | A | 47.47 |
| Classes/GribUtil.py | A | 57.07 |
| submit.py | A | 58.90 |
| _config.py | A | 77.35 |
| Classes/UioFiles.py | A | 100.00 |

Version 7.1

| File | Rank | MI score |
|------|------|----------|
| FlexpartTools.py | C | 0.00 |
| opposite.py | A | 45.25 |
| install.py | A | 48.96 |
| getMARSdata.py | A | 56.28 |
| GribTools.py | A | 59.10 |
| submit.py | A | 67.72 |
| prepareFLEXPART.py | A | 71.40 |
| UIOTools.py | A | 85.18 |