# Peer review of "Flex\_extract v7.1.2 – A software package to retrieve and prepare ECMWF data for use in FLEXPART"

_Geoscientific Model Development, 2019_

## Referee Comment (RC1) · Anonymous Referee #1 · 19 May 2020

The manuscript 'Flex_extract v7.1 - A software to retrieve and prepare ECMWF data for use in FLEXPART' describes a tool used for the pre-processing of data from various ECMWF model streams, preparing them for the use with the atmospheric transport model FLEXPART. The tool builds on a long history going back to a loose script collection that was previously used for FLEPXART data retrieval. The manuscript and the software are a great opportunity to bring to light some of the somewhat obscure pre-processing steps that were applied by many groups using FLEXPART and were based on unpublished descriptions of the pre-processing. However, the manuscript could be significantly improved in terms of clarity and detail. Currently, it is quite difficult to use it as a user's manual. Detailed comments are provided below. Next to reviewing the manuscript I followed the installation and use instructions and took some peeks at the

underlying code. Some comments concerning this are provided below. However, these experiences/tests are far from complete and I feel further feedback by other test users would be very beneficial. In summary, I suggest major revisions to the manuscript (as outlined below) and suggest that the authors also consider comments concerning the code before this work can be published in GMD.

Major Comments

1) Type of manuscript The manuscript was submitted as a 'model description paper'. However, it does not describe a geoscientific model as such, but a pre-processor for the FLEXPART transport model. As such I strongly feel it should rather be in the category 'development and technical papers'.

2) Paper structure

Overall the manuscript is too long. Considering that this is 'only' a pre-processor and only few scientific methods are described, it should be possible to present the material in a more concise way, focussing more on the application side of things.

First of all, the manuscript should more clearly distinguish between methods/scientific background and the application/user's manual section in the appendix. Right now there seems to be too much of a mix between the two, especially in section 3. Here are my suggestions for improving the flow of the main manuscript. Section 1: Start with description of FLEXPART/FLEXTRA and the need for appropriate input data. Then the history of flex_extract. Wherever possible move historical information from sections below into this section. Section 2: This section is very valuable, especially the provided Tables summarising the ECMWF data availability, which is cumbersome to obtain from ECMWF/MARS itself. Section 3: Especially convoluted part between methods and application. Large parts of section 3 especially sub-sections 3.4 and 3.5.x are focussing only on application and should be moved to appendix. Section 4: Very valuable, but could have clear description of settings for recommended best practices. Section 5: Although, documenting, testing and benchmarking of a code are as important as the code

itself, I suggest to significantly shorten this section and put the details into supplementary material. I can see that a lot of work went into this part, but for most readers/users this information is less relevant and of no direct consequence. Appendix: This part should work as a user's manual. A such I think the structure would work better if those parts that differ between application modes would be organised by application mode and not by different installation/application steps. This would allow a user, who will usually be running the software in only one of the modes, to follow along without having to jump from section to section.

3) Code structure and maintenance

As stated by the authors themselves, flex_extract is a collection of shell scripts and python scripts and FORTRAN code. As such there seem to be many possibilities to break the code in case external dependencies change. I am wondering if it would not be a much cleaner approach to provide this tool as a single python package. This should make maintenance, documentation and installation a lot easier. Python codes are able to work with external FORTRAN codes as well. So it would still be possible to make use of the arguably faster computations done in the FORTRAN code. But there seems to be little need for mixing in korn shell scripts (other than for job execution on ECMWF servers). Some interesting information in this respect can be found here: https://packaging.python.org/tutorials/packaging-projects/ https://www.numfys.net/howto/F2PY/ This is more a comment about the future of the code and may not be achieved quickly as part of a revision of this submission. It would also be beneficial to host the code on a more commonly used platform like github or gitlab. I was not able to git clone the code from the address given in the manuscript and would not know how to provide code feedbacks in form of merge/feature requests on the platform that is currently used for hosting the code.

Minor Comments

P2,L39: 'non-authorised' is a bit strange. Users still need to be registered and as such

are authorised to use the system. I would rather call this non-member state users.

Version number, title and first mention P2,L44: If a totally revised version was created that is fundamentally different from the previous versions and as such not compatible with, for example, previous config files, it may be more appropriate to express this by incrementing the major version number. So why not go for 8.0?

P3,L70f: New version of what? I guess flex_extract, but could also refer to FLEXPART. There is also no clear mentioning above about how the first version looked like.

P3,L85: Also mention what happened to the FORTRAN part.

P4,L100: 'most accurate data source'. That's a bit general. So ECMWF is better than any other atmospheric model? I would not go that far, but if you do you should link this comment to some kind of model evaluation that provides a similar conclusion.

P4,L103: Last sentence can be removed, should be clear from context.

P4,L108: I would not say that they share the code, since these are two completely different software packages. But one could say that some of the code goes back to the same original routines. Sharing the code would mean that their would be a common specific library of routines that would be used by both models.

P4,L108: 'It ingests ...', It should be referring to FLEXTRA, but previous sentence on FLEXPART. Replace.

P4,L131: 'are to be used' better 'are available' or 'can be used'.

P4,L131: 'Flex_extract automatically ...': Not clear how flex_extract comes in at this point. So for this section was describing the different access kinds to ecmwf. It needs to be made clear here what are the access possibilities and then in a second step how flex_extract is using them. Maybe don't mention Flex_extrac in this section at all.

P5,L141: Later on this dataset/model version is referred to as DET-FC. Why not keep the ECMWF terminology.

P6,L157ff and Table 1: While the text nicely refers to the evolution in the IFS operational setup, I am missing a similar warning in Table 1. Seen on its own the Table may be misleading. Please add reference to Tables 2 and 3.

P6,L177: 'member-state' instead of 'member'.

P6,L178: 'not good'. Can you more specific what is not good about? Not robust/reliable or simply (too) slow?

P7,L184ff: The use of access and applications modes is a bit confusing here. Especially when the text refers first to access and application and then the combination of these is once more called application mode. Maybe the primary application modes could be described more clearly as 'execution location'?

P7,L213: 'module system': Rather call it 'environment modules framework'.

Figure 1: Could this not be drawn much simpler without the emphasized distinction between the two different web APIs? The main point of this schematic should be an illustration of the 4 application modes. To user it does not matter so much that there are two different web APIs.

P8,L231: Why not mention this python package in Table 7 as well?

P9,L254f: In the schematic (Figure 4) the "remote mode" is on top. So why start with explaining the "gateway mode" first?

Figure 4: The schematic may be easier to comprehend if ECMWF and local machines would always be on one side. Also the Labels for the different modes should be given rather on top with the sub-panel index. Right now the may be mistaken as only referring to the boxes on the right.

P9,L256: Once again the use of he different modes is confusing. While describing the 'gateway mode' it is mentioned that flex_extract is started in 'local mode'? Confusing. If I understand correctly, it is started locally to generate the ECMWF scripts and then

sends them through the gateway. 'local mode', however, is not using the gateway but the WebAPIs. Correct? The same confusion is created in the caption to Figure 2, where 'local mode' is mixed into both gateway and remote mode.

Figure 2: What is the purpose of the 'do local work' branch of the flow chart?

Figure 3: Again, I am not sure that I understand the purpose of the two side-branches 'submit' and 'PRINTING MARS_REQUEST'.

Section 3.4.1 and Tables 8 and 9: I feel that these tables include the core information of how to setup and run the data extraction correctly. As such I think the description of the individual parameters should be given more space in the main text (as part of the Appendix) and not only in the supplement. This is crucial information for using flex_extract!

P14,L340: 'via the gateway server': not in 'remote mode'!

P14,L351: Do these modules correspond to the according boxes in the flowchart (Fig3)?

P15,L363: Who is Paul James. Is there a reference to this work?

P16,L390ff: At this stage it is not clear how these additional inputs are treated. Will there be separate output files for these times? How does FLEXPART deal with these? From which FLEXPART version onwards is this feature supported?

P16,L394: Maybe it would be useful to explain briefly why other fluxes are treated differently from precipitation.

P18,L453: grib_api seems to removed from all the makefiles and is also not mentioned in the dependencies (Table 7).

P18,L456: Usually one would refer to these versions as optimised and debug versions, not fast and debug. It would also be better to have a single makefile and only use a switch in the makefile to create either of the desired executables.

P19,L483: Does the comment in braces mean, that no two instances of the submit.py script should ever be running at the same time? Where do you expect clashes if done anyway.

P20,L530: What about forecasts longer than 99 hours. How will they be written into a 2 digit number?

P22,L577: This is comment that cannot be fixed in the manuscript, but should be addressed in the future. I assume you are referring to the Global Forecast System of NCEP here? If the availability of model level data is a shortcoming to the community. NCEP should be approached to make model levels available as well. Internally GFS operates on a similar hybrid sigma-pressure coordinate system as ECMWF IFS.

Section 4: The section gives many valuable hints for best practices. However, I would find it very useful if these could be manifested in more concrete instructions, maybe in form of clear example CONTROL files. For example it is mentioned that only a subset of levels can be extracted, but how is this done concretely and does it work for all etadot modes (didn't in the past)?

P28,L746f: Actually one does not need any code for this task. It can be performed using standard bash cat command. That is one of the few beauties of grib format.

P28,L752ff: See also previous comment on package structure. The possibility of calculating etadot should not just be discarded, but there certainly are better ways of integrating the FORTRAN code into the python package.

P28,L755f: Again as mentioned above, instead of just providing a tar ball, a public git repository that would allow for swift user feedbacks and contributions would be beneficial.

P29,L790: Better refer to this as dependencies or prerequisites.

Code sections in Appendix (e.g., P31,L850ff): It would be easier if such sections would be directly given as example config files or scripts. Copy and paste from a manuscript

is rather cumbersome and error-prone.

P34,L994: Does testing CERA-20C refer to CONTROL_CERA or is there another prepared example to test it. Please mention.

P35,L103: This unload was not mentioned above for the installation process. Is it necessary? Because once you unload python on ecgate the python command is linked to the system pyhton version 2.6! So I don't see the gain of the unload.

Table 7: It is mentioned in the text that compilers other than GNU compilers were tested as well. This should be reflected in the table. emoslib should just be emos. The information for fortan/pyhton is not required since this is given by the header line.

Table 11: With files job.temp and job.template it is difficult to distinguish their function just from file name. The second one should maybe be renamed to install.job.template Why does the first template not have the template file name extension. Is it treated in a different way than the other templates? Same for convert.nl. Even if the output of the template processing is a namelist shouldn't it have the same template ending.

Table 20: Is this table really needed? It is only a list of file names without further description. I can obtain this also by looking into the corresponding folder! Either remove or add a meaningful description of the listed files. The caption alone does not explain the details.

Technical Comments on Manuscript

P3,L83/84: Past tense in previous sentence. Why not here?

P4,L106: Style. Twice representing in same sentence.

P5,L146: Citation style for Laloyaux et al. (2018)

P6,L181: 'necessary' not 'ecessary'

P16,L387: Citation format of '(Hittmeir et al., 2018, p. 2513)'. No pages in journal

reference.

P17,L414f: So in the end FLEXPART converts etadot (or etapoint) back to a Cartesian vertical velocity? So why the painful calculation of etadot in the first place? Only because of the historic development from FLEXTRA?

P17,L416: 'pressure' not 'perssure'

Code Comments

As stated above these are some comments on code and on my personal experience running the code in 'remote mode' on ECWMF's ecgate server. Some of the remarks document that the ease of using the code should still be improved before final publication of this manuscript. Other comments can be seen as recommendations for the future.

The FORTRAN code works with default FORTAN unit names for I/O (see table 17). This seems a bit outdated and difficult to follow/debug/etc. Please change to meaningful filenames wherever possible. This would allow removing Table 17 from the manuscript.

Besides the documentation in the main routine the comments in the FORTRAN code are mostly in German. This should be changed in future submissions.

Section 3.8.3: And why would the FORTRAN program need to save these variables to a file instead of storing them internally? Makes little sense if they are not used again, for example, by the python scripts.!

Installation process: I only tested the remote mode on ecgate. My experience was not as straightforward as the description may suggest and as one would expect on a system the code should have been tested on. Since the remote installation is the one installation mode that is running on a defined system it should be easier to get this done without having to change various parameters in the setup script. First of all, I tried the git clone command on page 28. However, this does not clone flex_extract, but FLEXPART!? In the following I could not figure out the correct git address for

flex_extract and resigned to to downloading the tar ball instead. Once unpacked, I was able to proceed. However, I don't see any reason why the installation process is submitted to the queueing system on ecgate. We are talking about a very small compile task here. It would be more simply done on the login nodes so that one has a more direct feedback when things go wrong. Anyway, first email I get back from the submitted install job was a very general compile error ("No rule to make target 'phgrreal.f', needed by 'phgrreal.o'. Stop"). So I checked if I selected the correct makefile. Yes, one would think: makefile_ecgate (the one system where this should work. Right?). Taking a look into makefile_ecgate shows that file extensions seem to have changed from f to f90, but this was not adjusted in the makefile. So I modified the makefile. Second try. Got an email which may indicate success of the FORTRAN compile, but does not tell me anything else about the rest of the installation. So let's see if it is there. Hm. Not in the directory I had specified in $INSTALLDIR. But there is a new folder flext_extract_v7.1 in my home. OK. Fine. Let's test it. So I follow the instructions in Appendix A4. However, the executable mentioned there is called calc_etadot_fast.out, while in my installation it is simply called calc_etadot. No problem. But shouldn't the exe name be harmonised in the makefiles? I did not change it. Running the executable seems to work but in my case only produced 2 lines of output in contrast to 7 mentioned in Appendix A4. Once again, did I do something wrong. Did it really work? There are 2 output files in the directory now and one can assume that that's sufficient. grib_ls on fort.15 (the obvious output file) at least shows some etadot fields. However, I was still worried about the missing lines of output. So I quickly searched the FORTRAN code for where these messages should be produced. I found them but they are commented out. So no wonder that I did not get these messages! But why are they given in the instructions.

Test retrieval: A test retrieval using the CONTROL_CERA example worked without much trouble. I would only suggest that the loading of the python3 module should be added to the run.sh script.

EcFlexpart.py: As far as I can see this is where the ectrans command is used to transfer

the data. The ectrans command is used in a way that assumes that it would crash if an invalid transfer was submitted. But this is not how ectrans works. Much like a queuing system ectrans submits the requested file transfer into a queue returning a request ID. The transfer request will then be repeated if not successful at the first try. So if the settings for source and target file are not correct ectrans will still return a request ID and try the transfer until it reaches its repeat limit. Hence, it is not possible to simply capture the return code from ectrans to detect an error.

Work folders on $SCRATCH are called python<PID>. Why not use something that would clearly identify these as flex_extract folders?

---

## Referee Comment (RC2) · Anonymous Referee #2 · 21 May 2020

The manuscript "Flex_extract v7.1 – A software to retrieve and prepare ECMWF data for use in FLEXPART" presents an exhaustive introduction of software written to retrieve driving variable data for the Lagrangian particle transport model, FLEXPART. This software is critical to the use of FLEXPART and has been widely used by the research community for a number of years. However, the complexity and required knowledge of the software certainly presented an obstacle for many in using FLEXPART itself.

The major refactor of the software is a welcomed contribution and will greatly facilitate the use of FLEXPART. Further, however, it seems there will be stand-alone use cases of Flex_extract given some of the features and facilities it provides for working with the ECMWF data archives.

[Figure]

The manuscript itself is extensive, and upon a cursory review it would be easy to say it is too long, and should be shortened. However, as a reference document it provides some key information and gives a good overview of the developer's decisions regarding certain choices. If the editor or other reviewers find shortening is required, it is this reviewer's opinion that Section 5 on quality assurance – though interesting – is not necessarily required. In particular the section on code metrics seems rather extensive.

I recommend publication with a few minor corrections, noted below.

line 58, 1990ies (and later), change to 1990s or '90s: MLA Handbook for Writers of Research Papers, 7th edition. This, from p. 84. "Decades are usually written out without capitalization (the nineties), but it is acceptable to express them in figures (the 1990s, the '60s). Whichever form you use, be consistent."

line 62, interSect –> intersect

line 67, and later of –> and later on

line 86, ecCodes (earlier referred to as eccodes) be consistent

line 86, would change 'was implemented' to 'was utilized'

line 98, ECMWF and IFS are already defined... and throughout, drop repeat acronym defintions

line 103, "All details can be found in the literature mentioned" is an empty sentence, suggest dropping it.

line 113, use ECMWF as it has been definged...

line 116, MARS has been defined...

line 150, choose meta data or metadata and use consistently

line 155, "The horizontal grid type refers to the way how it fields are archived in MARS" not clear...

line 169, inplied –> implied

line 181, ecessary –> necessarily

line 720, Lag rangian –> Lagrangian

————————————————————

---

## Referee Comment (RC3) · Anonymous Referee #3 · 16 Jun 2020

This manuscript describes the newest version, 7.1, of the flex_extract software. This piece of open source software is employed by a broad community of users of both FLEXPART and FLEXTRA in order to retrieve ECMWF meteorological fields required to run these models. It is very encouraging to see that in parallel to an adaptation and enhancement of functionalities of flex_extract, professional software quality assurance methods have been implemented in the most recent version of the tool. In addition to the unit and regression testing, modern code metrics have been used to assess notable improvements of the newest release with respect to the previous one, due to a reduced complexity. Cumbersome as it may seem, this is certainly the way the scientific software development needs to undertake in the times of continually growing codes and increased collaborative effort required. The tool itself is very versatile, in terms of

different mode of data accessibility, streams of input ECMWF data, including backward compatibility and anticipating future releases, grid-spacing and temporal frequency, which can be configured. I could not emphasise enough the importance of an early transition towards python3. Often, in the scientific community, we find ourselves with legacy issues trailing far too long with respect to the contemporary work environment. In case of flex_extract these issues are addressed in a timely manner and I would like to congratulate software developers on it.

I must admit that when I downloaded the manuscript for the first time and saw 62 pages, I sighed. But the manuscript is not as voluminous as the number of pages may indicate as the Authors included many diagrams and tables to support the content. The manuscript, effectively containing about one thousand lines of text, is very pleasant to read, the content and the logical flow of information keeps a reader focussed and is presented in smooth English. The paper is very well structured, informative, broader context of this work is described, some historical aspects are addressed and their connection to legacy issues explained. A new way of disaggregating precipitation fields is a noteworthy scientific development. Information on the sources of the ECMWF meteorological fields and advice on how to intertwine forecast and analysis fields as well as on the importance of maintaining consistency between a target application and horizontal and temporal resolution are very valuable, especially for for new end users. The content of Tab.1, Tab.2 and Tab.3 is a treasure. Respect to the Authors for gathering this information. Included are clear instructions to follow for software installation and usage. The tool as well as the instructions make an impression of being user-friendly. There are illustrative diagrams supporting the information described in the text.

I recommend the manuscript for publication after the following issues and typos have been addressed:

l. 5 I suggest replacing 'therefore' with 'consequently' to avoid a repetition with the previous sentence

l. 22 should be 'Integrated Forecast System'

l. 41 I believe you want ECMWF here

l. 58 should read '1990s'

l. 62 You probably want 'intersect' and not 'interSect.'

l. 99 Should read US National Centers

l. 107/108 – the two reference should be in brackets

l. 116 reference should be in brackets

l. 146 reference should be in brackets

l. 155 suggest skipping 'it' or better replacing it with 'the'

l. 169 'implied; instead of 'inplied', I believe

l. 181 missing 'n' in 'ecessary'

l. 184 - 187 and then also l. 206 – 209: are 'access mode' and 'application mode' two different things? Are 'ECMWF Member States Linux servers', 'Member State Gateway server' and 'local host' rather access modes than application modes? And based on those three access modes, four application modes are derived? Maybe it is then clearer in conjunction with the header of section 3.1, where 'Remote', 'Gateway' and 'Local' are referred to as application modes. And then in conjunction with user mode it results in four user application modes.

l. 187 results

l. 227 – depends which spelling you follow; if it is BE then 'licence'; but I noticed indeed that Copernicus Publications follow AmE spelling of this word, namely 'license'

l. 240 – I may be missing something important in the software structure but I do not understand why sending a script to the ECMWF batch queue in step 1 contrasts in

steps to and 3; in particular do not understand 'or' here; I would imagine that that job sent to the batch queue also retrieves data from MARS and post-processes them to obtain FLEXPART input fields. From l. 249-252 I understand that flex_extract proceeds with steps 2 and 3 locally; is it correct? It is a small thing but it would be good to clarify

l. 268 – you want to use 'which' once

l. 283 – should read 'for a correct setting'

l. 284/285 – you probably want to use the word 'combination' only once

l. 355/356 – these papers should be referenced in brackets

l. 363 – does it stem from private communication with Paul James?

l. 373/374 – what is the position of 1 and 2 with respect to a,b,c,d

l. 416 should read 'pressure'

l. 504 – what is a pure forecast? Do you mean deterministic forecast? 'Pure forecast'also later appears in l.527

l. 513/514 – not sure what this sentence mean? Is there just 'is' missing in this sentence? Or is there more to add?

l. 524 - is YYMMDDHH in this file name the analysis hour?

l. 529 would 'base time' be better than 'starting forecast time'

l. 535 – I may be missing something here but why the file names selected for the ensemble members do not account for FORECAST_STEP?

l. 548 – should read 'makes'

l. 581 – I would skip the coma after e.g.

l. 615 – do you mean 'paths'?

[Figure]

l. 624 – should read 'except'

l. 687 – better 'indicates a lower complexity'

l. 720 'Lagrangian' should be one word

l. 721 – 1990s

l. 940 – skip 'an'

l. 942 and 943 are too tightly formatted in vertical (also true for l. 848 and 849 and l. 888 and 889 then for lines 966, 967 and 968; subsequently lines 982, 983 and 984 have different vertical formatting than the rest of the manuscript; l. 1062 and 1063 are too close in vertical as well)

l. 954 you may want to insert a blank space after the bracket

In the caption of Tab.2 'resolution' instead of 'resoltion'

In the body of Tab.6 – should be 'Public users have to register for obtaining an account'

In the caption of Tab. 18 you need to insert a blank space after etadot

In the caption of Tab. 24 it is difficult to understand the first sentence starting from: 'Python code'; could you, please, re-phrase. I would also suggest putting sections v7.0.4 and v7.1 of this table side by side and not one on top of the other, if possible (I am aware there is no strict correspondence between theses two sections)
* * *

---

## Author Comment (AC1) · 10 Jul 2020

**Answers to comments by Anonymous Referee #1**

We thank the reviewer for his/her careful reading and his/her comments on our manuscript. We are very grateful for the comments resulting from thorough testing of the installation process and the usage instructions. This is very beneficial for improving the software and its documentation. We received additional bug reports from users, which we processed and provided as a bugfix release version v7.1.1 already. Feedback about new features or cosmetic changes were collected and kept for future releases.

The point-by-point replies to the comments are provided below:

**Major comments:**

Comment #1: *Type of manuscript The manuscript was submitted as a 'model description paper'. However, it does not describe a geoscientific model as such, but a pre-processor for the FLEXPART transport model. As such I strongly feel it should rather be in the category 'development and technical papers'.*

Answer: We agree and have already asked the editor to change it.

Changes in manuscript: The manuscript type was changed to the category 'development and technical paper'.

Comment #2: *Overall the manuscript is too long. Considering that this is 'only' a pre-processor and only few scientific methods are described, it should be possible to present the material in a more concise way, focussing more on the application side of things.*

*First of all, the manuscript should more clearly distinguish between methods/scientific background and the application/user's manual section in the appendix. Right now there seems to be too much of a mix between the two, especially in section 3. Here are my suggestions for improving the flow of the main manuscript.*
*Section 1: Start with description of FLEXPART/FLEXTRA and the need for appropriate input data. Then the history of flex_extract. Wherever possible move historical information from sections below into this section.*
*Section 2: This section is very valuable, especially the provided Tables summarising the ECMWF data availability, which is cumbersome to obtain from ECMWF/MARS itself.*
*Section 3: Especially convoluted part between methods and application. Large parts of section 3 especially sub-sections 3.4 and 3.5.x are focussing only on application and should be moved to appendix.*
*Section 4: Very valuable, but could have clear description of settings for recommended best practices.*
*Section 5: Although, documenting, testing and benchmarking of a code are as important as the code itself, I suggest to significantly shorten this section and put the details into supplementary material. I can see that a lot of work went into this part, but for most readers/users this information is less relevant and of no direct consequence.*

*Appendix: This part should work as a user's manual. A such I think the structure would work better if those parts that differ between application modes would be organised by application mode and not by different installation/application steps. This would allow a user, who will usually be running the software in only one of the modes, to follow along without having to jump from section to section.*

Answer: Although it is "only" a preprocessor, it is still a very complex task. In the past, users often had problems to use the software, for example with respect to choosing suitable combinations of parameter values and setting the parameter values correctly. A focus of this manuscript is therefore on thorough documentation of the software structure, its behaviour and the parameters that the software controls so that users understand what the software can do, where its limits are and which data sets can be extracted. We have kept the description as short as possible. It is necessary to include some hints and suggestions for use; we consider it appropriate to mention them in  in the main part of the paper and do not think that they belong into the appendix. The appendix is intended to contain purely technical instructions for installation and use without additional scientific or application-related information.

Due to the complexity, we have created an additional on-line documentation that will provide more application examples in the future and should be continuously updated based on user feedback. Frequently occurring problems and their solutions should be described there as well as future changes.

Changes in manuscript: As for the structure of the manuscript, we have largely implemented the comments: Section 1 has been changed as suggested. We have moved some application-related comments from section 3 to section 4 and have also shortened section 3 somewhat. Section 4 has been supplemented with a few more hands-on details and an additional reference to the on-line documentation, where we already describe many of the parameters and CONTROL files in more detail. We shortened section 5  by moving a lot of material to the supplement. The technical description in the appendix has been optimised to keep the appendix short. We believe that restructuring as the Referee describes it would worsen the overview.

Comment #3: *As stated by the authors themselves, flex_extract is a collection of shell scripts and python scripts and FORTRAN code. As such there seem to be many possibilities to break the code in case external dependencies change. I am wondering if it would not be a much cleaner approach to provide this tool as a single python package. This should make maintenance, documentation and installation a lot easier. Python codes are able to work with external FORTRAN codes as well. So it would still be possible to make use of the arguably faster computations done in the FORTRAN code. But there seems to be little need for mixing in korn shell scripts (other than for job execution on ECMWF servers). Some interesting information in this respect can be found here: https://packaging.python.org/tutorials/packaging-projects/ https://www.numfys.net/howto/F2PY/ This is more a comment about the future of the code and may not be achieved quickly as part of a revision of this submission. It would also be beneficial to host the code on a more commonly used platform like github or gitlab. I was not able to git clone the code from the address given in the manuscript and would not know how to provide code feedbacks in form of merge/feature requests on the platform that is currently used for hosting the code.*

Answer: We are aware of the software's complexity and that there are many possible problems due to external dependencies. From user feedback about previous versions, we know that there were many problems precisely because of this and, among other benefits, the new version is supposed to make the whole installation and usage process a lot easier without changing too much of the

original behaviour of the software. This was achieved mainly by making use of system and Python packages instead of building the libraries from source, in addition to improving the documentation. We agree that there are many additional possibilities to improve this tool, and are aware of several of them; some are already mentioned in the "Future work" -Section 6.3, such as creating a single package for installation. So far, we have not decided on the method to accomplish that.

We would also like to note that there are only korn shell scripts for the job script submission to the ECMWF batch queue and nothing else. The additional bash scripts for the setup and running the software are not mandatory to use; however, we know from experience that many new users to flex_extract appreciated this easy-to-use access point. We also know from feedback that users of v7.1 were more often successful in installing the software in the first approach than with previous versions.

The code is hosted in the FLEXPART community git repository, and we will not host it on another platform to avoid confusion and to minimise maintainance efforts. We are happy if there are users who would like to contribute as additional developers and they can be granted write access to the git repository upon request (ticket or email). So far, only cloning of the git repository is possible for the public, and we are very sorry to admit that the link to the git repository in the manuscript was wrong. It was the link to  FLEXPART and not flex_extract, which was the reason why the reviewer was not able to clone it. We corrected this link and added some details about possible contributions to Section 6.2 (Support). Feedback for new features or about bugs is very much welcome; it can be submitted through our ticket system.

Changes in manuscript: corrected link to git repository in the cloning instruction:
`git clone --single-branch --branch master` https://www.flexpart.eu/gitmob/flex_extract
Added a sentence for code contributions to Sect. 6.2: *Future contributions to the code are welcome; for granting permission of write access to the git repository, communication via email or ticket is necessary.*

**Minor comments:**

Comment #4: *P2,L39: 'non-authorised' is a bit strange. Users still need to be registered and as such are authorised to use the system. I would rather call this non-member state users.*

Answer: We agree.

Changes in manuscript: changed as suggested

Comment #5: *Version number, title and first mention P2,L44: If a totally revised version was created that is fundamentally different from the previous versions and as such not compatible with, for example, previous config files, it may be more appropriate to express this by incrementing the major version number. So why not go for 8.0?*

Answer:  We can understand the question regarding the version number, but we want to stick with version number 7.1.x for the current release because of the following reasons:
1. Even though the code as well as the directory structure were changed substantially, the general behaviour did not change. Moreover, previous CONTROL files can still be used. We mostly added new features and eased the usage and understanding of some of the CONTROL parameters. We also had to rearrange the program flow to be able to allow the retrieval of CERA-20C and ERA5 data as well as using the different application modes. From the perspective of users of version 7.0, these changes are not fundamental.

2. Version 7.1 was already advertised through several presentations and posters at conferences, and we don't want to confuse users by changing the version number.
3. Before releasing version 7.1, we had some minor releases as versions 7.0.2, 7.0.3, and 7.0.4; each of these versions already included first quick-and-dirty implementations of some of the new features present in v7.1. They were not yet bug-free, adequately documented, or adequately tested. (We had to prepare them upon request and release them faster than we could prepare v7.1.) Releasing the current version as 7.1 shows this continuity.

As some bug fixes have been released after the manuscript submission as v7.1.1, and as the corrections resulting from user feedback and reviews warrant another increase of the micro-version, the revised manuscript now refers to version 7.1.2.

Changes in manuscript: No changes.

Comment #6: *P3,L70f: New version of what? I guess flex_extract, but could also refer to FLEXPART. There is also no clear mentioning above about how the first version looked like.*

Answer: We rephrased the sentence for clarification.

Changes in manuscript: *When the Comprehensive Nuclear Test Ban Treaty Organization (CTBTO) started to use FLEXPART operationally, it became necessary to adapt the extraction software (consisting of Korn shell scripts and Fortran programmes for the numerically demanding calculation of etadot) such that it could be integrated into ECMWFs automatic data dissemination system. This became the first numbered …*

Comment #7: *P3,L85: Also mention what happened to the FORTRAN part.*

Answer: A sentence was added.

Changes in manuscript: *… The Fortran part underwent some mostly cosmetic changes (source format, file names, messages, etc. and a minor bug fix) and an overhaul of the makefiles. …*

Comment #8: *P4,L100: 'most accurate data source'. That's a bit general. So ECMWF is better than any other atmospheric model? I would not go that far, but if you do you should link this comment to some kind of model evaluation that provides a similar conclusion.*

Answer: We reformulated the sentence to include the arguments why ECMWF provides the most accurate data for FLEXPART.

Changes in manuscript: *… which is considered to be the most accurate data source, ECMWF being one of the leading global weather forecast centres and providing data on model-level and at high time resolution.*

Comment #9: *P4,L103: Last sentence can be removed, should be clear from context.*

Answer: We agree. This comment is in line with Anonymous Referee #2's opinion.

Changes in manuscript: removed

Comment #10: *P4,L108: I would not say that they share the code, since these are two completely different software packages. But one could say that some of the code goes back to the same original routines. Sharing the code would mean that their would be a common specific library of routines that would be used by both models.*

Answer: We agree.

Changes in manuscript: *FLEXPART is based on it and some code goes back to the same original routines from FLEXTRA.*

Comment #11: *P4,L108: 'It ingests ...', It should be referring to FLEXTRA, but previous sentence on FLEXPART. Replace.*

Answer: Ok

Changes in manuscript: *"FLEXTRA ingests …"*

Comment #12: *P4,L131: 'are to be used' better 'are available' or 'can be used'.*

Answer: We would rather say "are utilised" instead.

Changes in manuscript: exchanged to *"are utilised"*

Comment #13: *P4,L131: 'Flex_extract automatically ...': Not clear how flex_extract comes in at this point. So for this section was describing the different access kinds to ecmwf. It needs to be made clear here what are the access possibilities and then in a second step how flex_extract is using them. Maybe don't mention Flex_extrac in this section at all.*

Answer: Since Sec. 2 is supposed to give a general overview specifically for explaining the data and access modes which are available through flex_extract, we think it is proper to mention flex_extract there. Regarding the sentence on "Flex_extract automatically …", we think it is the best place for mentioning this. Otherwise, we would unnecessarily need to repeat the information about the databases.

Changes in manuscript: No changes.

Comment #14: *P5,L141: Later on this dataset/model version is referred to as DET-FC. Why not keep the ECMWF terminology.*

Answer: This is due to the fact that it was originally the "Deterministic Atmospheric Model", thus DET-FC for the deterministic forecast. It was later changed to "Atmospheric High-Resolution Model". Since flex_extract can retrieve data from the whole available period, we decided to call it DET-FC. Nevertheless, it is confusing to start with HRES and then change back to DET-FC without mentioning a reason. We think that it is essential for the user to know both namies and modified the item accordingly.

Changes in manuscript: *The operational deterministic atmospheric forecast model (DET-FC), nowadays called atmospheric high-resolution forecast model (HRES),*

Comment #15: *P6,L157ff and Table 1: While the text nicely refers to the evolution in the IFS operational setup, I am missing a similar warning in Table 1. Seen on its own the Table may be misleading. Please add reference to Tables 2 and 3.*

Answer: We agree.

Changes in manuscript: … *The specifications for the operational data sets are valid for current data at the time of publication (except ENS-CV -- deprecated since 8 August 2016). For details about resolution and other parameters which have changed in the course of time, see Table 2 and Table 3. ...*

Comment #16: *P6,L177: 'member-state' instead of 'member'.*

Answer: Ok

Changes in manuscript: changed as suggested

Comment #17: *P6,L178: 'not good'. Can you more specific what is not good about? Not robust/reliable or simply (too) slow?*

Answer: Ok.

Changes in manuscript: … *this access mode is very slow, …*

Comment #18: *P7,L184ff: The use of access and applications modes is a bit confusing here. Especially when the text refers first to access and application and then the combination of these is once more called application mode. Maybe the primary application modes could be described more clearly as 'execution location'?*

Answer: We agree that this sentence is not clear enough. We will stick to the "application modes" since this is now an established phrase in the manuscript and the online documentation. However, we will exchange "application mode" with "execution location" in this sentence as it is more accurate and precise. We also rename the "user access mode" to "user group", as this is in line with Sect. 2.1. This phrase would then also be in line with the description in Sec 3.1.

Changes in manuscript: *The actions executed by flex_extract (also called "the software'" henceforth) depend on the user group (see Sect. 2.1), the location of execution, and the data to be retrieved. There are three possible locations of execution, namely the ECMWF Member State Linux servers, the Member State Gateway server, or a local host.*

Comment #19: *P7,L213: 'module system': Rather call it 'environment modules framework'.*

Answer: Ok

Changes in manuscript: changed as suggested

Comment #20: *Figure 1: Could this not be drawn much simpler without the emphasized distinction between the two different web APIs? The main point of this schematic should be an illustration of the 4 application modes. To user it does not matter so much that there are two different web APIs.*

Answer: We don't think that there is a reasonably way to significantly simplify Figure 1. It is important for the users to know which API they are going to use for a specific data set, as this has implications for the API which needs to be installed and for which one has to register.

Changes in manuscript: no changes

Comment #21: *P8,L231: Why not mention this python package in Table 7 as well?*

Answer: Because this table contains the software needed for all application modes, and ecmwf-api-client and cdsapi are only needed for the local mode.

Changes in manuscript: no changes

Comment #22: *P9,L254f: In the schematic (Figure 4) the "remote mode" is on top. So why start with explaining the "gateway mode" first?*

Answer: Ok. We rephrased this paragraph accordingly.

Changes in manuscript: *The remote and gateway mode both create a job script using the command-line parameters and the content of the specified CONTROL file and then sent it to an ECMWF batch queue. In remote mode this happens on an EMCWF server while the gateway mode uses the local gateway server for the creation and submission of the job. As the job script is executed from whichever of the two modes, it creates the job environment (in particular, the working directory) and starts "submit.py" to retrieve and post-process the data. Note that this locally started instance of "submit.py" triggers the work flow of the local mode but uses the MARS client to extract the requested fields from the database. The final output files are sent to the local member-state gateway server only if the corresponding option was selected in the "CONTROL" file. When flex_extract is used on a local host and in local mode, fields are extracted from MARS using one of the Web API's (which sends HTTP requests to ECMWF/CDS) and are received by the local host without storage on ECMWF servers.*

Comment #23: *Figure 4: The schematic may be easier to comprehend if ECMWF and local machines would always be on one side. Also the Labels for the different modes should be given rather on top with the sub-panel index. Right now the may be mistaken as only referring to the boxes on the right.*

Answer: Ok.

Changes in manuscript: Figure was changed accordingly.

Comment #24: *P9,L256: Once again the use of he different modes is confusing. While describing the 'gateway mode' it is mentioned that flex_extract is started in 'local mode'? Confusing. If I understand correctly, it is started locally to generate the ECMWF scripts and then sends them through the gateway. 'local mode', however, is not using the gateway but the WebAPIs. Correct? The same confusion is created in the caption to Figure 2, where 'local mode' is mixed into both gateway and remote mode.*

Answer: This comment is connected to #25, #26 and #22. We rephrased to whole paragraph for clarification.

Changes in manuscript: see answers to the other comments

Comment #25: *Figure 2: What is the purpose of the 'do local work' branch of the flow chart?*

Answer: As indicated in the figure caption it points to the branch where the local mode is executed, which constitues of the retrieval and post-processing tasks. The local mode is demonstrated in Figure 3. We rephrased the caption for clarification.

Changes in manuscript: *Flow diagram for the remote and gateway mode. A job script is created and submitted to the batch queue on an ECMWF server. The job script will then be executed on the ECMWF server to start flex_extract again for retrieving and post-processing of the data. The branch indicated by "queue = None" refers to the work flow shown in Figure 3.*

Comment #26: *Figure 3: Again, I am not sure that I understand the purpose of the two side-branches 'submit' and 'PRINTING MARS_REQUEST'.*

Answer: The submit branch indicates that flex_extract was started in remote or gateway mode and a job script has to be submitted. The "PRINTING MARS_REQUEST" branch indicates that flex_extract was started with the parameter REQUEST = 1, which means that only the mars_request.csv file is created and no retrieval or post-processing is done. We rephrased the caption for clarification.

Changes in manuscript: *If queue != None, flex_extract was started in remote or gateway mode and Fig. 2 applies.*

Comment #27: *Section 3.4.1 and Tables 8 and 9: I feel that these tables include the core information of how to setup and run the data extraction correctly. As such I think the description of the individual parameters should be given more space in the main text (as part of the Appendix) and not only in the supplement. This is crucial information for using flex_extract!*

Answer:  We rather consider the Appendix as a technical description. Therefore, we don't discuss the parameters from the Tables there, and rather do that in Section 4, where we added further information on the most critical and potentially confusing parameters in combination with references to example CONTROL files.
Additionally, we recommend to use the online documentation as the future up-to-date source of information for the list of parameters and their value range. Additional detailed examples, which we thought would be too much for this paper, are already available there.

Changes in manuscript: See comment #38 for changes to Section 4.

Comment #28: *P14,L340: 'via the gateway server': not in 'remote mode'!*

Answer: Yes, this is a misleading statement, and it has to be made clear that the gateway server is needed in the gateway mode only.

Changes in manuscript: … *submitted to the ECMWF batch system. In case of the gateway mode, this is done via the local gateway server.*

Comment #29: *P14,L351: Do these modules correspond to the according boxes in the flowchart (Fig3)?*

Answer: They do. To make this connection recognisable, we added a reference to the figure. However, this description was moved to Section 4 due to the restructuring from major comment #2.

Changes in manuscript: *There are two more entry points into the software which can be used for debugging; they are described here for the sake of completeness: the Python scripts "getMARSdata.py and "prepare_flexpart.py". In the standard way of running the software, they are both imported as modules (as shown in Fig. 3), but they can also be used as executable programs.*

Comment #30: *P15,L363: Who is Paul James. Is there a reference to this work?*

Answer: Paul James developed and wrote the first disaggregation method, around 2000. There is no publication on that. We have removed the reference to his name, as admittedly it appears a bit as out of context, and rephrased the sentence slightly for clarification.

Changes in manuscript: *A pre-processing scheme is therefore applied to convert the accumulated values to point values valid at the same times as the main input fields while conserving the integral quantity with FLEXPART's linear interpolation.*

Comment #31: *P16,L390ff: At this stage it is not clear how these additional inputs are treated. Will there be separate output files for these times? How does FLEXPART deal with these? From which FLEXPART version onwards is this feature supported?*

Answer: The paragraph starting at p.16, l.390 already describes the process of how theses times are treated in flex_extract. The current FLEXPART version can not distinguish between the three time steps of a single interval. It will read all of them one after each other and keep the last one read from the file. Therefore, it would always work with the data from the second additional grid point and, obviously, we don't recommend using this scheme with current FLEXPART version. The FLEXPART version which supports the additional data will probably be the next one, v10.5. However, its release date is not yet determined. We added this information to the manuscript.

Changes in manuscript: *Current FLEXPART versions can not properly handle this input files generated with the new disaggregation scheme; they would use the third field (second additional sub-grid point in time), which would be worse than using the current method. The next minor version of FLEXPART to be released shall support the new scheme.*

Comment #32: *P16,L394: Maybe it would be useful to explain briefly why other fluxes are treated differently from precipitation.*

Answer: The other fluxes assume both signs and thus don't require conservation of positive definiteness. Furthermore, their impact on the FLEXPART results is smaller than that of precipitation, which is why the new scheme has been developed first for this latter variable. Motivated by the review comment, we have looked more in depth at the historical scheme implemented for the other fluxes and discovered that it does not correspond to FLEXPART, as it assumes cubic (not bicubic as wrongly stated in the manuscript) interpolation. In the future, we therefore want to adapt the scheme for the other fluxes similar to that of precipitation. However, we simplified the current formula.

Changes in manuscript: *The accumulated values for the other fluxes are first divided by the number of hours, and then interpolated to the times of the major fields. The algorithm was designed to conserve the integrals of the fluxes within each time interval when reconstructed with a cubic polynomial. It uses the integrated values F during four adjacent time intervals ($F\_0,F\_1,F\_2,F\_3$) to generate a new, disaggregated point value F which is output at the central point of the four adjacent time intervals:*

$$F = -\frac{1}{12}F\_0 + \frac{7}{12}F\_1 + \frac{7}{12}F\_2 -\frac{1}{12}F\_3$$

*Note that a cubic interpolation was never implemented in FLEXPART. We therefore plan to replace this scheme by an adaption of the scheme used for precipitation, adapted to the situation where both positive and negative values are possible.*

Comment #33: *P18,L453: grib_api seems to removed from all the makefiles and is also not mentioned in the dependencies (Table 7).*

Answer: Yes, the makefiles now expect eccodes instead of grib_api. However, as both provide the same functionality with respect to GRIB, one may still use grib_api. We rephrased the respective sentence to make that more clear.

Changes in manuscript: *The code is provided with a set of makefiles. The standard version assumes a typical GNU/Linux environment with the "gfortran" compiler and the required libraries: "OpenMP" for parallelisation which is included in the "gcc" compiler package ("libgomp"), "ecCodes" for handling GRIB files, "EMOSLIB" for transformation between the various representations of fields. Note that the latter two typically require also so-called developer packages containing the Fortran module files. One may substitute "ecCodes" by its predecessor "GRIB_API", if "ecCodes" is not available.*

Comment #34: *P18,L456: Usually one would refer to these versions as optimised and debug versions, not fast and debug. It would also be better to have a single makefile and only use a switch in the makefile to create either of the desired executables.*

Answer: "fast" is shorter, so we use it in the file name. A makefile switch for the selection of debugging vs optimisation is certainly an option, but we do not consider it a big difference. In that case, instead of the makefile name, the switch value to be used when calling make would have to be provided in the installation script. There might also be other options for makefiles, for example using a different compiler, therefore the flexibility provided by the current solution appears desirable.

Changes in manuscript: no changes

Comment #35: *P19,L483: Does the comment in braces mean, that no two instances of the submit.py script should ever be running at the same time? Where do you expect clashes if done anyway.*

Answer: There can be multiple instances at the same time if each instance uses a unique input directory. This is done automatically in remote and gateway mode. In local mode, the user has to take care of setting the input directory. The retrieved data from MARS are stored in *.grb files which have process ids in their filenames for identification. This feature is used in the post-processing part of flex_extract to adress them. It also allows users to start only the post-processing part again if the program, for example, terminated earlier due to time limit.
However, mixing of different retrievals can happen in the post-processing where several files with the same names are created. They can be overwritten without knowing, mixing the data sets. Additionally, files containing a time stamp ( e.g. flux files) can also be mixed if the retrieval time period is overlapping.
Therefore we rephrase the sentence and add a note in the Section "Execution" (3.5.2). A proper solution may be implemented in a future version of flex_extract.

Changes in manuscript: *The process IDs are incorporated so that the GRIB files can be addressed properly in the post-processing.*
In Sect. 3.5.2: *Please note that when flex_extract is started in local mode, the parameter INPUTPATH in the run_local.sh script must be set, so that each retrieval uses a unique directory to avoid mixing of data files.*

Comment #36: *P20,L530: What about forecasts longer than 99 hours. How will they be written into a 2 digit number?*

Answer: The forecast always starts from an analysis at a specific hour which is called forecast time. From this time, each hour into the forecast is called a forecast step, and the forecast could indeed be longer than 99 hours. Therefore, the forecast step is already designated by a 3-digit number as noted in this Section 3.9.2. The forecast time, marked by HH, is always a two-digit number. To get the full date of any forecast step, the date, forecast time and forecast step have to be used to calculate the real date and time.

Changes in manuscript: No changes.

Comment #37: *P22,L577: This is comment that cannot be fixed in the manuscript, but should be addressed in the future. I assume you are referring to the Global Forecast System of NCEP here? If the availability of model level data is a shortcoming to the community. NCEP should be approached to make model levels available as well. Internally GFS operates on a similar hybrid sigma-pressure coordinate system as ECMWF IFS.*

Answer: Yes, we refer to GFS. However, we do not see it in our responsibility to approach NCEP for model level data. Of course, users are free to pursue this and we assume that the FLEXPART developers would support a corresponding update of the FLEXPART code (which would be necessary in that case).

Changes in manuscript: no changes

Comment #38: *Section 4: The section gives many valuable hints for best practices. However, I would find it very useful if these could be manifested in more concrete instructions, maybe in form of clear example CONTROL files. For example it is mentioned that only a subset of levels can be extracted, but how is this done concretely and does it work for all etadot modes (didn't in the past)?*

Answer: Where appropriate, we added references to example CONTROL files, or described how the parameters have to be set. We also moved some of the usage instructions from section 3 to section 4 and extended the instructions to describe new and/or confusing parameter settings.

Changes in manuscript: Section 4 was further divided into subsections "4.1 Example CONTROL files", "4.2 Changes in CONTROL file parameters in comparison to previous versions" and "4.3 Scientific considerations".   Section 4.1 was previously Section 5.5  with some additional information about file names. Section 4.2 lists the most important changes to the CONTROL files in version 7.1 and Section 4.3 is the previous Section 3 with some additional references to example CONTROL files.

Comment #39: *P28,L746f: Actually one does not need any code for this task. It can be performed using standard bash cat command. That is one of the few beauties of grib format.*

Answer: We are aware that standard bash cat commands could be used. However, as the Python ecCodes package contains a specific command to do the merging, and as the workflow is implented in Python, we deem it more useful to use the ecCodes command.

Changes in manuscript: No changes

Comment #40: *P28,L752ff: See also previous comment on package structure. The possibility of calculating etadot should not just be discarded, but there certainly are better ways of integrating the FORTRAN code into the python package.*

Answer: Calculation of etadot will not be discarded, especially because of compatibility reasons and older data sets, such as ERA-Interim. We rather anticipate a solution where the software calls calc_etadot only if etadot needs to be calculated. We reformulated the comment in the "Future work" Section (6.3). Please see also comment # 3 (major section) for the answer (repetition).

Changes in manuscript: *In future versions of flex_extract, calc_etadot will probably only be called if etadot really needs to be calculated, not just for multiplying it with \partial p / \partial eta  as this can be done with sufficient efficiency in Python.*

Comment #41: *P28,L755f: Again as mentioned above, instead of just providing a tar ball, a public git repository that would allow for swift user feedbacks and contributions would be beneficial.*

Answer: Please see comment # 3 (major section) for the answer (repetition).

Changes in manuscript: No changes

Comment #42: *P29,L790: Better refer to this as dependencies or prerequisites.*

Answer: Ok

Changes in manuscript: We changed the heading of appendix sub-section A2 to *System prerequisites*

Comment #43: *Code sections in Appendix (e.g., P31,L850ff): It would be easier if such sections would be directly given as example config files or scripts. Copy and paste from a manuscript is rather cumbersome and error-prone.*

Answer: We prepared the shell scripts and Python code snippets as supporting scripts for the users. However, we refrained from including the installation commands for the additional software and library packages into a script since the installation commands are distribution-dependent.

Changes in manuscript: *Shell scripts and Python code snippets mentioned in the Appendix can be found in the directory Testing/Installation/ after unpacking the tarball.*

Comment #44: *P34,L994: Does testing CERA-20C refer to CONTROL_CERA or is there another prepared example to test it. Please mention.*

Answer: The details of the data set are described in the following subparagraphs, where we provide step-by-step instructions for this first test with CERA-20C data, depending on the application mode. Since it is thus mentioned twice, we decided to move the details up, out of the subparagraphs, and to add the name of the CONTROL file.

Changes in manuscript: *In the following, we will provide step-by-step instructions for all application modes to retrieve a single day (08 September 2000) from the CERA-20C dataset with 3hourly temporal resolution and a small domain over Europe with 1° resolution, using CONTROL_CERA[.public].*
We removed the details from the subparagraphs "Remote and gateway modes" and "Local mode".

Comment #45: *P35,L103: This unload was not mentioned above for the installation process. Is it necessary? Because once you unload python on ecgate the python command is linked to the system pyhton version 2.6! So I don't see the gain of the unload.*

Answer: This is indeed not necessary and might confuse readers. Therefore, we removed this line.

Changes in manuscript: Line 1003 (*… module unload python ...*) deleted

Comment #46: *Table 7: It is mentioned in the text that compilers other than GNU compilers were tested as well. This should be reflected in the table. emoslib should just be emos. The information for fortan/pyhton is not required since this is given by the header line.*

Answer: A makefile is provided for the ECWMF HPC environment tested with the Fortran compiler included in CrayPE 2.5.9. Originally, also an Intel Fortran makefile was included, but this has been removed for the sake of simplicity, as gfortran is sufficient nowadays. As ECMWF itself calls the EMOS library "emoslib" in its Wiki system, we think it is better to keep this designation. We accept the proposal to simplify the caption.

Changes in manuscript: Caption has been simplified, information on HPC ftn compiler added.

Comment #47: *Table 11: With files job.temp and job.template it is difficult to distinguish their function just from file name. The second one should maybe be renamed to install.job.template Why does the first template not have the template file name extension. Is it treated in a different way than the other templates? Same for convert.nl. Even if the output of the template processing is a namelist shouldn't it have the same template ending.*

Answer:  We agree and have renamed the template files as follows.

Changes in manuscript:
*convert.nl -> calc_etadot_nml.template*
*compilejob.template -> installscript.template*
*job.template -> jobscript.template*
*job.temp -> submitscript.template*

Comment #48: *Table 20: Is this table really needed? It is only a list of file names without further description. I can obtain this also by looking into the corresponding folder! Either remove or add a meaningful description of the listed files. The caption alone does not explain the details.*

Answer: We agree with the comment and delete this table; instead some more specific references for examples are added in Section 4. Additionally, the on-line documentation will be expanded in the future and will include more detailed descriptions.

Changes in manuscript: See comment #38 for changes to Section 4.
We removed Table 20 and its reference.

**Technical comments:**

Comment #49: *P3,L83/84: Past tense in previous sentence. Why not here?*

Answer: Agreed.

Changes in manuscript:  *Version 7.0.4 enabled the retrieval of multiple ensemble members at a time and included bug fixes for the retrieval of ERA5 and CERA-20C data.*

Comment #50: *P4,L106: Style. Twice representing in same sentence.*

Answer: Ok

Changes in manuscript: changed as suggested

Comment #51: *P5,L146: Citation style for Laloyaux et al. (2018)*

Answer: Ok

Changes in manuscript: changed to … *CERA-20C (Laloyaux et al., 2018) reanalysis.*

Comment #52: *P6,L181: 'necessary' not 'ecessary'*

Answer: Typo

Changes in manuscript: corrected

Comment #53: *P16,L387: Citation format of '(Hittmeir et al., 2018, p. 2513)'. No pages in journal reference.*

Answer: We thought that it might be useful for the readers if they don't have to search within the cited article. If it is contrary to the GMD style, the technical editor will remove it.

Changes in manuscript: no changes

Comment #54: *P17,L414f: So in the end FLEXPART converts etadot (or etapoint) back to a Cartesian vertical velocity? So why the painful calculation of etadot in the first place? Only because of the historic development from FLEXTRA?*

Answer: The multiplication of etadot with deta/dp (partial derivative!) does not create a Cartesian vertical velocity, as can be easily seen by looking at the situation close to sloping ground, where etadot is close to zero while w and omega won't be.

Changes in manuscript: no changes

Comment #55: *P17,L416: 'pressure' not 'perssure'*

Answer: Typo

Changes in manuscript: corrected

**Code comments:**

Comment #56: *The FORTRAN code works with default FORTAN unit names for I/O (see table 17). This seems a bit outdated and difficult to follow/debug/etc. Please change to meaningful filenames wherever possible. This would allow removing Table 17 from the manuscript.*

Answer: We plan to do this, but not for the current release. It will require some adaptations in the regression testing, which goes too far for the manuscript revision. As users don't normally interact with these file names, we don't see it as a priority.

Changes in manuscript: no changes

Comment #57: *Besides the documentation in the main routine the comments in the FORTRAN code are mostly in German. This should be changed in future submissions.*

Answer: We are aware of this, but because of the uncertain future of the code, we haven't yet put much work into it. However, the main description at the start of each program unit (which appear also in the FORD-generated on-line documentation) are now all in English.

Changes in manuscript: no changes

Comment #58: *Section 3.8.3: And why would the FORTRAN program need to save these variables to a file instead of storing them internally? Makes little sense if they are not used again, for example, by the python scripts.!*

Answer: As stated in a comment line of the Fortran programme, orginally this file was used by a simulation model called POP model. We have commented out the write statement in the code now.

Changes in manuscript: The section was removed.

Comment #59: *Installation process: I only tested the remote mode on ecgate. My experience was not as straightforward as the description may suggest and as one would expect on a system the code should have been tested on. Since the remote installation is the one installation mode that is running on a defined system it should be easier to get this done without having to change various parameters in the setup script. First of all, I tried the git clone command on page 28. However, this does not clone flex_extract, but FLEXPART!? In the following I could not figure out the correct git address for flex_extract and resigned to to downloading the tar ball instead. Once unpacked, I was able to proceed. However, I don't see any reason why the installation process is submitted to the queueing system on ecgate. We are talking about a very small compile task here. It would be more simply done on the login nodes so that one has a more direct feedback when things go wrong. Anyway, first email I get back from the submitted install job was a very general compile error ("No rule to make target 'phgrreal.f', needed by 'phgrreal.o'. Stop"). So I checked if I selected the correct makefile. Yes, one would think: makefile_ecgate (the one system where this should work. Right?). Taking a look into makefile_ecgate shows that file extensions seem to have changed from f to f90, but this was not adjusted in the makefile. So I modified the makefile. Second try. Got an email which may indicate success of the FORTRAN compile, but does not tell me anything else about the rest of the installation. So let's see if it is there. Hm. Not in the directory I had specified in $INSTALLDIR. But there is a new folder flext_extract_v7.1 in my home. OK. Fine. Let's test it. So I follow the instructions in Appendix A4. However, the executable mentioned there is called calc_etadot_fast.out, while in my installation it is simply called calc_etadot. No problem. But shouldn't the exe name be harmonised in the makefiles? I did not change it. Running the executable seems to work but in my case only produced 2 lines of output in contrast to 7 mentioned in Appendix A4. Once again, did I do something wrong. Did it really work? There are 2 output files in the directory now and one can assume that that's sufficient. grib_ls on fort.15 (the obvious output file) at least shows some etadot fields. However, I was still worried about the missing lines of output. So I quickly searched the FORTRAN code for where these messages should be produced. I found them but they are commented out. So no wonder that I did not get these messages! But why are they given in the instructions.*

Answer: Unfortunately, the git clone command in the installation instructions incorrectly pointed to FLEXPART. Testing of flex_extract was done regularly. During the whole development process, and right before the submission of the manuscript, the code has been tested. However, because the existing flex_extract directory had not been removed from the ECWMF server before we started the new installation, we did not notice the problems with the makefiles and outdated source files. This

was corrected right after we received the first bug report in a new flex_extract version 7.1.1. We also harmonised the name of the Fortran executable by linking it to "calc_etadot", regardless of the makefile selected. There might be additional names which are used for the regression tests and not important for the user. We are very sorry that you had this rough start with our software. Regarding the easier usage of the setup.sh script, please note that flex_extract users choosing gateway or remote mode always need to change user-specific parameters in the setup.sh script, therefore it is not easily possible to further simplify the process. The script already includes default settings for gateway and remote mode.

Concerning the installation process in remote mode, we would like to state that flex_extract distinguishes in its behaviour between ECMWF servers and local hosts. Even though, technically, the gateway and remote modes work differently, they share the same code. Therefore, we will stick to this procedure for now and keep the idea of a local installation for the remote mode as an option for a new feature. Additionally, the remote and gateway mode have both a fixed installation target, namely the $HOME directory, as described in the paper. Currently, this is not changeable.

The differences in the output from the Fortran code compared to what was given in the installation instructions are due to modifications made to the code in a late stage of preparation, unfortunatly having overlooked to update the manuscript.

We went through the steps in the installation instructions again to make sure that everything is now up-to-date.

Changes in manuscript: Corrected the git clone command and the Fortran test output.

Comment #60: *Test retrieval: A test retrieval using the CONTROL_CERA example worked without much trouble. I would only suggest that the loading of the python3 module should be added to the run.sh script.*

Answer: Thank you for the suggestion. We have added automatic loading of Python3 module when the setup or run script detects an ECMWF server.

Changes in manuscript: Removed instruction to load Python3 module in the installation and usage sections.

Comment #61: *EcFlexpart.py: As far as I can see this is where the ectrans command is used to transfer the data. The ectrans command is used in a way that assumes that it would crash if an invalid transfer was submitted. But this is not how ectrans works. Much like a queuing system ectrans submits the requested file transfer into a queue returning a request ID. transfer request will then be repeated if not successful at the first try. So if the settings for source and target file are not correct ectrans will still return a request ID and try the transfer until it reaches its repeat limit. Hence, it is not possible to simply capture the return code from ectrans to detect an error.*

Answer: We know how ectrans works. However, the ectrans command is executed, like all other command-line calls, from Python scripts, making use of an extra function. Therefore it undergoes error handling, even if an error will never occur. We don't think that changes in the code are necessary. As one might perceive a need for an explanation, which is lacking in the manuscript so far, we added a note in the usage section for remote and gateway modes.

Changes in manuscript: *Please note that success of the submission of the ectrans command does not guarantee that the file transfer will succeed. It means only that the output file has been successfully submitted to the ectrans queueing system. One still has to check manually in the local directories or with ECaccess tools whether the files reached their final destination.*

Comment #62: *Work folders on $SCRATCH are called python<PID>. Why not use something that would clearly identify these as flex_extract folders?*

Answer: This has historical reasons; it comes from version 6 where Python was first introduced. However, we agree that the prefix could be more telling, and changed it to "extract" instead of "python".

Changes in manuscript: We changed the code and text accordingly.
Manuscript p. 36, l.1048: … *temporary retrieval directories (usually extractXXXXX, where XXXXX is the process id). Under extractXXXXX a copy of the CONTROL file ...*

---

## Author Comment (AC2) · 10 Jul 2020

**Answers to comments by Anonymous Referee #2**

We thank the reviewer for his/her comments on our manuscript. Since the other referee (#1) suggested to shorten the manuscript, we have shortened Section 5 on quality assurance, transferring a major part from this section to the supplement.

The point-by-point replies to the comments are provided below:

**Minor comments:**

Comment #1: *line 58, 1990ies (and later), change to 1990s or '90s: MLA Handbook for Writers of Research Papers, 7th edition. This, from p. 84. "Decades are usually written out without capitalization (the nineties), but it is acceptable to express them in figures (the 1990s, the '60s). Whichever form you use, be consistent."*

Answer: Ok

Changes in manuscript: changed to 1990s

Comment #2: *line 62, interSect –> intersect*

Answer: Ok

Changes in manuscript: changed as suggested

Comment #3: *line 67, and later of –> and later on*

Answer: Ok

Changes in manuscript: changed as suggested

Comment #4: *line 86, ecCodes (earlier referred to as eccodes) be consistent*

Answer: Thank you for this hint. We agree.

Changes in manuscript: changed to ecCodes throughout the manuscript

Comment #5: *line 86, would change 'was implemented' to 'was utilized'*

Answer: Ok

Changes in manuscript: changed as suggested

Comment #6: *line 98, ECMWF and IFS are already defined... and throughout, drop repeat acronym defintions*

Answer: We agree, except for the definition in the introduction section,  although it was already defined in the abstract (abstract may not be read by everybody).

Changes in manuscript: We eliminated repeated acronym definitions throughout the manuscript for ECMWF/IFS/MARS and others.

Comment #7: *line 103, "All details can be found in the literature mentioned" is an empty sentence, suggest dropping it.*

Answer: Ok

Changes in manuscript: eliminated as suggested

Comment #8: *line 113, use ECMWF as it has been defined...*

Answer: Ok

Changes in manuscript: changed as suggested

Comment #9: *line 116, MARS has been defined...*

Answer: Ok

Changes in manuscript: changes as suggested

Comment #10: *line 150, choose meta data or metadata and use consistently*

Answer: We chose meta data.

Changes in manuscript: changed throughout the manuscript to meta data

Comment #11: *line 155, "The horizontal grid type refers to the way how it fields are archived in MARS" not clear...*

Answer: We slightly rephrased the sentence, refering to the spatial representation which was already defined on page 4 lines 118-120.

Changes in manuscript: The horizontal grid type refers to the spatial representation.

Comment #12: *line 169, inplied –> implied*

Answer: Ok

Changes in manuscript: changed as suggested

Comment #13: *line 181, ecessary –> necessarily*

Answer: Ok

Changes in manuscript: changed as suggested

Comment #14: *line 720, Lag rangian –> Lagrangian*

Answer: Ok

Changes in manuscript: changed as suggested

---

## Author Comment (AC3) · 10 Jul 2020

**Answers to comments by Anonymous Referee #3**

We thank the reviewer for his/her comments on our manuscript.

The point-by-point replies to the comments are provided below:

**Minor comments:**

**Comment #1:** *l. 5 I suggest replacing 'therefore' with 'consequently' to avoid a repetition with the previous sentence*

**Answer:** Ok

**Changes in manuscript:** changed as suggested

**Comment #2:** *l. 22 should be 'Integrated Forecast System'*

**Answer:** Yes, that's an error.

**Changes in manuscript:** corrected

**Comment #3:** *l. 41 I believe you want ECMWF here*

**Answer:** Yes

**Changes in manuscript:** corrected

**Comment #4:** *l. 58 should read '1990s'*

**Answer:** Ok

**Changes in manuscript:** changed as suggested

**Comment #5:** *l. 62 You probably want 'intersect' and not 'interSect.'*

**Answer:** Yes

**Changes in manuscript:** corrected

**Comment #6:** *l. 99 Should read US National Centers*

Answer: Ok

Changes in manuscript: corected

Comment #7: *l. 107/108 – the two reference should be in brackets*

Answer: Agreed.

Changes in manuscript: corrected

Comment #8: *l. 116 reference should be in brackets*

Answer: Agreed

Changes in manuscript: corrected

Comment #9: *l. 146 reference should be in brackets*

Answer: Agreed

Changes in manuscript: corrected

Comment #10: *l. 155 suggest skipping 'it' or better replacing it with 'the'*

Answer: Ok

Changes in manuscript: We rephrased the sentence. (See comment # 11 of Anonymous Referee #2)

Comment #11: *l. 169 'implied; instead of 'inplied', I believe*

Answer: Yes

Changes in manuscript: corrected

Comment #12: *l. 181 missing 'n' in 'ecessary'*

Answer: Ok

Changes in manuscript: corrected

Comment #13: *l. 184 - 187 and then also l. 206 – 209: are 'access mode' and 'application mode' two different things? Are 'ECMWF Member States Linux servers', 'Member State Gateway server' and 'local host' rather access modes than application modes? And based on those three access modes, four application modes are derived? Maybe it is then clearer in conjunction with the header*

*of section 3.1, where 'Remote', 'Gateway' and 'Local' are referred to as application modes. And then in conjunction with user mode it results in four user application modes.*

Answer: This comment is somehow related to Anonymous Referee #1's comment 18. Yes, access modes and application modes are different. However, in this context the term "access mode" is better replaced by "user group" and the "application mode" in this first sentence should be replaced by the "location of execution". The description was not clear and we renamed the respective terms.

Changes in manuscript: *The actions executed by flex_extract (also called "the software'" henceforth) depend on the user group (see Sect. 2.1), the location of execution, and the data to be retrieved. There are three possible execution locations, namely the ECMWF Member State Linux servers, the Member State Gateway server, or a local host.*

Comment #14: *l. 187 results*

Answer: Ok

Changes in manuscript: changed as suggested

Comment #15: *l. 227 – depends which spelling you follow; if it is BE then 'licence'; but I noticed indeed that Copernicus Publications follow AmE spelling of this word, namely 'license'*

Answer: Yes. We use British English. Only in the case of the filename LICENSE.md AE is used following the dominant usage in the software community. In any case, the Copernicus editors will have the final word.

Changes in manuscript: no changes.

Comment #16: *l. 240 – I may be missing something important in the software structure but I do not understand why sending a script to the ECMWF batch queue in step 1 contrasts in steps to and 3; in particular do not understand 'or' here; I would imagine that that job sent to the batch queue also retrieves data from MARS and post-processes them to obtain FLEXPART input fields. From l. 249-252 I understand that flex_extract proceeds with steps 2 and 3 locally; is it correct? It is a small thing but it would be good to clarify*

Answer: Flex_extract sends a korn shell script as job script to the batch queue and when this script will be started from the queue eventually, it starts flex_extract again to finally retrieve (task2) and post-process (task3) the data on ECMWF servers. Other Reviewers had the same problems in understanding this logic and we therefore rephrased it. Please see comments #22, #24, #25 and #26 from Anonymous Reviewer 1 for our answer.

Changes in manuscript: Please see comments #22, #24, #25 and #26 from Anonymous Reviewer 1 for the changes.

Comment #17: *l. 268 – you want to use 'which' once*

Answer: Yes

Changes in manuscript: corrected

Comment #18: *l. 283 – should read 'for a correct setting'*

Answer: Ok

Changes in manuscript: changed as suggested

Comment #19: *l. 284/285 – you probably want to use the word 'combination' only once*

Answer: Yes

Changes in manuscript: corrected

Comment #20: *l. 355/356 – these papers should be referenced in brackets*

Answer: Yes

Changes in manuscript: corrected

Comment #21: *l. 363 – does it stem from private communication with Paul James?*

Answer: This comment is in line with Anonymous Referee #1's comment #30. Please see there for our answer.

Changes in manuscript: Please, see comment #30 of Anonymous Referee #1 for the applied changes.

Comment #22: *l. 373/374 – what is the position of 1 and 2 with respect to a,b,c,d*

Answer: This would be 'b' and 'c' respectively. Nevertheless, we rephrased the sentence for clarification.

Changes in manuscript: *... which is output at the central point of the four adjacent time intervals:*

Comment #23: *l. 416 should read 'pressure'*

Answer: Yes

Changes in manuscript: corrected

Comment #24: *l. 504 – what is a pure forecast? Do you mean deterministic forecast? 'Pure fore-cast' also later appears in l.527*

Answer: In our context, a pure forecast means that we extract only forecast fields and the retrieval period is longer than one day without analysis fields in between. We added this information in Sect. 3.9.2. and added a reference to this section from the first appearance in 3.8.5. We also changed the naming in "long forecast".

Changes in manuscript: *For a long forecast, where only forecast fields are retrieved for more than 23 hours,…*

Comment #25: *l. 513/514 – not sure what this sentence mean? Is there just 'is' missing in this sentence? Or is there more to add?*

Answer: There is just an 'is' missing.

Changes in manuscript: corrected.

Comment #26: *l. 524 - is YYMMDDHH in this file name the analysis hour?*

Answer: No, it is the valid time.

Changes in manuscript: *There is one file per time step and YYMMDDHH indicate the date and hour for which the fields are contained in the file.*

Comment #27: *l. 529 would 'base time' be better than 'starting forecast time'*

Answer: We added the term 'base time' since ECMWF uses it.

Changes in manuscript: *The HH represents the starting time (base time) of the forecast.*

Comment #28: *l. 535 – I may be missing something here but why the file names selected for the ensemble members do not account for FORECAST_STEP?*

Answer: If we retrieve a combination of ensemble members and pure forecast, the naming scheme would be a combination of Sect. 3.9.2 and 2.9.3.

Changes in manuscript: no changes

Comment #29: *l. 548 – should read 'makes'*

Answer: OK

Changes in manuscript: corrected

Comment #30: *l. 581 – I would skip the coma after e.g.*

Answer: OK

Changes in manuscript: changed as suggested

Comment #31: *l. 615 – do you mean 'paths'?*

Answer: Yes.

Changes in manuscript: corrected

Comment #32: *l. 624 – should read 'except'*

Answer: Yes

Changes in manuscript: corrected

Comment #33: *l. 687 – better 'indicates a lower complexity'*

Answer: Ok

Changes in manuscript: changed as suggested

Comment #34: *l. 720 'Lagrangian' should be one word*

Answer: Yes

Changes in manuscript: corrected

Comment #35: *l. 721 – 1990s*

Answer: Ok

Changes in manuscript: corrected

Comment #36: *l. 940 – skip 'an'*

Answer: Ok

Changes in manuscript: corrected

Comment #37: *l. 942 and 943 are too tightly formatted in vertical (also true for l. 848 and 849 and l. 888 and 889 then for lines 966, 967 and 968; subsequently lines 982, 983 and 984 have different vertical formatting than the rest of the manuscript; l. 1062 and 1063 are too close in vertical as well)*

Answer: Yes, we noticed that this comes from the scriptsize environment where we missed to introduce a new paragraph before the following section.

Comment #38: *l. 954 you may want to insert a blank space after the bracket*

Answer: Ok

Changes in manuscript: changed as suggested

Comment #39: *In the caption of Tab.2 'resolution' instead of 'resoltion'*

Answer: Ok

Changes in manuscript: corrected

Comment #40: *In the body of Tab.6 – should be 'Public users have to register for obtaining an account'*

Answer: Ok

Changes in manuscript: corrected

Comment #41: *In the caption of Tab. 18 you need to insert a blank space after etadot*

Answer: OK

Changes in manuscript: corrected

Comment #42: *In the caption of Tab. 24 it is difficult to understand the first sentence starting from: 'Python code'; could you, please, re-phrase. I would also suggest putting sections v7.0.4 and v7.1 of this table side by side and not one on top of the other, if possible (I am aware there is no strict correspondence between theses two sections)*

Answer: We agree to rephrasing the first sentence. But, putting the two versions next to each other does not make sense since the names of the classes and methods are not the same and do not correspond to each other. We think it would be misleading for the reader. Anyway, since we shortened the manuscript, this table is now moved to the supplement.

Changes in manuscript: *Python code blocks and their cyclomatic complexity (CC) which ranks between C and F and their corresponding CC score.*

---

## Author Response (AR2)

Dear Dr. Ullrich,

thank you for handling our manuscript. We are happy that it is now accepted subject to technical corrections. We have implemented them, and provide the following answers to the referee:

Dear anonymous referee #1,

thank you for your second review of our manuscript.
We implemented your suggestions for technical corrections as follows:

1) L6: Remove 'authorised'. This was mentioned in another comment at another location previously. Both types of user are 'authorised' but through different registrations and with different rights.

Answer: Done

2) L31: 'used' seems to be obsolete and misleading here.

Answer: removed

3) L49f: I am not sure how this is generally handled in GMD, but shouldn't the revised manuscript directly refer to the latest release and, hence, also reflect that already in the manuscript title? In which case this sentence would not be required.

Answer: We have added the minor version in the title, and therefore shortened the sentence to "This paper contains the first documentation of flex_extract published in open literature."

4) L256: 'send' instead of 'sent'.

Answer: Done

5) L357: Instead of 'current FLEXPART' please refer to version number. For example: 'FLEXPART up to version xxxx cannot properly handle ...'

Answer: Done

6) L359: Same here: Add the version for which you expect FLEXPART to contain this feature (given in answer but not in manuscript).

[revised manuscript text omitted]